# Preserving Privacy in Large Language Models:
# A Survey on Current Threats and Solutions

**Michele Miranda**                                                   *miranda@di.uniroma1.it*
*Sapienza University of Rome*
*Translated*

**Elena Sofia Ruzzetti**                                    *elena.sofia.ruzzetti@uniroma2.it*
*University of Rome Tor Vergata*

**Andrea Santilli**                                                  *santilli@di.uniroma1.it*
*Sapienza University of Rome*

**Fabio Massimo Zanzotto**                          *fabio.massimo.zanzotto@uniroma2.it*
*University of Rome Tor Vergata*

**Sébastien Bratières**                                        *sebastien@translated.com*
*Translated*

**Emanuele Rodolà**                                                  *rodola@di.uniroma1.it*
*Sapienza University of Rome*

**Reviewed on OpenReview:** https://openreview.net/forum?id=Ss9MTTN7OL

## Abstract

Large Language Models (LLMs) represent a significant advancement in artificial intelligence, finding applications across various domains. However, their reliance on massive internet-sourced datasets for training brings notable privacy issues exacerbated in critical domains (e.g., healthcare). Moreover, certain application-specific scenarios may require fine-tuning these models on private data. This survey critically examines the privacy threats associated with LLMs, emphasizing the potential for these models to memorize and inadvertently reveal sensitive information. We explore current threats by reviewing *privacy attacks* on LLMs and propose *comprehensive solutions* for integrating privacy mechanisms throughout the entire learning pipeline. These solutions range from anonymizing training datasets to implementing differential privacy during training or inference and machine unlearning after training. Our comprehensive review of existing literature highlights ongoing challenges, available tools, and future directions for preserving privacy in LLMs. This work aims to guide the development of more secure and trustworthy AI systems by providing a thorough understanding of privacy preservation methods and their effectiveness in mitigating risks.

# Contents

# 1   Introduction

Large Language Models (LLMs) represent a significant advancement in artificial intelligence (AI), demonstrating remarkable capabilities in understanding and generating human-like text, and fueling innovation across various domains (Brown et al., 2020; Touvron et al., 2023; Zhao et al., 2023). However, their reliance on massive internet-sourced datasets for training raises significant privacy concerns, considering that these models are prone to memorize segments of their training data (Feldman, 2020; Feldman & Zhang, 2020). This is further exacerbated if we consider certain application-specific scenarios (e.g. healthcare) that may require fine-tuning publicly available models on user private data (Tramèr et al., 2024a). Since language models are specifically trained to generate text, this poses a fundamental threat considering that they might inadvertently disclose memorized private data at any moment (Carlini et al., 2019; 2021).

Privacy is a fundamental human right that encompasses the protection of personal information from unauthorized access and misuse (United Nations, 1948; Nissenbaum, 2010). Preserving privacy in the context of language models thus means implementing measures that ensure the protection of personal and sensitive information from being exposed or misused, following the permissions granted by users to use those data, whether implicit or explicit (Brown et al., 2022a). Preserving privacy is not only crucial to preserve users and public trust in AI technologies but also a legal requirement in many countries such as the EU with the General Data Protection Regulation (GDPR), the US with the Privacy Act, and China with the Personal Information Protection Law and Data Security Law (Weidinger et al., 2021; Solaiman et al., 2019; Veale et al., 2018). These regulations may include additional guarantees, such as the "right to be forgotten", which allows individuals to withdraw their consent and request the deletion of their personal data, posing an additional challenge for LLMs as erasing a data point from a trained model might require retraining the model from scratch or discarding the entire model (Zhang et al., 2024b).

Understanding the distinction between privacy and copyright is crucial in this context. Privacy focuses on protecting and controlling personal information, ensuring that individuals' sensitive data remains confidential and is not shared or exploited without their consent. Copyright, on the other hand, pertains to the legal rights of creators over their original works, such as text, music, or art, and governs how these works can be used, distributed, and reproduced by others (Torremans, 2007; Karamolegkou et al., 2023). While privacy is about protecting personal data from unauthorized access, copyright protects intellectual property and ensures rights holders receive recognition and compensation for their work. This study focuses on addressing privacy issues related to the potential exposure of personally identifiable information (PII) in LLMs rather than exploring copyright infringement or intellectual property rights.

> **PII**
>
> Personal Identifiable Information (PII) refers to any data that can be used to identify a specific individual. This may include direct information such as a person's name, address, email address, social security number, phone number, genome, fingerprints, and face, as well as any other data that, when combined, could lead to the identification of an individual (National Institute of Standards and Technology, 2017; General Data Protection Regulation, 2016).

This survey aims to provide a holistic view of the problem of preserving privacy in LLMs by combining the current *privacy threats* and the current *available solutions* and tools to implement them. We believe this approach presents both aspects in tandem and is an essential resource for researchers and practitioners seeking a comprehensive understanding of this field. Specifically, this survey targets (i) readers interested in understanding the current privacy risks associated with LLMs (§3); (ii) readers seeking to learn about the available solutions to defend against privacy attacks (§4 - 5); and (iii) readers looking for tools to make their LLMs resistant to privacy attacks (§6).

The survey is organized as follows and divided into two main categories: *privacy attacks* (§3) and *solutions* (§4 - 5). The taxonomy in Figure 1 provides a structured view of our survey on preserving privacy in LLMs. We provide a preliminaries section for readers unfamiliar with the terminology (§2) that explains the basic concepts related to language models, differential privacy, and federated learning.

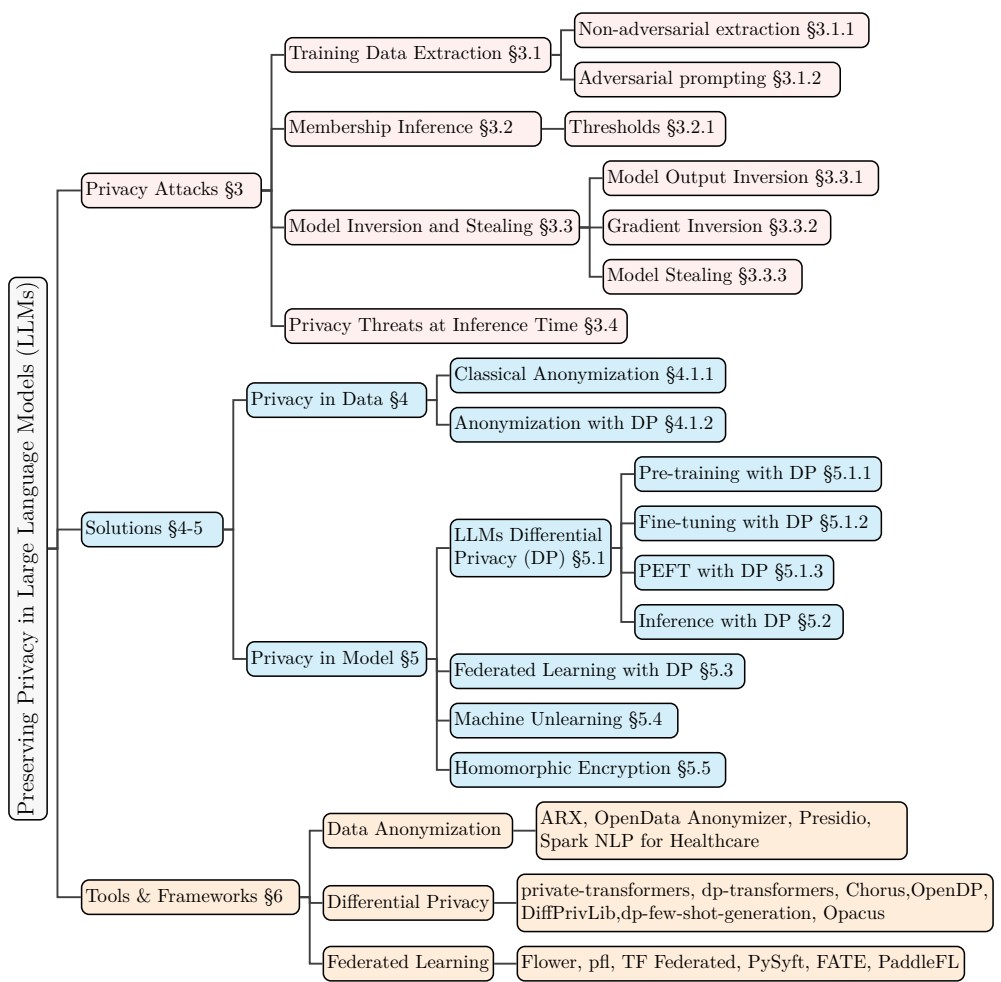

Figure 1: **Taxonomy of Preserving Privacy in Large Language Models.** Red indicates various attack techniques, and blue represents current possible solutions to preserve privacy by acting on the training data or model. Finally, in orange we highlight the currently available tools to preserve privacy.

*Privacy attacks* discuss how it's possible to extract training data from a model and encompass training data extraction (§3.1), membership inference (§3.2), and model inversion and stealing (§3.3). Training data extraction involves techniques such as non-adversarial extraction (§3.1.1) and adversarial prompting (§3.1.2), where attackers can potentially extract sensitive information from the training data. Membership inference (§3.2) utilizes shadow models or thresholds (§3.2.1) to infer whether specific data points were part of the training set. Model inversion includes model output inversion (§3.3.1) and gradient inversion (§3.3.2) methods that can reconstruct the input data from model outputs while Model stealing is proposed to reconstruct a model's parameters from its output (§3.3.3). However, as prompting LLMs with private input – such as in-context demonstrations – is an increasingly popular approach, threats at inference time are also discussed 3.4.

*Solutions* to these privacy challenges are categorized into privacy in data (§4) and privacy in model (§5). Privacy in data includes classical anonymization techniques to anonymize data before it is used in training (§4.1.1) and anonymization with differential privacy (DP) (§4.1.2), which ensures that individual data points cannot be distinguished from the dataset. Privacy in model encompasses the application of DP during different phases such as pre-training (§5.1), fine-tuning (§5.1.2), Parameter-Efficient Fine-Tuning (PEFT) (§5.1.3), and inference (§5.2). Federated learning with DP (§5.3) is implemented to maintain privacy across distributed datasets, and machine unlearning (§5.4) involves techniques to effectively remove specific data

from trained models. Homomorphic Encryption (§5.5) involves cryptographic methods that allow computation on encrypted data without decrypting it. Finally, we present available tools and frameworks to implement privacy in language models (§6).

In summary, this survey serves as a critical resource for researchers and practitioners, providing insights into the current privacy threats in LLMs and the currently available solutions to address these challenges. By systematically categorizing privacy issues and their corresponding mitigation strategies, this paper highlights the importance of preserving privacy in LLMs and paves the way for future advancements in the field. All the papers in this survey are also available at the following GitHub repository[1].

---

[1]https://github.com/michele17284/Awesome-Privacy-Preserving-LLMs

## 2    Preliminaries

In this section, we present preliminary concepts essential for comprehending the subsequent content of the document. Readers already acquainted with these concepts may proceed directly to the next section. For a comprehensive introduction to the concepts of Machine Learning and Deep Learning, we direct the reader to Bishop (2006) and Goodfellow et al. (2016).

In this survey, we use the term large language model (LLM) to indicate any model trained on broad textual data that can be adapted (e.g., via fine-tuning or in-context learning) to a wide range of downstream tasks (Bommasani et al., 2021). While the focus of the survey is mostly on generative LLMs (Zhao et al., 2023)—models that are capable of solving a broad set of tasks by generating text, such as GPT-3 and GPT-4—it is not uncommon to encounter smaller models like BERT in our discussion. We have chosen to include some of these models when we believe that their associated findings, both in terms of potential threats and proposed solutions, remain still relevant and generalizable to larger models nowadays.

### 2.1    Large Language Models

Large language models are deep learning systems trained on vast amounts of not-annotated text data to predict the likelihood of word sequences. Currently, most of the generative language models are trained in an autoregressive fashion, meaning they generate text by predicting each word sequentially based on the previous words in the sequence - usually from left to right. A model trained in this way is capable of generating and manipulating language effectively, which has been shown useful across a broad set of tasks and scenarios (Zhao et al., 2023). However, this was not always the case. Other popular techniques train LMs by masking certain words in a sentence and then predicting those masked words based on the surrounding context, like in the case of BERT (Devlin et al., 2019). These models are typically much smaller and less capable than modern LLMs (for instance, BERT-large has 340 million parameters). Language modelling is not a new task: the groundwork for it was laid in 1948 by Claude Shannon (Shannon, 1948), and the task was extensively investigated for decades, but the recent advances with the introduction of the Transformer architecture (Vaswani et al., 2017) and the rollout of GPT-3 in 2022 (Brown et al., 2020) shed new light on the task and this new class of very big language models, namely large language models. Training a Large Language Model from scratch, sometimes referred to as pre-training, is usually very computationally intensive given the scale of these models and requires several resources generally available only to a few organizations in the world (Sharir et al., 2020; Hadi et al., 2023). For example, to give a sense of this scale, GPT-4 is rumoured to have around 1.8 trillion parameters with an estimated training cost of around $63 million at the time of its release.

With these models, a new training paradigm arose: the pre-training/fine-tuning paradigm. Pre-training is usually on the next token prediction task, while fine-tuning is on the downstream task (which can still be the next token prediction, but it can also be something else like a simple classification task). Organizations building these large language models typically pre-train them on extensive text corpora and then fine-tune them for specific tasks before making them accessible to the public. White-box access allows users to download the model's weights and see or modify the model's internal parameters and architecture, enabling detailed inspection and customization. In contrast, black-box access is usually offered through APIs or web interfaces, where users can input data and receive outputs without any visibility into the model's internal workings or the ability to modify its parameters. There's also a middle ground called grey-box access, which allows users a partial view and limited interaction with the model's internal workings. Users can gain some insights into specific aspects of the model, such as the architecture or certain parameters, but do not have full access to download or modify the model's weights and internal components. This type of access might enable customization and tuning through pre-defined interfaces or tools, offering more flexibility and control than black-box access but not as much transparency or openness as white-box access. Researchers and scientists can efficiently adapt the models to their specific tasks and needs through fine-tuning, which uses a relatively small fraction of the original pre-training data and iterations. Since these are language models, it is possible to interact with them in natural language by feeding them sentences in input, which in this context are called prompts. This means that black box models can still be adapted by conditioning them with specific prompts, either in a zero-shot or few-shot fashion (Brown et al., 2020). This process is known as in-context

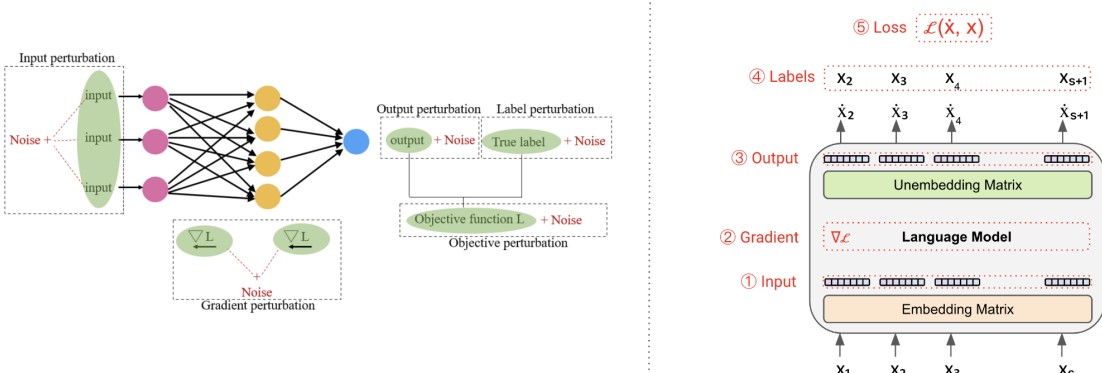

Figure 2: **Differential Privacy.** **One the left:** classical DP applied to a neural network, picture from Baraheem & Yao (2022). **On the right:** DP applied on a Language Model (analyzed in detail in Section 5.1). Differential privacy can be applied by adding a perturbation (noise) on different positions of its architecture: 1) **Input** noise perturbation is added to the input; 2) **Gradient** noise perturbation is added during training to the gradient update step; 3) **Output** noise perturbation is added to the output; 4) **Labels** noise perturbation is added to the labels used during training; 5) **Loss** noise perturbation is added to the loss function during training.

learning: in a zero-shot setting, the LLM takes only an instruction as input, while in a few-shot setting, it also takes a few examples, significantly enhancing its conditioning.

Here, we offer the reader a comprehensive explanation of several terminologies frequently used in the context of language models throughout the paper. These explanations aim to clarify key concepts and enhance understanding of the material presented:

- Sampling: LLMs are trained to predict the next token (word or punctuation) when given a text (prompt). To do so, LLMs learn a probability distribution over every possible token. Sampling consists of selecting the next word from this distribution, given the current context. Various methods, such as greedy sampling, random sampling, top-k sampling, and top-p sampling, allow for different balances between coherence and creativity in the output. These techniques are essential for tailoring the model's responses, impacting the diversity and appropriateness of text generation across applications like chatbots and content creation.

- Instruction tuning: Instruction tuning is a technique used to enhance the performance of large language models (LLMs) by training them to follow specific instructions. It involves fine-tuning a pre-trained model on a dataset that includes tasks described in natural language, along with the desired outputs. This technique allows the model to shift from general next-token prediction to more specialized tasks (e.g., text summarization). By using instruction tuning, the model learns to interpret and execute tasks based on user-given instructions, thereby improving its ability to generate relevant and accurate responses given a user prompt.

- PEFT (Parameter Efficient Fine-Tuning): Parameter Efficient Fine-Tuning is a class of methods to adapt pre-trained language models by altering only a small fraction of the model's parameters. This approach maintains the general capabilities of the model while enabling customization to new requirements with minimal computational cost and memory overhead. It's particularly useful for deploying models in resource-constrained environments or for applications requiring rapid adaptation to new data.

- RLHF (Reinforcement Learning from Human Feedback): Reinforcement Learning from Human Feedback is a technique used to train large language models (LLMs) by incorporating human preferences into the learning process. After a model has been instruction-tuned, its outputs are evaluated by

humans, who provide feedback on the quality and relevance of the responses. This feedback is used to fine-tune a reward model that assigns scores to the outputs based on human preferences. The main model is then fine-tuned using reinforcement learning, guided by the reward model, to better align with human values and enhance its performance on tasks requiring nuanced understanding and judgment.

## 2.2 Differential Privacy

According to Dwork & Roth (2014): ***"Differential privacy*** *describes a promise, made by a data holder, or curator, to a data subject: "you will not be affected, adversely or otherwise, by allowing your data to be used in any study or analysis, no matter what other studies, data sets, or information sources, are available."* The fundamental goal of Differential Privacy is to enable the extraction of useful insights from data while minimizing the potential impact on any individual's privacy within that dataset. The core idea is to introduce controlled noise or randomness into the analysis or results in a way that prevents the identification of specific individuals. In simpler terms, when a system or algorithm is differentially private, the inclusion or exclusion of any individual's data should not significantly affect the overall outcome or conclusions of the analysis. There are several kinds of Differential Privacy definitions, but the original one is $(\varepsilon, \delta)$-Differential Privacy.

---

**$(\varepsilon, \delta)$-Differential Privacy**

A randomized algorithm $\mathcal{M}$ with domain $\mathbb{N}^{|\mathcal{X}|}$ is $(\varepsilon, \delta)$-differentially private if for all $\mathcal{S} \subseteq \mathrm{Range}(\mathcal{M})$ and for all $x, y \in \mathbb{N}^{|\mathcal{X}|}$ such that $\|x - y\|_1 \leq 1$ :

$$\Pr[\mathcal{M}(x) \in \mathcal{S}] \leq \exp(\varepsilon) \Pr[\mathcal{M}(y) \in \mathcal{S}] + \delta$$

---

The formal guarantee provided by $(\varepsilon, \delta)$-Differential Privacy is that for any two datasets x and x' that differ by only one element (these are often referred to as "adjacent datasets"), and for any subset of outputs S of the algorithm's range Y, the probability that the algorithm outputs something in S given x is not substantially higher than given x'.

$\epsilon$ is the parameter that measures the privacy loss associated with a query in the system. A smaller $\epsilon$ provides stronger privacy guarantees. The use of $\epsilon$ in the inequality indicates that the probability of a certain output does not increase by more than a multiplicative factor of $e^\epsilon$ when any single individual's data is added or removed.

The privacy budget is the total loss of privacy sustained by the system during the execution, and it can be computed by compounding the privacy loss indicated by $\epsilon$ over the total number of executions/iterations. This is done with the composition theorems, which are explained in more detail later in this section.

$\delta$ is the parameter that allows for a small probability of failure of the privacy guarantee. Ideally, $\delta$ should be very small, close to 0, which indicates a lower chance of the algorithm failing to maintain privacy. When $\delta$ is 0, it is defined $\varepsilon$-differential privacy, which is the strongest kind of differential privacy.

This means that for all subsets S of the output range Y, the probability that the algorithm on input x gives an output in S is at most e raised to the power $\epsilon$ times the probability that the algorithm on input x' gives an output in S, plus the small probability $\delta$.

The simplest method to achieve differential privacy in a query is by adding random noise to its answer. The challenge lies in balancing the noise added to meet differential privacy standards without rendering the answer too noisy to be useful. To assist with this, foundational mechanisms have been developed in the field, detailing the type and amount of noise required. One such mechanism is the Laplace mechanism.

**Laplace Mechanism**

According to the Laplace mechanism, for a function $f(x)$ which returns a real number, the following definition of $F(x)$ satisfies $\epsilon$-differential privacy:

$$F(x) = f(x) + \text{Lap}\left(\frac{\Delta f}{\epsilon}\right)$$

where $\Delta f$ is the sensitivity of $f$, and $\text{Lap}(b)$ denotes sampling from the Laplace distribution with centre 0 and scale $b$.

Sensitivity $\Delta f$ refers to the maximum change in the output of a function or query when a single individual's data in the input dataset is added or removed. It quantifies how much a query's result can vary with the smallest possible change in the dataset, which is crucial in determining the amount of noise that needs to be added to ensure differential privacy. A function with high sensitivity requires more noise to protect individual privacy, while a function with low sensitivity requires less. Sensitivity is a key factor in calibrating the noise for mechanisms like the Laplace mechanism to ensure that individual data points do not significantly affect the output, thereby maintaining privacy.

**Global Sensitivity**

For a function $f : \mathcal{D} \to \mathbb{R}$ mapping datasets ($\mathcal{D}$) to real numbers, the global sensitivity of $f$ is defined as follows:

$$GS(f) = \max_{x,x':d(x,x')<=1} |f(x) - f(x')|$$

Here, $d(x, x')$ denotes the distance between two datasets $x$ and $x'$, with the notion that two datasets are neighbours if their distance is 1 or less. The way this distance is defined significantly impacts the privacy definition we achieve, and we'll explore the dataset distance metric in more detail later.

Global sensitivity is defined such that for any two neighbouring datasets $x$ and $x'$, the difference between $f(x)$ and $f(x')$ does not exceed $GS(f)$. This is referred to as "global" sensitivity because it is independent of the specific dataset being queried, applying to any pair of neighbouring $x$ and $x'$.

Based on the noise added, there are several mechanisms in the literature. The Gaussian mechanism is an alternative to the Laplace mechanism, which adds Gaussian noise instead of Laplacian noise.

**Gaussian Mechanism**

The Gaussian mechanism does not satisfy pure $\epsilon$-differential privacy but does satisfy $(\epsilon, \delta)$-differential privacy. According to the Gaussian mechanism, for a function $f(x)$ which returns a number, the following definition of $F(x)$ satisfies $(\epsilon, \delta)$-differential privacy:

$$F(x) = f(x) + \mathcal{N}(0, \sigma^2)$$
$$\text{where } \sigma = \frac{\Delta f \cdot \sqrt{2\log(1.25/\delta)}}{\epsilon}$$

where $\Delta f$ is the sensitivity of $f$, and $\mathcal{N}(0, \sigma^2)$ denotes sampling from the Gaussian (normal) distribution with mean 0 and variance $\sigma^2$. Note that here (and elsewhere in these notes), log denotes the natural logarithm.

The primary mechanisms we've discussed (Laplace and Gaussian) are designed for numerical outputs, adding noise directly to the results. But what if we need to provide an exact answer without injecting noise yet still maintain differential privacy? The exponential mechanism addresses this by enabling the selection of the "best" element from a set while ensuring differential privacy. The "best" element is identified by using a scoring function that assigns scores to each item within a given set of options. The exponential mechanism ensures differential privacy by approximately maximizing the score of the chosen element. To preserve privacy, it might sometimes select an element that doesn't have the highest score.

> **Exponential Mechanism**
>
> The exponential mechanism satisfies $\epsilon$-differential privacy:
> 1. A set $\mathcal{R}$ of possible outputs is selected
> 2. A scoring function is specified $u : \mathcal{D} \times \mathcal{R} \to \mathbb{R}$ with global sensitivity $\Delta u$
> 3. The exponential mechanism outputs $r \in \mathcal{R}$ with probability proportional to:
>
> $$\exp\left(\frac{\epsilon u(x,r)}{2\Delta u}\right)$$

The main practical distinction between the exponential mechanism and previous mechanisms like the Laplace mechanism is that the exponential mechanism's output is always an element from the specified set. This is particularly advantageous when choosing an item from a finite set, where a noisy result would be inappropriate.

Another prominent definition of Differential Privacy, which is actually a relaxation of the original definition, is Rényi Differential Privacy (Mironov, 2017).

> **Rényi Differential Privacy**
>
> A randomized algorithm $\mathcal{M} : \mathcal{D} \to \mathcal{R}$ is said to have $\epsilon$-Rényi differential privacy of order $\alpha$, or $(\alpha, \epsilon)$-RDP for short, if for any adjacent $D, D' \in \mathcal{D}$ it holds that:
>
> $$D_\alpha(\mathcal{M}(D)\|\mathcal{M}(D')) \le \epsilon$$

Here, $D_\alpha(P\|Q)$ represents the Rényi divergence of order $\alpha$ between two probability distributions $P$ and $Q$, and is defined as:

$$D_\alpha(P\|Q) = \frac{1}{\alpha - 1} \log\left(\sum_{x \in \mathcal{X}} P(x)^\alpha Q(x)^{1-\alpha}\right)$$

for all $\alpha > 1$, where $\mathcal{X}$ is the support of the distributions $P$ and $Q$. The parameter $\alpha$ here controls the sensitivity of the divergence measure to differences between $P$ and $Q$, with higher values of $\alpha$ focusing on events that are more probable under $P$ than under $Q$. The parameter $\epsilon$ still quantifies the privacy guarantee, with smaller values indicating stronger privacy.

By providing a mathematically rigorous definition of privacy guarantees, Differential Privacy has become a key principle for designing privacy-preserving algorithms, especially in scenarios where personal data is involved. From this mathematical definition, some interesting properties arise. Among these, there are composition theorems that allow the computation of the exact privacy loss through many applications of the DP algorithm, which corresponds to the privacy budget. Sequential composition theorem (Kairouz et al., 2015) applies when multiple differentially private mechanisms are applied to the same dataset in sequence. The theorem quantifies how the privacy guarantees accumulate over the sequence of operations. Parallel composition theorem (McSherry, 2009) applies when multiple differentially private mechanisms are applied to disjoint subsets of the dataset. This theorem shows that the worst-case privacy guarantee of any single mechanism governs the privacy guarantee of the overall process. Differential Privacy also exhibits a Post-Processing property that states that once data has been processed through a DP mechanism, the output retains its privacy guarantees regardless of any further processing.

Differential privacy was born as a statistic analysis technique for databases. Initially, it was used to query databases without compromising the privacy of anybody enrolled, and that is why we discussed queries and compounding privacy over them to compute the total privacy loss or privacy budget. Luckily, all this came in handy in recent years, with the worldwide diffusion of deep neural networks, when differential privacy was applied to them. That is when databases became datasets and queries became (mostly training) iterations. We will see more in detail what is the result of this transition in the next paragraph §2.2.

### 2.3 Deep Learning with Differential Privacy

Differential privacy was initially devised for database analysis, but it is possible to apply it to deep learning models in several ways. In practice, since we primarily work with deep learning models, we will often refer to datasets rather than databases. However, this distinction does not entail any practical difference in the context of our discussion. Figure 2.1 shows how Differential Privacy can be incorporated in 1) the input data, 2) the gradients, 3) the output data, 4) the labels or even in the loss function 5). The most used technique is by training using DP-SGD (Differentially Private Stochastic Gradient Descent), while DP-Adam is less used - we discuss the reasons for this in §5.1.2. Both of these techniques involve gradient manipulation, falling under point 2) in Figure 2.1. Initially presented by Song et al. (2013), DP-SGD is a variation of standard Stochastic Gradient Descent (SGD) that enhances privacy by incorporating gradient clipping and adding Gaussian noise. In DP-SGD, gradients are first clipped to a maximum norm to limit the influence of any single data point. Then, Gaussian noise is added to the average of these clipped gradients, ensuring that the updates to the model parameters maintain differential privacy by obscuring the contribution of individual data points. It is possible to consider the several layers separately in order to have different clipping thresholds, and the amount of noise should be calibrated over the sensitivity of the function (in this case, the network), as explained by Dwork et al. (2006b). The sensitivity of a function $f$ indicates how much the function's output changes in response to changes in its input. In DP, there is also a focus on privacy costs, so DP-SGD needs a way to compute them. In Abadi et al. (2016), along with a refined version of DP-SGD, the authors present a novel technique for privacy accounting during training, which is called moments accountant. The composability property of DP allows us to compute the privacy cost at every iteration and then sum them up. Privacy cost is computed in an asymptotic way, and while the mathematical details are beyond the scope of this survey, it is worth noting that moments accountant saves a non-negligible factor in the computation with respect to previous techniques, allowing to guarantee a lower privacy budget with the same magnitude of injected noise.

Also gradient clipping is an important step in the DP learning process. Gradient clipping can be done in several ways. It is possible to use a per-sample clipping (Lee & Kifer, 2021) and this can even be done automatically (Bu et al., 2023a), or a flat clipping style (also called all-layer) (Abadi et al., 2016) or something in between, like group-wise clipping (McMahan et al., 2017b). As stated by Bu et al. (2024), the first leans towards better memory accuracy while the second favours memory efficiency. Another option to enforce DP is PATE (Private Aggregation of Teacher Ensembles) (Papernot et al., 2017) and its variations (Papernot et al., 2018; Tian et al., 2022). In this method, several teacher models are trained on private data, and then a student model is trained on the predictions given by a majority vote with added noise made by the ensemble on public data. In this way, the actual model never sees the data, and the individual models (and, for extension, their data) cannot influence the outcome too much. A new, alternative way to enforce DP, as shown by Du et al. (2023b), is to perturb only forward pass embeddings, and according to the experimental evidence, it offers stronger performance with a lower vulnerability to attacks and more efficient memory usage. DP has been shown to work in several settings like NLU (Qu et al., 2021; Dupuy et al., 2022), image classification (De et al., 2022), diffusion models for image generation (Dockhorn et al., 2023; Ghalebikesabi et al., 2023), neural machine translation (Igamberdiev et al., 2024) and, of course, language modelling (Dinh & Fioretto, 2023) and LLMs in particular (Carranza et al., 2024). All these related works prove the interest of the scientific community in the problem and the commitment to finding an effective solution. Differentially private models are already in place with several solutions, and they work to different extents based on models, data and privacy budgets.

**Federated Learning** Federated learning (McMahan et al., 2017a) is a machine learning approach that enables model training across decentralized devices or servers holding local data samples without exchanging them. Instead of centralizing all data in one location, the model is trained collaboratively across multiple devices or servers. Each local device computes updates to the model based on its local data, and only these updates are shared and aggregated to improve the global model. Federated learning aims to preserve privacy and security by keeping sensitive data localized while still benefiting from the collective knowledge gained during the collaborative training process. It allows to be involved in the training of a model without fully trusting the external entity training the model (which can be a government or a company), because you are not giving up your actual data. As we will see more in detail later, this technique does not really manage to

protect privacy by itself because gradient updates, even after aggregation, can be reversed to recover training data §3.3.2. A big advantage of federated learning is allowing to train better models thanks to the higher availability of high-quality local data, like the case of next word prediction for smartphone keyboards (Hard et al., 2018).

**Secure Multi-Party Computation**  Secure Multi-Party Computation (SMPC) (Yao, 1982; Zhao et al., 2019a) is a communication protocol that enables several parties to jointly compute a function over their private inputs, without revealing those inputs to each other. This way, each participant's data remains confidential while still allowing for the collective computation of the function. It is particularly useful in scenarios where data privacy is crucial, yet there is a need for collaborative computation, such as in privacy-sensitive data analysis and secure voting systems. There are several ways to enforce SMPC, like secret sharing (Shamir, 1979; Blakley, 1899), homomorphic encryption (Gentry, 2009; Kim & Yun, 2021; Joo & Yun, 2014), zero-knowledge proofs (Goldreich & Oren, 1994; Feige et al., 1988) and oblivious transfer (Rabin, 2005; Even et al., 1985; Brassard et al., 1986; Chailloux et al., 2013; Cheong et al., 2009). Homomorphic encryption is particularly interesting in this context as it allows the computation of a function on encrypted data. This makes it possible to perform training and inference of machine learning models on encrypted data. While these techniques can enforce strong data security, they also add a considerable computational overhead that makes them difficult to apply in situations where the learning process is already heavy.

# 3 Privacy Attacks

As the training data becomes huge, it is increasingly urgent to understand whether models tend to expose the data on which they have been trained since they are more likely to capture sensitive information. Moreover, the leakage of the original training data is not the only possible threat. In fact, the versatility of LLM is enhanced by the use of in-context demonstrations of the task that a user wants to perform (Brown et al., 2020). While being a useful technique, unconscious use can cause sensitive information – that should not be disclosed – to be included in these examples in context as well. For commercial systems, those data can also be leaked by exploiting vulnerabilities of the systems storing them (OpenAI, 2023). Even cyber security measures are insufficient since those data can be used in further training steps and subsequently leaked by a future version of the model. This aspect is also concerning for data with commercial value from a business perspective (Coles, 2023), and companies have decided, in some cases, to warn their employees about the responsible use of LLMs (Kim, 2023).

A broad research area is devoted to understanding whether deep learning algorithms memorize training examples and, hence, can be used to reconstruct (potentially sensitive) training information. While some works question the effect of memorization during generation – meaning that texts generated starting from prompts extracted from training are different from the original ones (McCoy et al., 2023) – the discussion around memorization and its consequent risks is growing as larger models become more popular. In some cases, memorization is discussed as the key to good performance since long-tailed distributions often characterize data. Hence, for a model, it is hard to distinguish between outliers and actually informative data points (Feldman & Zhang, 2020). However, memorization of training data by a model allows an attacker to reconstruct sensitive information or to understand whether a certain example was used during training. It has been shown by Carlini et al. (2023a) that it is possible to reconstruct training data with only black-box access.

For LLMs, it might be sufficient to input the right prompt, which is not necessarily an adversarial one (Carlini et al., 2021), in order to condition a model to generate sequences that are exactly memorized from the training data. Attacks that allow verbatim extraction of training data are generally referred to as Training Data Extraction attacks. When LLMs are fine-tuned with Reinforcement Learning with Human Feedback (RLHF), they tend to be more robust and avoid revealing the training set examples and, in particular, revealing sensitive information. However, some adversarial prompts have emerged to condition a model to show some unintended behaviour. Since sensitive information is memorized by a model, RLHF is used to prevent the model from disclosing such information, but it does not ensure that new attacks cannot retrieve it. We discuss both non-adversarial and adversarial prompting in §3.1.

In some cases, an attacker may not only have access to a model but also to some data possibly included in the training phase of the target model. In this scenario, a Membership Inference Attack (MIA) can be performed: the attacker aims to claim that the information is included in the training set of a threatened model. The initial formulation of those attacks (Shokri et al., 2017) has also been extended against larger models and Transformer-based Language Models in particular. Since some sequences are memorized by a model, overfitting over them can be detected and used to attest to the presence of certain data in the original training data. The effectiveness of MIA is also discussed in light of the training practices generally used to develop LLMs, such as training for a single epoch on a huge amount of data, which may influence the measure of overfitting used in these attacks. In §3.2, we discuss that kind of attack.

When private data owned by different holders need to be used to train a joint model on them, protocols like federated learning (§2.3) seem to be the appropriate solution: in this scenario, data are not distributed across multiple servers, only model snapshots and updates are shared. However, model outputs and gradients can also be used to infer the input that justifies the observed model behaviour. The model outputs or gradients can be inverted to obtain the sequence that justifies them, allowing an attacker to reconstruct private data without them being explicitly shared. We discuss Model Inversion Attacks as a method to track back model input from the model's output or gradients in §3.3). Moreover, from those rather opaque sources, like model outputs, an attacker can gain information regarding a model's parameters: those attacks, called Model Stealing attacks, while still not able to reconstruct entire LLMs, can potentially be directed against API-gated models, making them less opaque and, hence, more vulnerable against privacy attacks.

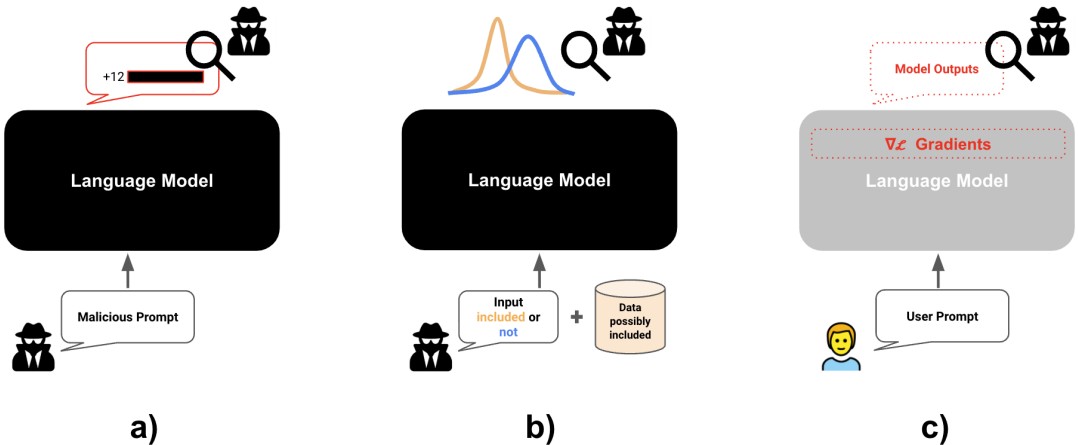

Figure 3: Attacks against LLMs aim to reconstruct private training data. **Training Data Extraction Attack (a):** The attacker has no internal access to the model (black box) and can only design a malicious prompt to force the model generation to reveal private and potentially sensitive training data. **Membership Inference Attacks (b):** The attacker has access to data that might have been included in the training set. The goal is to determine whether specific data points were indeed part of the training set. This is achieved by detecting behavioural differences in the model's responses to data that were included in the training set versus data that were not. **Model Inversion Attack (c):** In a black-box or grey-box setting, the attacker observes the model's outputs or gradients on unknown inputs. The attacker then uses this information to reconstruct the original input from these seemingly opaque sources.

Finally, we notice that since fine-tuning is more expensive than in-context learning, many systems use in-context demonstration to specialize an LLM on specific tasks. Despite the flexibility of this method, some risks may emerge from using private information for in-context demonstrations. For example, a system that answers a user query by leveraging a prompted LLM could be vulnerable to privacy attacks performed by a malicious user (see §3.4 for further details.)

## 3.1 Training Data Extraction

In Training Data Extraction attacks, an adversary aims to reconstruct training data having only access to a target Language Model. In particular, those attacks aim to extract the entire example by querying the target model (Carlini et al., 2021). Retrieval of examples is based on the assumption that the target models memorized them, and the definitions of memorization underlying these attacks may vary. However, the various definitions revolve around the key concept that a textual example or part of it can be extracted from a language model. In the original definition of Carlini et al. (2021), an example is defined to be extractable if there is a prefix used to condition the generation of the model so that the string can be exactly generated from the model. This definition is first introduced by Carlini et al. (2021) and then largely adopted with slight modifications (Huang et al., 2022; Carlini et al., 2023a; Nasr et al., 2023).

> **Training Data Extraction**
>
> Extracting Training Data consists of conditioning an LM to the right prompt so that it will likely generate a training example to complete it. Formally, a textual example $x$ is extractable from a LM $g$ if there exists a prompt $p$ such that: $x = \arg\max_{x'} g(x'|p)$. Once the right prompt has been identified, this procedure can be used to reconstruct training data, and hence, it can be used to extract sensitive information like Personally Identifiable Information (PII) from a model.

### 3.1.1  Non-adversarial extraction

Using training data extraction techniques, it has been shown that Language Models are prone to reveal private information when prompted with sentences requiring disclosing such information. While trying to recover memorized training examples, Carlini et al. (2021) demonstrated that GPT-2 tends to memorize personal information such as Twitter handles, emails, and UUIDs (Universal Unique Identifiers). In particular, in their experiments, the language model generation is conditioned to some prompts that the attacker predefined: to maximize the probability of success of the attack, they choose prompts that are likely to be in the training data, like samples from the Common Crawl [2]. Then, once the generation is complete, the attacker tries to figure out which sequences are most likely to be generated as a result of the memorization, that is, which of the generated strings exactly matches the training data. In order to identify sequences that are more likely to be generated because of memorization, Carlini et al. (2021) compare the likelihood that is assigned to each sequence by two models: one is the model under attack – GTP-2 – and the other is a reference model, another LM. The attacker retains as possibly memorized sentences only those that have a high likelihood in the target model when compared to the likelihood assigned by the reference model. This is a form of Membership Inference Attack §3.2. Finally, once the attacker has retained a subset of strings that are likely to be generated due to memorization, the success of the attack must be verified by checking whether these high-likelihood sequences are actually part of the training set. However, for this step, Carlini et al. (2021) could only manually verify the effectiveness of the attack by searching the Internet for matching sequences, as the tested model did not have accessible training data.

The choice of the right prompt is crucial in this setting: in fact, when prompted with a prefix already seen during training, models tend to complete the prompt with the rest of the training sequence (Carlini et al., 2023a). Huang et al. (2022) further explore this direction and show that conditioning a model to a prompt already observed during training can lead to a leakage of PII, like emails, and that is more effective than designing new, previously unseen prompts. In particular, they distinguish between memorization and association. A model memorizes personal information if a prompt exists- already part of the training data for that model- that leads the model to generate personal information. The association requires that a prompt – not necessarily observed during training and containing some reference to an individual – exists that can lead to the generation of PII by the model. Huang et al. (2022) show that the former approach is more effective than the latter, quantifying the number of e-mails that a model from the GPT-Neo family can predict when conditioned by the two different kinds of prompts. Moreover, they notice how these attacks get more effective as the size of the models increases, and – as also noted in as also noted in Carlini et al. (2023a) and Nasr et al. (2023)–larger models tend to be more exposed to training data extraction attacks.

Those attacks can also be used to reconstruct the dataset used to fine-tune the model. To reconstruct personal information from a generative language model like GPT-2, Lukas et al. (2023) propose to solve a masked language modelling task to reconstruct masked personal information from a sentence, conditioning a model on both the prefix and suffix of the masked information: they call this approach PII Reconstruction. However, the attacker must gain access to masked training data before performing the attack. Panchendrarajan & Bhoi (2021) reconstruct the training data of a fine-tuned GPT-2. This work stems from the observation – also shared by Membership Inference Attacks as described in §3.2 – that sentences that have been used in the fine-tuning phase will be given different probabilities by the fine-tuned version of a model and by the base model. They generate candidate sentences by starting the generation of the fine-tuned model with an empty string. Then, they observe the generation of the base model over the same sentence. Results show that it is possible to reconstruct about 75% of informative words in the fine-tuning dataset.

These experiments were the foundation for exploring other datasets and larger models. Nasr et al. (2023) demonstrate that the method proposed by Carlini et al. (2021) is more effective than initially thought: querying open-source models, they were able to verify the success of the attack procedure by accessing the training data only to verify the attack. They performed the attacks on models belonging to GPT-Neo, Pythia and RedPajama-INCITE families and using as initial prompts samples from Wikipedia. Wang et al. (2024a) explore the leakage of training data of OpenAI's GPT closed models evaluating the accuracy of information extraction of sensitive information contained in pre-training data. The private information they

---

[2]https://commoncrawl.org/

are able to extract is emails from the Enron Email dataset (Klimt & Yang, 2004). They hypothesize that this dataset is used during the training phase of GPT-3.5 and GPT-4 since it is included in the Pile dataset (Gao et al., 2020): while no information is available on the training set of those models, the Pile was used in the pre-training of the GPT-Neo family of models and hence, possibly, also in the closed models under investigation. They adopt the same prompts proposed by Huang et al. (2022) and show results also in few-shots prompting settings. The experiments demonstrate that GPT-3.5 and GPT-4 can predict, respectively, around 5% and 4% of the email addresses accurately. Moreover, in the few-shot prompting setting with emails extracted from the same dataset, these models accurately predict up to 48.19% of emails correctly. The same trend is not observed for prompts that contain emails that have a different domain from the target one, suggesting the importance of the right choice of demonstrations for training data extraction attacks. While LLMs are generally more resistant to attacks where association is required, the work of Shao et al. (2024) shows that it is possible to recover 3% of the training emails from a 20 billion parameter model with attacks that require association, meaning that the extraction procedure does not rely on prompts already observed during training. It has also been proposed to extract targeted examples from a model pre-training data: Zhang et al. (2023b) propose an attack to recover a certain suffix given a precise prefix known to be in the pre-training data. They suppose that the attacker can also obtain some pairs of prefixes and suffixes and train an attack model on them. They tune soft prompt embeddings and then estimate the confidence that a generated suffix is, in fact, as it is in the corresponding prompt in the pre-training data. A similar approach, based on soft prompt tuning, is also adopted by Ozdayi et al. (2023). As the size of the models increases, this type of attack appears to be more effective (Nasr et al., 2023). Moreover, the same techniques have also been discussed for other models, like diffusion models (Carlini et al., 2023b).

As concerns about the potential leakage of private information via training data extraction attacks grow, a number of tools are being investigated to assess and potentially prevent the leakage of private information by LMMs before it spreads. On the evaluation side, ProPILE (Kim et al., 2023) is a tool designed to test models with black-box and white-box attacks against their tendency to release PII: the black-box approach is designed to allow data subjects to test whether their PII is leaked; the white-box approach is based on soft prompting and allows providers to conduct their model surveys to prevent leaks of private information. Differential privacy (see §5 for further details) is also effective against the memorization of private information and can successfully defend against these attacks. Downey et al. (2022) demonstrate that by hand-crafting canary sentences containing sensitive content in the training data, the last two words of each canary are designed to contain the most sensitive information, and the LM is asked to complete the canary. The GPT model they train with Differential Privacy – with some loss in model performance – successfully preserves privacy.

### 3.1.2 Adversarial prompting

There have been initiatives to ensure LLMs are aligned with human values and used appropriately. For example, the OpenAI usage policy requires that users do not use the models to compromise the privacy of others, including collecting or generating personal data. Moreover, OpenAI's ChatGPT has been aligned with Reinforcement Learning from Human Feedback (Ouyang et al., 2022) and avoids answering questions regarding its training data. In this scenario, the techniques based on non-adversarial prompts may not be effective: Nasr et al. (2023) discuss how ChatGPT tends not to expose training data when prompted with prefixes that are likely seen during training. For these reasons, some attacks against LLMs define adversarial prompts that force the model to diverge from its originally intended use. For example, asking the model to repeat a word indefinitely, Nasr et al. (2023) notice that ChatGPT diverges and starts emitting training data, also containing personally identifiable information.

A new form of adversarial prompt attacks is jailbreak prompts. These prompts are designed to circumvent safeguards and manipulate LLMs into generating harmful content and are largely used agaist closed LLMs alligned via RLHF. In some cases, a goal hijacking is performed, meaning that the original goal of a prompt is changed to a new goal, like generating a target phrase, by injecting malicious text in the original prompt (Perez & Ribeiro, 2022). Shen et al. (2024) characterize a large set of jailbreak prompts and evaluate their effectiveness against different LLMs. They notice that most of these prompts encourage the model to act like another fictional assistant, such as DAN (short for Doing Anything Now), which has no limitation in

the answers. Along with other potentially harmful scenarios, they show that this type of prompt allows a user to receive suggestions on how to collect personal information.

In this context, Li et al. (2023c) focused on the usage of adversarial prompts to pose privacy threats: they propose a multi-step jailbreak prompt to extract personal information, such as emails, based on Chain-of-Thought (Kojima et al., 2024). They study the effect of adversarial prompting against ChatGPT. They first notice that the model tends to avoid sharing private information if direct data extraction prompts are used. Then, they propose the Multi-step Jailbreaking Prompt in three steps: they assume the role of the user to input the jailbreaking prompt, then act as the assistant to confirm that the jailbreak mode is enabled Lastly, they play the role of the user and query the assistant with a non-adversarial prompt. They show that more frequent emails are less difficult to recover: experiments on the Enron dataset show that more than 50% of frequent emails in the dataset can be extracted.

## 3.2 Membership Inference Attacks (MIA)

Membership inference attacks (MIA) (Shokri et al., 2017) are currently a de facto standard when it comes to verifying model privacy (Murakonda & Shokri, 2020). In this kind of attack, the attacker has access to (1) **some records** (supposed to come from a similar distribution of the training data) and (2) to **the model**, which could be provided as an API, in a black box setting, or be fully available as a white box. The aim of the attacker is to find out if these records (1) are actually part of the training set for that model (2). The initial formulation of this attack by Shokri et al. (2017) required the attacker to build several "shadow models" that mimic the original model, trained on datasets that may resemble the distribution of the original one. Then, the attacker trains a classifier on the shadow models' outputs to assess the usage of some records during the training phase by the shadow models (where the ground truth is known since the attacker also designed the training set of those models). The classifier, trained on shadow models' outputs, is then used against the original model with the objective of identifying data that are part of its training set.

> **MIA as Classification of Behavioural Changes**
>
> MIA can be framed as a classification problem: the attacker determines whether a certain example has been used during the training phase of the model or not. Attacks are based on detecting changes in the target model when it is exposed to data points that are in the training set versus when it is exposed to examples that are not part of it.

Stemming from the general setting described in Shokri et al. (2017), MIAs have been successfully tailored for Language Models to extract sensitive information from them (Song & Shmatikov, 2019; Hisamoto et al., 2020). Carlini et al. (2022) observe that MIAs could be better framed as the procedure of comparing models that are trained on a given example and models that are not trained on it, and to find a difference in their behaviour, for example, having a likelihood of the example much higher in the first case rather than in the other. They train numerous shadow GPT-2 models to measure the probability of observing a certain likelihood of an example in models trained and not trained on it.

However, training shadow models may be too expensive in the context of LLMs: the original formulation of MIA can be used as long as it is feasible for the attacker to construct shadow models that are similar to the target model. For this reason, in the context of LLM the formulation for MIA has been adapted, generally eliminating shadow model developments.

### 3.2.1 MIA with Thresholds

Since the construction of shadow models can be too expensive, MIAs have also been formulated without relying on them. In fact, some MIAs against LLMs have been designed to exploit some measure of overfit over the training data in order to understand whether a given example has already been seen by a model: overfitting the training data is a sufficient condition to detect memorized examples (Yeom et al., 2018). Shi et al. (2024) observe that the probability distribution is more spread over low probability words for an unseen example: they design attacks that threshold the average log-likelihood of the top-k rarest words to ascertain whether or not the example is part of the training data. Language models trained on clinical

data have been shown to be sensitive to MIAs. Mireshghallah et al. (2022) demonstrate the effectiveness of MIAs designed against Language Models trained with Masked Language Modeling objectives, obtaining a substantial gain both in AUC and in recall of the attack over the baseline. Their method is based on thresholding the ratio between the likelihood of a sentence on the target model and the likelihood of the same sentence on a model trained on a general population. While their experiments are conducted on Transformer-based networks trained on Masked Language Modeling objective, the method proposed can be, at least in principle, generalized also to the generative context. Mattern et al. (2023) argue that the limited availability of training data for the reference model poses a strong restriction on the applicability of previous work. They propose to generate perturbations of a sentence and to measure the average loss of a target model on the original sentence and on the perturbations: they hypothesize that if the sentence is not in the training set, the model should behave similarly also on the perturbed examples, while it should have a lower loss conditioned on the original sentence if it is part of the training set. These kinds of attacks are also useful to understand, given two snapshots of a model – for example, a pre-trained model and a fine-tuned version of it – on which data the fine-tuning phase was performed. In particular, the difference between the probability assigned from different snapshots to each token in a sentence can help identify whether an example was used during later training phases (Zanella-Béguelin et al., 2020). Duan et al. (2023) show that LLMs, while being extremely versatile in a number of tasks thanks to their in-context capabilities, can be subject to MIAs that target datasets used during prompt training. They demonstrate that an LLM trained to answer prompts exposes more information than the corresponding fine-tuned model.

However, a recent body of research poses some critical challenges to the use of MIAs to assess their reliability in demonstrating the inclusion of a sample in LLM training data. In some cases, the theoretical foundation of MIA is discussed. Zhang et al. (2024c) argue that, since it is no longer possible to use shadow models that do not include the data point in their training set, the statistical framework behind MIA is no longer applicable since other approaches are statistically unsound. Duan et al. (2024b) demonstrate empirically that a range of MIA is not reliable against LLMs. According to their findings, training for a single epoch- as is becoming standard practice for LLMs- and increasing amounts of training data may negatively affect MIA accuracy. They also notice that MIA become more difficult as the difference between data included in the training set and data not included becomes more subtle. However, as training corpora become larger – their statistics are based on the Pile (Gao et al., 2020) – this similarity increases: for example, non-members Github samples share on average 76.9 % of their 7-grams with the Pile. Hence, the combined effect of training techniques that diverge from the one defined for other models and the huge training sets may challenge MIA's effectiveness against LLMs.

### 3.3   Model Inversion and Stealing

Model inversion attacks generally exploit the output of the model and use this information to reconstruct the input data that generated the observed outputs. In the initial formulation (Fredrikson et al., 2014; 2015), an adversary is given access to the trained model and to a partially masked input, for which only non-sensitive features are disclosed. The attacker aims to infer the value of the sensitive attributes that serve as input to the model: since deep learning models create dense, vectorial representations for textual data, model inversion attacks against deep learning architectures aim to decode human-readable features from them. While the initial formulation is implemented for linear regression models (Fredrikson et al., 2014) and a neural network trained for image processing (Fredrikson et al., 2015), this type of attack has been successfully used to extract sensitive input information from different models, including LSTMs (Welleck et al., 2018). Those kinds of attacks are challenging against LMs, especially since, in principle, the optimization process to reconstruct the initial input of a model has to deal with the discrete nature of text (Parikh et al., 2022).

Model inversion attacks are often performed against distributed models; in fact, some protocols like federated learning ensure that private data are not shared across multiple servers (see §2.3 for further details). However, with some access to model outputs, it is possible for an attacker to reconstruct the input data that causes the observed output.

> **Model Inversion Attacks and Model Stealing**
>
> This family of attacks threatens models, especially when they are distributed across multiple servers: in fact, while data holders do not directly share private data – which are thus maintained locally– they may be willing to share seemingly opaque information related to a machine learning model, such as model outputs, embeddings or gradients. However, Model Inversion attacks are designed to reconstruct private data from these apparently opaque sources. In some cases, also centralized private models parameters can be directly reconstructed: this is the objective of Model Stealing attacks.

In some cases, from those opaque sources, it's also possible to reconstruct a model's parameters from its output: in this case, the attack technique is often defined as Model Stealing (§3.3.3). These attacks aim to recover model weights from an API-gated model, making it less opaque and, hence, less robust against other types of privacy attacks. While the stealing of entire LLMs has not yet been described, some works efficiently reconstruct a subset of the model parameters in black-box attacks.

### 3.3.1 Model Output Inversion and Model Stealing

The outputs of a model, like the representation of textual data that a Transformer-based LM produces in the last layer, can be totally or partially inverted to reconstruct the text that is originally the input of the model. In particular, these representations – often called embeddings – are only apparently opaque and can be inverted to reconstruct the entire sentence or the words that it contains. Pan et al. (2020) design the first model inversion attack against Transformer-based Language Models. They observe that directly framing the inversion problem as an optimization problem is challenging due to the discrete nature of sentences. They train an attack classifier that is designed to extract fixed patterns and keywords – like, for example, location or medical information – from sentence representations of models. Similarly, Song & Raghunathan (2020) predict unordered sets of words from sentence embeddings: their attack model is a neural network trained with the multiset prediction loss. While the experiments in Pan et al. (2020) and Song & Raghunathan (2020) are performed on relatively smaller Transformer-based network compared to more recent models, their methods can generalize to larger networks with some additional difficulty due to the increasingly larger embedding dimension in LLMs.

In some cases, the entire input text is also reconstructed. Li et al. (2023d) implement the inversion as a generative language modelling task with a decoder-only model, conditioned on the embedding of the sentence to invert as the first token representation. They show that the model successfully inverts meaningful tokens containing a large fraction of the named entity. The threatened model in their experiment is a BERT model, but the general framework can be extended given model representations for sentences. Morris et al. (2023) propose an iterative method to reconstruct the input text of Transformer-based models that produce embeddings for documents: their attack is based on the idea of iteratively generating different hypotheses that may justify the observed embeddings. They learn the distribution of texts in a dataset given their embeddings: their approach starts by assuming an initial hypothesis and then iteratively improves it by adjusting the hypothesis to progressively align its embedding to the target one. This approach has also been extended to a multilingual setting, adding a translation step before iterating again over the generated hypothesis Chen et al. (2024b). Only observing the logits of LLMs can also be sufficient to reconstruct the original input: Morris et al. (2024) train an inversion model on a collection of instruction datasets to reconstruct the prompt of an LLM – in their experiments the seven billion version of Llama-2 and Llama-2 Touvron et al. (2023) – given the probability distribution of the token following the prompt. In a completely black box scenario – in which only the model's output is observed – an attacker may be able to reconstruct the original prompt training an inverter – another LM – on previously observed outputs on known input prompts Zhang et al. (2024a).

This kind of attack has also been applied to Language Models fine-tuned on classification tasks to reconstruct private training data. In this case, the output of the model is a probability distribution over each of the possible classes. This information can be effectively exploited: in particular, an attacker may exploit the greater confidence that a model has in classifying sentences that are already part of the training set. Elmahdy et al. (2022) inject canaries in the training data (sequences that are intended to be later extracted and out of

distribution with respect to the other data) and then reconstruct a partially masked sentence to search for tokens that maximize the probability of the target label. They perform the reconstruction by enumerating tokens in the vocabulary, weighted by the number of occurrences that they have in the training data. Zhang et al. (2022c) propose instead to build a dataset that resembles the unknown training data and initially train a text generator on these collected data, similarly to what is done with shadow model MIA (§3.2). Then, they further train the text generator via word embedding perturbation with the objective of minimizing the classification loss on the target model. The idea is that the target model's prediction on training sentences will exhibit a lower loss, and this information can be used to perturb the text generator model and obtain data closer to the private one. Although the use of generative models as classifiers is not widespread at the time of writing, the results of this works could be generalized to the setting of generative LLMs with some additional difficulties in defining the target class of fine-tuned generative models.

### 3.3.2 Gradient Inversion

In Federated Learning (Ray et al., 2021), multiple nodes train a model cooperatively without directly sharing data samples: the system is designed to make training feasible while exchanging model updates rather than data across nodes (see §2.3 for further details). Gradient Inversion attacks are directed against federated learning environments that perform training by sharing gradients. Those attacks can be used to reconstruct training data from seemingly opaque sources.

Zhu et al. (2019) demonstrated that if the gradients are openly accessible, it is possible to reconstruct the training data. The attacker aims to reconstruct the training data by minimizing the distance between some generated gradients and the ground truth one. In particular, the attacker initially generates gradients from random inputs and labels. Then, having access to the ground truth gradients solves the optimization problem of minimizing the $L_2$ distance between the generated gradients and the ground truth one, parameterized by real inputs and labels. In their experiments, Zhu et al. (2019) applied this attack against a BERT model. This idea was further explored by inverting gradients for image classification tasks (Zhu & Blaschko, 2021; Geiping et al., 2020; Yin et al., 2021).

In reconstructing textual data, Deng et al. (2021) further attempted to perform a gradient attack against Transformer-based language models, introducing also a $L_1$ term in the reconstruction loss. Experiments on a Transformer model, BERT model and TinyBERT demonstrate the effectiveness of this gradient attack, recovering up to 50% of the original tokens on average. In LAMP (Balunovic et al., 2022), more effective results are achieved by simultaneously training the attack model to minimize the difference between the reconstruction gradients and choosing at each iteration only sequences that have low perplexity according to an external language model (like GPT-2). Reconstructing the correct word order is, in fact, one of the main challenges: Gupta et al. (2022) proposes to recover from the gradients a bag of words for the sentence to extract and then perform a beam search to effectively reconstruct the sentence.

Should be noted, however, that the moment of writing the application of those attacks to larger models is mostly limited by the increasing usage of centralized pretrained models.

### 3.3.3 Model Stealing

As exposing only black-box access to a LLM is a standard practice for popular commercial models, an adversary might want to reconstruct the gated model with the aim of imitating its behaviour. A family of attacks called model stealing attacks have been designed with this goal. The adversary, in this case, can have no information regarding the original training data. The attack usually requires observing the model outputs on inputs that the attacker fed to the gated model to reconstruct its internal parameters. Tramèr et al. (2016) introduce this family of attacks and demonstrate that an attack could lead to an accurate reproduction of logistic regression, neural networks, and decision trees. They also successfully demonstrate on those systems that model inversion as defined by Fredrikson et al. (2015) could be applied to the stolen model to reconstruct private input, with direct privacy implications.

Krishna et al. (2020) demonstrate the feasibility of those attacks against Transformer-based models. They use random words as input for gated fine-tuned BERT models and use their outputs for fine-tuning an equivalent model. They also demonstrate that the quality of the stolen model increases if the attacker has

some background knowledge regarding the domain of the original fine-tuning dataset. Moreover, Zanella-Beguelin et al. (2021) notice that an attack that exploits the characteristics of the Transformer's last layer can effectively reconstruct a fine-tuned BERT model with frozen pre-trained weights and learnable embedding projection matrix. In particular, they reconstruct the embedding projection matrix by feeding the gated model random inputs and collecting the corresponding output. They reconstruct the embedding projection matrix optimizing its weight on the observed embedding and outputs. Their technique, in combination with other techniques like the one proposed by Krishna et al. (2020), can lead to a refined model that resembles the target one.

Recently, a similar idea has been applied to threat LLMs: Carlini et al. (2024) proposed an attack to recover the embedding projection matrix at the last layer of a transformer language model. They attack a model in a black-box fashion, collecting the model's output logits on random prompts. They notice that since the embedding projection matrix is low rank – since it projects the hidden dimension to a logit vector that has higher dimensionality – it's possible to apply the SVD algorithm to the collected logits to obtain a linear transformation of the original embedding projection matrix. Moreover, they demonstrate that the logits over the entire vocabulary are not necessary, and that the probabilities over a predetermined number of words are sufficient for a successful attack. A similar approach has also been proposed by Finlayson et al. (2024), and they describe the utility of the attack in discovering whether a gated model has undergone some update.

### 3.4  Privacy Threats at Inference Time.

In the LLM context, the leakage of the original training data is not the only possible threat. The inclusion of additional, potentially private information at inference time presents a number of challenges. Here, we summarize three of the major challenges derived from such systems. These threats have pushed a great deal of research to focus on how to perform calculations securely on sensitive user input, either by obscuring them through differential privacy (§5.2) or by keeping them encrypted during the entire calculation process (§5.5).

**Computation over Private Input.**  Given the challenges of training such models from scratch, the option of using LLMs in a Machine Learning as a Service modality is becoming increasingly popular. In this setup, an organization hosts the model and users simply submit their data for processing. This approach raises significant privacy concerns for user data: vulnerabilities in the systems storing this data could lead to breaches (OpenAI, 2023), and model providers must offer strong assurances to comply with privacy regulations, such as GDPR. Similar concerns even led to a temporary ban on ChatGPT in Italy (Lomas, 2023).

This issue is particularly concerning for businesses dealing with proprietary data or commercially valuable information (Mitchell, 2023; Coles, 2023), and companies have decided, in some cases, to warn their employees about the responsible use of LLMs (Kim, 2023).

**Prompting Private Data.**  At inference time, the versatility of LLM is enhanced by the use of additional data, which are often provided in the form of demonstrations in the context of the task the user wants to perform (Brown et al., 2020). While being a useful technique, some sensitive information could be included in these examples in context as well, and an increasing body of research suggests that some risks can be identified during inference. In particular, since in-context learning is more cost-effective than fine-tuning, some systems may deploy LLMs that handle user requests by utilizing pre-loaded in-context demonstrations, which may include private or sensitive information. This efficacy, however, comes with an increased risk in terms of privacy leaks, which can also outweigh the risks against fine-tuned models (Duan et al., 2023). Wu et al. (2024b) propose a threat model in which a user of a system submits some data containing private information, such as medical records, and another user of the same system could potentially access that sensitive data. They focus on addressing this threat through the application of Differential Privacy during inference (see §5.2 for further details). Duan et al. (2024a) demonstrate the efficacy in a membership inference attack (MIA) against prompts containing sensitive, in-context learning demonstrations: they suppose that a model is prompted for a classification task and that the adversary may want to understand whether a certain example has been used or not as in context demonstration. They argue that an adversary can test whether an example has been used in context since the probability for the corresponding class will be much higher for

members (in context examples) than for non-members. Similarly, in this scenario, a malicious user –instead of performing the intended task on the system– can potentially extract the in-context demonstrations via jailbreaking attacks (Tang et al., 2024). Chu et al. (2024) demonstrate that entire conversations can be reconstructed via jailbreaking when they are used as contextual information for prompting an LLM.

**Contextual Privacy.** As LMs' capabilities increase and their role as assistants becomes more prominent, it is important to understand whether they are able to distinguish if a piece of certain sensitive information is safe to be shared in a given context. For example, there is no privacy leakage when a doctor shares with his patients information about their own health. However, the disclosure of such information to an unauthorized insurance company violates patients' privacy. Those considerations are recently emerging as language models can be reliably used as assistants by groups of users that want to share some information while preserving others. Brown et al. (2022b) discuss the discrepancy between data sanitization and differential privacy that assumes a structure of private data and cannot solve the necessity to share private information only with the correct group of people. They also argue that some data protection techniques – like data sanitization and differential privacy – make assumptions on the nature of the data that should be protected: implicitly or explicitly, they state that private information is structured information that should not be shared. However, every bit of language can potentially contain secrets that are not meant to be shared indiscriminately. Mireshghallah et al. (2024) formalize the necessity to share secrets in the appropriate context stemming from the contextual integrity theory (Nissenbaum, 2004). In particular, the information to be protected is not only in the training data but also during inference. They introduce a new benchmark to assess the ability of such models to disclose private information only to users who should be allowed to access them. They measure that multiple LLMs include information that should not be disclosed when asked to generate summaries of meeting conversations. Priyanshu et al. (2023) similarly explore the tendency of ChatGPT to disclose personally identifiable information when asked to produce summaries of hiring decisions and medical notes, a behaviour that they notice can be mitigated by explicitly asking the model to comply with privacy regulations and feeding step-by-step prompts that suggest which information should be deleted. Wang et al. (2024a) also discusses the necessity of measuring the privacy leakages during inference. They focus on PII like emails and phone numbers: they propose to initially feed the model with prompts that contain PII with the indication that those should not be shared; then, they prompt the model asking for the same information. They notice that GPT-3.5 is prone to violating privacy restrictions, while GPT-4 is more robust. Both models, however, tend to reveal private information when prompted with some demonstrations containing leakages in the few-shot scenario.

# 4 Solutions for Preserving Privacy in LLMs: Data

After identifying the key vulnerabilities and potential attacks on LLMs as outlined in §3, in this section, we introduce the reader to the current possible solutions to the problem of preserving privacy in the context of LLMs and text generative models. We have to say that the field is fragmented as this is a recent concern, and the field is evolving fast. At the moment of writing this survey, there is no single gold standard solution to the problem of preserving privacy in LLMs and text generative models. The main solutions available can be broadly categorized into two categories: **i) Preserving Privacy in Data** §4 and **ii) Preserving Privacy in Model** §5. In the former section, we collected all the methods that try to solve the problem at its source, i.e., they act directly on the privacy-sensitive information in the dataset before the training happens, while in the latter, we selected all the methods that address the problem from the model perspective, i.e., try to remove from the model problematic information after the training. The two approaches fit different use cases, so it is important to be aware of both.

In this section, we propose several solutions that allow for preserving privacy from the *data perspective*, i.e., we preserve privacy directly in the training data used in machine learning models. This includes any approach that involves manipulating, carefully choosing, or generating training data in order to avoid the memorization and subsequent leakage of private information inside a large language model or any model that learns from data. This basically puts the focus on preserving privacy in the *data* used to train the model. Given that a machine learning model is moulded by its training dataset, training it on a privacy-preserving dataset ensures that these methods offer equivalent privacy assurances to the dataset itself. The methods presented here are thus not limited to LLMs as they can be used to anonymize textual data more generally. That is, it is possible to use them to strip datasets of sensitive information before using these datasets to train LLMs. Making the training data safe means making the trained model safe.

In the context of LLMs, there are very few technical reports that thoroughly describe the precautions necessary to preserve privacy via data anonymization, and even fewer that release the preprocessing software used for this anonymization. Among the notable examples, the LLaMA3 technical report (Dubey et al., 2024) mentions "filters designed to remove data from websites [to eliminate] unsafe content or high volumes of PII" but does not detail the exact procedure while noting the use of "Microsoft Presidio Analyzer" in the preprocessing of speech data (See our overview of the tool in §6). Similarly, Soldaini et al. (2024), in the context of training OLMo (Groeneveld et al., 2024), also considered using Microsoft Presidio but ultimately opted for custom regular expressions, given the scale of the pre-training dataset.

> **Preserving Privacy in Data**
>
> **Key Idea:** Preserve privacy directly on the dataset used to train the model.

Overall, the available open reports suggest that for removing PII from the *pre-training dataset*, the preferred solution at scale tends to involve custom regular expressions (Penedo et al., 2024). The broader subject of preserving privacy in *pre-training data*, however, remains largely unexplored beyond these rudimentary techniques (Subramani et al., 2023). The scope of this section is thus two-fold: i) propose to the reader possible new avenues to preserve privacy in the *pretraining* dataset by surveying existing approaches proposed in the literature that have not been applied yet in the context of the *pretraining phase* of LLMs; ii) propose to the reader anonymization techniques that can be used to anonymize data for the *instruction-tuning / fine-tuning* phase where the LLMs are possibly fine-tuned on sensitive in-domain data for the downstream task (e.g., healthcare). Since most of these techniques were not tested with training datasets for LLMs, it is difficult to discuss about the scaling of these methods for big datasets, but seeing examples like the previously cited OLMo (Groeneveld et al., 2024), it is quite clear that complex methods at a big scale tend to not be chosen. However these techniques can be applied to data regardless of the subsequent usage, which means that they could (and will) be used to clean data for inference, before feeding it to a model.

The section is structured in the following way: we first show classical anonymization techniques §4.1, then we focus on anonymization techniques built with Differential Privacy §4.1.2. Putting this last section in relation with Figure 2.1, the point of application of DP would be (1), the input, because all these techniques aim to apply Differential Privacy on the data before feeding it to the LLM (for either training or inference).

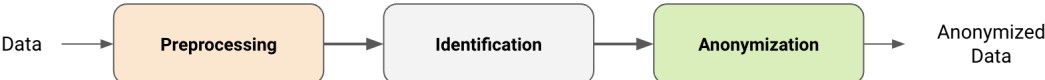

Figure 4: **Classical Anonymization Pipeline.** The figure illustrates the standard process for anonymizing data in natural language processing tasks. The pipeline consists of three sequential blocks: Preprocessing, Identification, and Anonymization. The Preprocessing block prepares the raw data for further analysis. The Identification block involves the use of a procedure to detect potential Personally Identifiable Information (PII) within the data (e.g., Named Entity Recognition). Finally, the anonymization block replaces identified PII with anonymous substitutes, ensuring data privacy preservation. Arrows indicate the flow of data through each stage of the process.

## 4.1 Anonymization

Data anonymization is not a new concept, with seminal work dating back to the 1990s: a pivotal development was the introduction of k-anonymity in 1997 (Sweeney, 1997), which means ensuring that data cannot be re-identified to fewer than k individuals. This foundational concept has spurred extensive research, leading to a plethora of anonymization techniques. Given the breadth and depth of the field, this section is organized into two main parts: the first, titled Classical Anonymization 4.1.1, explores traditional methods such as rule-based approaches and includes a discussion of how recent advancements in language models have been leveraged for anonymization purposes. These traditional methods have been the cornerstone of anonymization for decades. In more recent developments, the rapid adoption of Differential Privacy has marked a significant evolution in the field, offering robust privacy guarantees that, depending on the application context, can surpass those of classical methods. The second part of this section, Anonymization with Differential Privacy (§4.1.2), discusses these techniques. It also introduces the concept of Metric Differential Privacy, highlighting its unique applications and benefits within the field of LLMs and beyond.

### 4.1.1 Classical Anonymization

As already mentioned, anonymity definitions can be traced back to 1997 with k-anonymity. Anonymization aims to make the re-identification of an individual impossible, that is, the process of reconstructing the identity of an individual given a record that contains their information. Different attributes could be anonymized. While effectively masking PII – like names, emails, and social security numbers – is necessary, some features that are not exactly identifiers can be used in combination to identify an individual: those features are called quasi-identifiers. Quasi-identifiers are apparently non-personal attributes – like gender, date of birth, and zip codes – that can be used in combination with external resources to identify a person from its anonymized records. The definition of k-anonymity (Sweeney, 1997) ensures that a quasi-identifier cannot be used for re-identification. Essentially, a dataset is said to have k-anonymity if any given record in the dataset is indistinguishable from at least k - 1 other records with respect to certain identifying attributes called quasi-identifiers. Upon this concept, several implementation techniques were proposed by Samarati & Sweeney (1998), like generalization (modifying attribute values to less specific but accurate representations), suppression (i.e., entirely removing data that are outliers or too identifiable), minimal Generalization (ensuring data are not generalized more than necessary to meet k-anonymity). Of course, this method is highly susceptible to the choice of k (even inapplicable sometimes) and gives coarse data, making it impossible to carry out fine-grained analysis. It is also important to mention that, due to the reliance on quasi-identifiers, this method is much less effective with unstructured text. Since anonymization has such a long history – with the first paper theorizing k-anonymity dating back to 1997 (Sweeney, 1997) – much research was devoted to this field, resulting in the development of several techniques. Larbi et al. (2022) give an overview of many anonymization techniques, with their different pros and cons based on the use case. In this section, we will show the most relevant ones that are also available in a synthetic tabular format in Table 2.

The standard process is outlined in Figure 4. After any necessary preprocessing phase, the first challenge to solve is to identify the entities that may cause a violation of privacy. This can be done via a Named Entity

Recognition (NER) tool. To maintain the coherence of the text, a Co-reference Resolution (CRR) system can be used to resolve references between entities should be identified if the anonymization framework aims to ensure consistent anonymization across the document. Finally, private information should be replaced with anonymous information. The anonymization methods can vary. Suppression is used to completely remove the information. Tagging allows the replacement of sensitive information with artificial labels –like the one obtained by the NER system– that still retain information about the class of the data and an identifier. The sensitive information can also be replaced with other textual information: random substitution and generalization can both be used. Random substitution uses another random entity of the same class, while generalization replaces an entity with a more general term. It should be noted that, while this pipeline is intended for general data anonymization, regardless of which model is going to be trained on it (be that an LLM or not), it is specific to textual data, as it relies on some specific features of text.

**Anonymization before DNNs**   Mamede et al. (2016) present a modular anonymization framework for text documents in Portuguese that closely resembles the general workflow for this task. The system consists of four modules: pre-processing, Named Entity Recognition (NER), Co-reference Resolution (CRR), and Anonymization. They notice that the effectiveness of automatic text anonymization methods relies heavily on the performance of Named Entity Recognition (NER) and Co-reference Resolution (CRR) modules and while CRR shows good performance, the NER can be very lacking depending on the dataset. They also implement and test a number of different anonymization strategies: all the anonymization methods presented can be effective in some cases, but they all exhibit some drawbacks. Suppression, while dependent on NER accuracy, often compromises text comprehension due to loss of information, challenging readers to grasp entity references. Tagging helps maintain references between anonymized entities but detracts from natural text flow, necessitating improved NER and CRR accuracy. Random substitution maintains natural language output but can lead to semantic drifts due to arbitrary replacements, suggesting a need for a curated list of vague yet contextually appropriate entities. Generalization offers a balanced approach by replacing specific entities with broader categories and improving readability and relevance, especially for location entities, though it struggles with accurately linking entities with multiple knowledge base entries.

Another prominent anonymization technique is perturbation, as proposed by Zuo et al. (2021). It involves subtly altering the original data to preserve statistical significance, aimed at reducing the vulnerability of the dataset to linkage and enhancing privacy protection. Techniques like microaggregation and data swapping are commonly used for the implementation of perturbation. Microaggregation involves grouping data and using representative values, while data swapping involves interchanging values between records to maintain the overall statistical properties of the dataset. While perturbation effectively anonymizes raw data and retains its statistical utility, it may compromise the accuracy of anonymized data.

While there is no doubt that these methods could be used to clean a dataset for LLM training or inference, it is not very clear how they would scale for an LLM-sized training dataset, as they are developed for much smaller datasets.

**Anonymization with DNNs**   More recent anonymization methods also leverage Deep Neural Networks. Zhou et al. (2022) use BERT to impute previously masked sensitive information. This framework is called DataSifterText, and it works with a whitelist that holds sensitive terms that need to be masked and a blacklist for meaningful terms that should be retained. After a proportion parameter is chosen, sensitive tokens are masked according to that proportion, and then BERT is used to replace the masked tokens. These two steps in the procedure respectively cover the identification and anonymization step shown in 4). Results show that even when the obfuscation level is high, the framework retains significant data utility, demonstrating the method's effectiveness in maintaining informative content.

In some cases, the author of a text is meant to be identified: re-identification of an author can be done by spotting some features that characterize an author's writing. Romanov et al. (2019) analyze this scenario for the Russian language. They extract features like lemmas, parts of speech and morphological features. Then, a fast correlation filter – a multivariate feature selection method where the class relevance and the dependency between each feature pair are taken into account – is used to select the most informative features. Those informative features are the more likely to be identifying features. To preserve the privacy of the author, those

features are smoothed by making targeted adjustments. For example, words or punctuation marks that are characteristic of the author's style are replaced or altered based on average statistics derived from a broader corpus. The final step is to further process the smoothed text with a deep learning model: in this work, a Transformer-based model is chosen. This approach reduces the probability of authorship identification to random guessing.

Mosallanezhad et al. (2019) use an attention-based task-aware text representation learner for extracting latent embedding representations with a reinforcement-learning-based privacy and utility preserver for manipulating these embeddings. While it effectively balances privacy and data utility and is adaptable and efficient, it has notable drawbacks, including complexity, potential data-specific limitations, and dependency on precise parameter tuning. Complexity in particular is relevant to this context, as it is likely to make this method less suitable for large-scale datasets.

More recently, some works proposed the use of LLMs for anonymization. Vats et al. (2024) propose to hide sensitive tokens via a masking process and impute them by using a Large Language model (LLM). In relation to Figure 4, the masking acts as the identification part, as it finds and masks sensitive information, while the imputation represents the proper anonymization, where non-sensitive terms replace the (previously masked) sensitive ones. To carry out the masking process, several techniques are proposed: a handcrafted list of safe words (the ones that do not carry sensitive information) called *allowlist* created by linguistic experts, this is similar to the blacklist of Zhou et al. (2022) and identifies the terms that do not need to be masked; another method to identify safe words is vocabThres, where the N more frequent words in a vocabulary are considered safe; the last masking criteria is to find the unsafe words and involves using a NER module that identify potentially private tokens to mask them (similar to Mamede et al. (2016)). After choosing a masking method, the chosen tokens are masked and then imputed by an LLM. Also for the imputation, three separate methods are proposed: Top-1, which uses the one token with the highest probability from the LLM's output; Top-K, which randomly selects one of the first K tokens with the highest output probabilities; Fine-Tuning, which selects the records that do not contain sensitive information and uses them to fine-tune the LLM, then just takes the highest probability output (so after fine-tuning it circles back to the Top-1 approach). They compare models trained on the anonymized datasets against models trained on the masked dataset, showing a significant improvement in both tasks. Intuitively, this method is quite sophisticated and complex, which means that for an LLM-sized training dataset it is going to make up for a quite heavy computation, but it could be suitable for a smaller dataset that might be used for fine-tuning or inference.

As the usage of closed models increases, privacy concerns also emerge when sending prompts containing private information. Chen et al. (2023b) focus on sanitizing prompts before sending them out to some unsafe third party like a black-box model like ChatGPT, thus enforcing prompt privacy. Moreover, they argue that standard anonymization techniques – like masking – may be useful to ensure the privacy of prompts, but the model output could not be effectively de-anonymized to improve its utility. They propose a new framework called HaS (Hide and Seek) with two local small LMs, one trained for anonymization and one trained for deanonymization. The first one preprocesses the prompt before sending it to the black-box and the second one postprocesses the model output. This method is specifically designed to use LLMs in inference without worrying about personal data in the prompts. The hide model is trained with original sentences and extracted privacy entities and is prompted to substitute them randomly, while the seek model is trained with the same data and the addition of the remote LLM answer. The framework is evaluated using both adversarial models (black box and white box) to measure privacy protection effectiveness and usability experiments to assess the impact on LLM task performance, including translation and classification. Adversarial models are unable to recover original information, indicating stronger privacy protection. Usability is evaluated by metrics like BLEU for translation and precision, recall, and F1 scores for classification to see how well the framework maintains the utility of LLM outputs post-anonymization. The results demonstrate that HaS effectively balances privacy and utility, with label-based anonymization showing better privacy protection and generative methods offering slightly better utility retention.

### 4.1.2 Anonymization with Differential Privacy

In this section, we propose works that use Differential Privacy for data anonymization, i.e., before actually using the data in training. These methods are intended to anonymize datasets, regardless of the intended

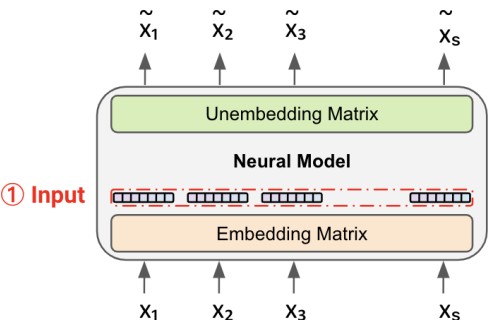

Figure 5: **Anonymization with Differential Privacy.** This figure illustrates the procedure to anonymize data via differential privacy. A neural model is used to embed data, data embeddings are then perturbed with noise using DP. The anonymized output is then returned by decoding the perturbed embeddings.

use of the datasets. This means that the anonymized datasets might be used to train LLMs, but also for other models and tasks. We refer the reader to §2.2 for an introduction to Differential Privacy, and we will propose later in §5 methods that use Differential Privacy directly on the model. The methods proposed in this section apply Differential Privacy to perform text rewriting, where the original text is modified by perturbing its representations to ensure privacy. The key advantage of this approach is that it adheres to the formal guarantees of Differential Privacy. All the works in this section are also available in tabular format in Table 3.

Referring to Figure 5, we identify the works in this section as examples of Differential Privacy applied to the input (1) of a neural model. Differential Privacy can also be applied in a separate, typically smaller, auxiliary preprocessing model to anonymize data before it is fed into an LLM.

**Metric Differential Privacy** Feyisetan et al. (2019b) propose a DP mechanism to transform text data that constructs a hierarchical representation to identify private phrases in the input sentence and then randomly replace them with neighbouring words in a word embedding space. They build upon Metric Differential Privacy that, as explained by Chatzikokolakis et al. (2013), extend Differential Privacy principles to location-based privacy. This approach, originally developed for spatial data, is adapted for textual data by utilizing word embeddings as coordinates, transitioning its use from spatial to textual domains. For obfuscation, target words are selected by favoring locations near the original one with a higher probability over distant ones. For a graphic representation of this process, refer to 5. Building further on the theme of differential privacy in metric spaces, Feyisetan et al. (2019a) introduce a noise distribution tailored to the d$\chi$-privacy criterion, which calibrates $\epsilon$ values based on the geometric properties of the word embedding space to maintain plausible deniability of original words. This methodology was tested across various tasks including binary sentiment classification on the perturbed IMDB dataset, multi-class classification on Enron, and question answering on InsuranceQA, revealing less than 2% utility loss in binary classification tasks, even at low $\epsilon$ values, indicating a favourable balance of privacy and utility compared to other methods like Versatile (Arampatzis et al., 2015) and Incognito (Masood et al., 2018). While the tests for this method do not include LLMs and text generation in general, nothing would prevent training or fine-tuning an LLM with a dataset cleaned with it. This would definitely cause a preprocessing overhead, but it could be balanced by the missing training overhead from removing DP in the training phase, so it is not clear whether this would be a favorable trade-off and it would probably depend on both dataset size and model size.

The Differential Privacy mechanism, in textual examples, should not disrupt the syntactic structure of a sentence. For this reason, Krishna et al. (2021) developed ADePT, an autoencoder-based differential privacy algorithm aimed at anonymizing text while preserving its utility. This method involves clipping and noise introduction—either Laplacian or Gaussian—between encoding and decoding to form a differentially private latent representation. They hypothesize that data regeneration via an auto-regressive decoder maintains the syntactic structure of the sentence. Since they focus on the anonymization for Intention Classification (IC),

an intention label is included before encoding to facilitate training an annotation-aware autoencoder. Despite ADePT's design to enhance privacy against Membership Inference Attacks and its superiority over traditional privacy-sensitive word replacement methods like Feyisetan et al. (2019b), Habernal (2021) point out that ADePT actually fails to achieve true differential privacy and exhibits higher sensitivity than reported, casting doubt on its effectiveness. In a similar fashion, Weggenmann et al. (2022) introduce a DP-VAE (Differentially Private Variational AutoEncoder) that also perturbs the latent space to secure privacy, following the model depicted in 5. This model contracts the mean vector of each latent representation towards the latent space's origin before adding isotropic Gaussian noise. Applied to anonymize IMDb and Yelp reviews, this technique successfully reduces authorship attribution accuracy significantly while only slightly impacting sentiment classification accuracy. Further testing with a GPT2-based model confirms that the anonymized texts remain coherent and understandable despite a slight increase in perplexity.

Also transformer-based models can be used in this setting. Extending the research on Differential Privacy for text privatization in this direction, Igamberdiev & Habernal (2023) introduced DP-BART, a privatization model using the BART architecture instead of an LSTM, employing a similar strategy of clipping and noise addition in the latent encoder output, as seen in 5. DP-BART tackles the challenges of high sensitivity and computational demands associated with transformers through a novel clipping technique (by value) and an iterative pruning and training method that simplifies the encoder's output vectors. Tested against ADePT on datasets like ATIS, Snips, IMDb, Drugs.com, and Amazon, DP-BART demonstrated superior performance in both privacy and utility. Another transformer-based approach is proposed by Du et al. (2023a), where they use BERT and propose the Purkayastha Mechanism to take advantage of the angular distance as a metric. They also propose to sanitize labels when too sensitive, using Randomized Response if the label space is small enough. Otherwise, they prune the label space, leveraging prior knowledge and then use RR. This approach is tested on SST-2, IMDb, and QNLI datasets, representing tasks for sentiment analysis and question-answering based on movie reviews and Wikipedia content, respectively. Performances are close to the non-private version with an increased robustness to MIAs.

**Leveraging the Exponential Mechanism**    Yue et al. (2021) argue that – due to the curse of dimensionality – directly injecting noise in large high-dimensional word embedding may fail to strike a nice privacy-utility balance since it becomes exponentially less likely to find a noisy embedding close to a real one on every dimension. Yue et al. (2021) developed two strategies, SANTEXT and SANTEXT+, to replace original text tokens with "sanitized" tokens via the exponential mechanism. Both methods utilize embeddings to map words to compute their semantic distance. Then, once a word is identified to be replaced, the target word is selected with a probability that is inversely proportional to the distance with respect to the original one. SANTEXT indiscriminately replaces any word, often resulting in excessive utility loss and protection, while SANTEXT+ specifically targets and replaces the top w low-frequency tokens, identified as most likely to be private. These approaches enforce Metric Local Differential Privacy (MLDP), maintaining utility by favouring tokens closer to the original, though requiring a high epsilon value, like 16, to achieve notable improvements in defence rate. Similarly, Chen et al. (2023a) introduced CUSTEXT, another MLDP approach where every token is replaced using a mapping function that determines the output set for an input token and a sampling function based on the exponential mechanism for selecting replacements. The main difference here is that every input embedding has a custom set of output embeddings, which is a subset of the full output space used in SANTEXT, and thus, it reduces the chances that the output is semantically irrelevant to the input. They claim the best privacy/accuracy tradeoff for these systems so far, achieving good results with relatively low epsilon values (up to 3), although there is still a notable performance drop compared to using plain text.

SANTEXT and CUSTEXT, while being effective, show some limitations. Tong et al. (2024) critique the small adjacency list used in SANTEXT and CUSTEXT, which could expose the methods to embedding inversion attacks and inefficacy in open-ended text generation, where bias matters more. Tong proposes RANTEXT, which dynamically generates a "random adjacency list" for each token, then adds noise to the token's embedding and selects nearby tokens based on the noisy embedding, leveraging the Laplace mechanism to ensure Differential Privacy. This method, developed within a framework for using black-box LLMs in private inference, includes a denoising module to reconstruct private outputs. RANTEXT, tested on CNN/Daily Mail (Hermann et al., 2015), Wikitext-103-v1 (Merity et al., 2017) and the ArXiv dataset

(Clement et al., 2019), generally outperforms other methods in both utility and privacy, with exceptions in some areas where CUSTEXT+ performs slightly better. This comprehensive approach enhances the robustness of textual differential privacy, especially against inference and embedding inversion attacks.

As already discussed, none of these methods is specific to LLMs but they all could be used to preprocess the training dataset for an LLM, although with some computational overhead given by the usual (huge) size of LLMs' training datasets.

**Beyond the exponential mechanism**  Carvalho et al. (2021) critique the exponential mechanism's fit for Metric Differential Privacy, noting it fails to consider the density of the space from which outcomes are selected. They propose the Truncated Exponential Mechanism (TEM), which adjusts noise addition based on the density of the embedding space around an input word, evaluated for utility with a sentiment classification model and privacy with MIAs, showing substantial improvements over the baseline (Feyisetan et al., 2019b). Adding to these developments, Xu et al. (2020) propose a Regularized Mahalanobis Metric for text perturbation to address the shortcomings of traditional noise injection methods that suffer from low utility due to the uniform application of noise in continuous embedding spaces. By using a regularized version of the Mahalanobis metric to add elliptical noise, accounting for the covariance structure in the embedding space, this method aims to ensure sufficient likelihood of replacement for words in sparse regions without sacrificing overall utility. Tested on Twitter and SMSSpam datasets, this method shows a lower number of unchanged words and a greater diversity of word substitutions compared to the Laplace mechanism, suggesting higher privacy while maintaining comparable text classification performances. These advancements highlight the ongoing challenges and innovations in achieving effective textual differential privacy without compromising the utility of the data.

**Known Issues**  Expanding on the theme of embedding-based perturbation methods, Xu et al. (2021) acknowledges a common limitation in such approaches, where even minor noise could still leave the original embedding as the nearest neighbour. To counter this, they introduce a method inspired by Vickrey auctions to balance the selection between the first and second closest neighbours using a tuning parameter. This parameter is optimized through a constrained optimization problem aiming to maximize utility while ensuring privacy guarantees. Conducted experiments on Product Reviews, IMDb Movie Reviews, and Twitter datasets reveal that this Vickrey mechanism can enhance utility by up to 50% on real text classification datasets compared to existing methods without compromising privacy, highlighting an innovative stride in balancing the privacy-utility tradeoff in textual differential privacy strategies. Further examining the methods in the field, Arnold et al. (2023) critiques the focus on semantic-based embeddings in existing Metric Differential Privacy techniques, noting their failure to consider syntax specifically. Analyzing these techniques' ability to preserve grammatical properties, they found that privatization mechanisms often privilege nouns even when replacing other grammatical categories. To address this, they incorporated grammatical categories into the privatization process, using methodologies from Feyisetan et al. (2019a) or Carvalho et al. (2021). Results indicate that this approach can enhance performance on downstream tasks, although it also shows a slight increase in self-substitution, suggesting a small decrease in privacy compared to previous approaches. This development underscores the complex interplay between maintaining linguistic integrity and ensuring robust privacy protection in textual data. Adding to the critique, Mattern et al. (2022b) identify significant limitations of Metric Differential Privacy methods, such as the increased noise necessary for long texts or their inability to rephrase effectively. Proposing an alternative, they suggest a paraphrasing model obtained by fine-tuning GPT2, inspired by Witteveen & Andrews (2019). This model introduces noise by sampling temperature based on the exponential mechanism, aiming to balance privacy with natural language generation capabilities. Tested for its effectiveness using the Matthews Correlation Coefficient (MCC) for authorship attribution and sentiment analysis, as well as semantic similarity and perplexity metrics to assess how well the original meaning is preserved and the quality of the generated text, this paraphrasing model proves superior to traditional word-level DP mechanisms. Although these methods show promise, it would be interesting to see more of the generated data used in tasks such as text generation, which is quite rare in the current literature. Most of the presented works focus on smaller tasks and models, but there is no limitation in using these methods to generate anonymized data to train LLMs.

# 5 Solutions for Preserving Privacy in LLMs: Model

In this section, we explore the various contemporary methods employed to ensure privacy preservation within machine learning models. The focus is on four primary approaches that operate directly on the model. Firstly, we explore the concept of Differential Privacy applied during the training phase (§5.1). This method involves training a model with techniques designed to protect the privacy of the data used by injecting noise (for an overview on Differential Privacy, see §2.2). Secondly, we examine the application of Differential Privacy during the inference phase (§5.2), where privacy-preserving mechanisms are applied when the model is making predictions. Thirdly, we discuss federated learning with Differential Privacy (§5.3), which combines federated learning techniques with Differential Privacy to provide data safety in decentralized environments. Lastly, we discuss machine unlearning (§5.4), a technique that enables an existing model to forget or unlearn specific privacy-sensitive information, and Homomorphic Encryption (§5.5), a cryptographic method that allows computation on encrypted data without decrypting it. Each of these methods offers unique benefits and challenges in the pursuit of robust privacy protection in large language models.

## 5.1 Training Large Language Models with Differential Privacy

> **Training LLMs with Differential Privacy**
>
> **Definition:** Training LLMs with Differential Privacy consists of all the approaches that apply Differential Privacy at training time during one of the training phases of Language Models.

Training a Large Language Model (LLM) typically involves three sequential phases: pre-training on a broad collection of unannotated corpora (phase 1), fine-tuning on a specialized corpus using instruction-tuning procedures (phase 2), and final alignment according to human preferences (phase 3). We refer the reader to §2.1 for an in-depth overview of how these phases work. In this section, we propose methodologies that guarantee privacy by using differential privacy during the pre-training phase (phase 1 - §5.1.1) or apply it during the fine-tuning phase (phase 2 - §5.1.2). We also analyze methods that focus on Parameter Efficient Fine-Tuning techniques (§5.1.3) while leaving out of the scope of this survey works that focus on alignment via Reinforcement Learning and differential privacy since, to the best of our knowledge, there are very few works that just started exploring this aspect (Wu et al., 2024a).

Despite the significant concerns over privacy, most of today's LLMs abstain from any privacy-specific measure for pre-training, relying instead on publicly available internet data for pre-training. While this data is public, it often contains sensitive information, posing risks that might harm individuals and undermine trust. Public data, thus, carries inherent risks yet remains a widely used resource. On the other hand, fine-tuning often requires using private datasets that entities prefer to keep confidential to protect sensitive information or maintain a competitive edge. We refer to *"public training"* when the training process involves using datasets that are available to the general public and to *"private training"* when the training process relies on restricted, confidential data that is not available to the public and must be carefully protected for privacy reasons. Here, Differential Privacy could offer a solution, minimizing risks associated with data memorization and safeguarding the privacy of individuals in these datasets.

We have already explained Differential Privacy (DP) in Section 2.2; here, we contextualize DP with its usage in modern large language models. Specifically, working with DP often involves deciding where to inject noise into the training process, typically by adding noise to the gradients, as shown in Step 2 of Figure 5.1. This approach generally involves using DP-SGD (Song et al., 2013), a variation of the classic SGD that clips gradient norms and then adds Gaussian noise to the average clipped gradient norm (see §2.2 for more details). It should be noted that, in principle, all the approaches involving DP-SGD are general, as DP-SGD is an optimizer that can be used with any network.

In the following sections, we will focus on how Differential Privacy can be applied to modern language models.

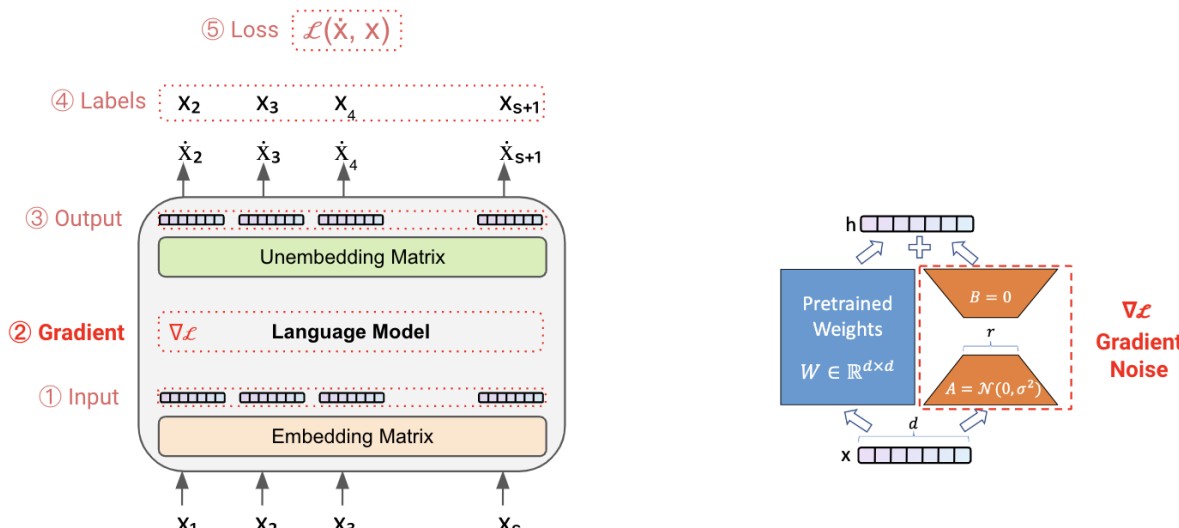

Figure 6: **Differential Privacy Applied on LMs.** Differential privacy represents one of the main solutions used to preserve privacy when training a language model. We have already discussed Differential Privacy (DP) in Section 2.2; here, we contextualize DP with its usage in modern large language models that, most of the time, consist of adding noise in the training gradients (DP-SGD). **On the right:** Adding noise to the gradients of all the training parameters in a language model is often expensive. Parameter-efficient fine-tuning techniques can reduce the number of training parameters, thus lowering the cost of applying Differential Privacy to these gradients. Picture adapted from LoRA (Hu et al., 2022), a popular PEFT method.

### 5.1.1 Pre-training with Differential Privacy

Pre-training a Large Language Model from scratch is usually a very cost-intensive procedure that just a few companies and institutions in the world can afford. In addition to this, adding differential privacy to this pre-training phase further increases the difficulties since (i) adding noise increases the already exorbitant training cost, slowing the convergence time, and (ii) the training phase requires a vast collection of private data, which is often scarce and difficult to obtain. Thus, most models pre-trained from scratch with Differential Privacy are relatively small compared to current Large Language Model standards. Nevertheless, they offer both theoretical and practical insights into why fully private pre-training is considered a sub-optimal strategy. Instead, a mixed solution involving pre-training on public data is recommended. In this section, we will explain why. All the works discussed are also summarized in Table 4.

Hoory et al. (2021) conduct a fully private training of BERT, utilizing DP-SGD during both pre-training and fine-tuning stages. The pre-training employs the standard BERT dataset from Devlin et al. (2019), supplemented with the MIMIC-III dataset (Johnson et al., 2016). Fine-tuning is then carried out on the i2b2-2010 dataset, sourced from the i2b2 National Center for Biomedical Computing and the NLP Shared Tasks Challenges, focusing on entity extraction within the clinical domain. The experiments are conducted with and without Differential Privacy, and the outcomes are measured by the F1-Score. Their findings indicate only a minor performance decrement with privacy-preserving methods, thus demonstrating a favourable privacy/utility tradeoff.

Yin & Habernal (2022) introduce a novel approach to fully pre-train BERT from scratch using a specialized legal tokenizer. They first perform conventional pre-training using BookCorpus and Wikipedia, followed by an additional pre-training phase using DP-SGD on a legal corpus, emphasizing their focus on safeguarding the privacy of the pre-training data. Subsequently, the model undergoes non-private fine-tuning on two downstream tasks: binary classification on the Overruling dataset (Zheng et al., 2021) and multiple-choice

QA on CaseHOLD (Zheng et al., 2021). These steps highlight the authors' intent to protect only the data used during the DP-SGD pre-training phase. The outcomes of this approach not only match but sometimes surpass those of non-private benchmarks, demonstrating the feasibility of maintaining high utility while employing privacy-preserving methods. However, the study lacks a specific privacy benchmark to assess any potential privacy leakage.

It is crucial to acknowledge that the discussed works were capable of conducting full training from scratch largely because they utilized BERT (Devlin et al., 2019), which possesses a relatively modest 109M parameters, a figure significantly smaller than those of modern LLMs such as GPT-3 with 175B parameters (Brown et al., 2020). Currently, the standard practice, even in settings where data are not private, and there is no need to perform DP tuning, involves leveraging a pre-trained model and fine-tuning it for specific tasks. This approach not only reduces computational demands but also allows the model to harness a broad and diverse initial training dataset, enabling it to subsequently adapt more effectively to a private dataset and its specific domain, often enhancing performance. However, the benefits extend beyond performance improvements. Recent studies, including Kerrigan et al. (2020) that compare public pre-training with differentially private fine-tuning against entirely private DP training, demonstrate significantly better results with the former method. This provides empirical evidence that DP learning greatly benefits from an initial phase of public pre-training. Full training solely on a private dataset often results in high perplexity values, indicating a model that would likely be ineffective. In Kerrigan et al. (2020), the comparison of models fully trained privately versus those benefiting from public pre-training followed by private fine-tuning demonstrates that public pre-training is essential for enabling effective private learning. Additionally, since DP-SGD introduces noise during training, limiting this noise to the fine-tuning phase rather than the entire training process results in less overall noise and better performance. This improvement is due to the general (clean) knowledge acquired during the pre-training phase. Applying DP-SGD to an LLM pre-training (which is a huge part of the training) would make it extremely slow and add a lot of noise, which could hinder the final utility.

Ganesh et al. (2023) present a study that compares the performance obtained via a public pre-training followed by private fine-tuning with a scenario where both stages are conducted in a differentially private manner. This paper emphasizes the theoretical underpinnings of Differential Privacy in training models, arguing that for a differentially private model to be effective on the task it is trained on, it is essential to first undergo pre-training on public data. The theoretical basis provided in the study explains that optimization involves two phases: initially, the model transitions from a random initialization towards a favourable basin, and subsequently, it addresses a convex optimization problem to find the local minimum within that basin. If attempted entirely within a private context, incorporating private data and noise addition, the initial stage would necessitate substantially more data to achieve comparable outcomes to those achievable with public data, as supported by experimental evidence. Moreover, given the typical scarcity of private data, *it might be impractical or even impossible to gather sufficient data to effectively train a model privately from scratch, especially if the model is an LLM.* Therefore, pre-training on a larger public dataset not only aids in overcoming these limitations but also enhances the model's resilience to distribution shifts between the public and private datasets.

Tramer & Boneh (2021) offer another perspective, arriving at a similar conclusion: public pre-training is crucial for achieving desirable outcomes in a private learning environment. Although this study is centred on image classification using CNNs, the findings reinforce that the advantages of public pre-training transcend various contexts, including this one. In their approach, the researchers enhance the performances of fully private training by incorporating handcrafted features, which yield improved results. Nevertheless, these outcomes still do not match those achieved through a combination of public pre-training and private fine-tuning. **The collective insight from numerous studies is that a public pre-training phase is indispensable in private learning scenarios.**

While the works reviewed so far performed a "classical pre-training" (i.e., gaining knowledge from a large dataset), Yu et al. (2024) demonstrate that pre-training can also be conducted selectively, i.e., by focusing on a specific (smaller) subset of data that allows DP training to become more efficient. Their approach involves a novel technique of selecting a subset of the pre-training dataset, OpenWebText (Gokaslan & Cohen, 2019), whose distribution closely resembles the fine-tuning dataset's distribution, Enron email dataset (Klimt & Yang, 2004). To filter the pre-training dataset, a classifier trained with Differential Privacy predicts whether

a pre-training sample belongs to the distribution of the fine-tuning (private) dataset. Each sample in the pre-training dataset is then scored for how likely it is to belong to the fine-tuning dataset, then the top-k samples are used for public pre-training. This is followed by differentially private fine-tuning on the private dataset, with the final privacy loss taking into account both the classifier and the ultimate model. This method of dataset selection is designed to minimize distribution shifts, enabling smaller models to achieve better results. The results demonstrate strong performances, surpassing more complex baselines and thus enhancing efficiency to the point where this can be considered a model compression tool. However, a potential oversight of this method is the non-consideration of biases within the fine-tuning dataset, which could lead to the inadvertent transfer of these biases into the model through the selected pre-training samples. This is not as likely when the pre-training dataset is fully utilized.

### 5.1.2 Fine-tuning with Differential Privacy

Pre-training a large language model from scratch, with or without Differential Privacy, has become wildly expensive, given the sheer size of these models. Currently, the prevalent approach consists of using an already available LLM pre-trained checkpoint and fine-tuning it for specific downstream tasks, including applications of Differential Privacy (DP). In this section, we survey several different works that apply Differential Privacy on top of already pre-trained LMs. These works are also available in tabular format in Table B. While the works in this section are all centered around LMs, they rarely adopt solutions specific only to the language domain, making these methods generalizable to other domains. Yue et al. (2023) show that using DP-SGD, it is possible to successfully train LMs for text generation. They apply DP-SGD to fine-tune various sizes of GPT2 using the Open Yelp Dataset. This is mainly done to generate synthetic reviews that cannot be traced to the actual reviewer. The generated reviews are then used to train classifiers to predict ratings and business categories. The quality of the synthetic text is assessed by comparing its distribution to the original dataset (through F1-score, Fréchet Inception Distance Heusel et al. (2017), and MAUVE Pillutla et al. (2021)) and the classifier performance against models trained on the original data. In this task, the largest GPT2 models produce data that more closely resemble the original data, probably because they are less influenced by DP. Experiments indicate that DP introduces a minimal performance drop, often less than 0.01%, within a privacy budget of $\epsilon = 4$. Remarkably, the GPT2-Large model with DP nearly matches the performance of its non-private counterpart, with only a 0.008% difference in accuracy on downstream tasks. This leads to the notion that the pipeline is particularly suitable for LLMs, as it tends to scale well with model size. Privacy testing was performed following the method by Carlini et al. (2019): this consists of injecting canaries inside the training set in order to extract them later on, to prove that the model is capable (or not) of memorizing training data. When these canaries were injected into the training data, the model could not be prompted to reproduce them under DP conditions, even with a repetition rate of 100. In contrast, without DP, the canaries were reproduced 80% of the time, demonstrating effective privacy preservation. This work highlights the feasibility of DP fine-tuning. When we investigate DP learning and the used techniques, there is a constant comparison with standard learning, as it is the most widely known and adopted. Papernot et al. (2020) specifically focuses on the differences between private and public training, trying to shed some light on the best conditions for private training. They investigate architectures, parameter initializations and optimizers. We are not very interested in architectures as they do not consider transformers but only convolutional architectures and we already know from §5.1.1 that it's best to start from parameters that were previously optimized through a public pre-training. The novelty that this research points out is that adaptive optimizers like Adam provide marginal gains (if any) in this context. Putting in direct comparison DP-SGD and DP-Adam, they find that while the latter converges faster initially, it eventually slows down to the point where DP-SGD can outperform it, reaching a lower loss. Another relevant difference they point out is how different a working learning rate is when tuning DP-SGD with respect to standard SGD (with all the other conditions being the same). Definitively this work highlights how DP-tuning, while effectively working, requires to be handled in a slightly different way than a standard tuning and, while this work only considers CNNs, there is no reason to believe that the findings would not apply to LLMs.

Li et al. (2022), building on the work of Papernot et al. (2020) on how to approach private-tuning, explore the effects of hyperparameters and fine-tuning objectives, identifying typical pitfalls of DP tuning when it's treated as normal tuning. Their findings are particularly relevant as they suggest that, with proper execution, the performance of DP-tuned models can rival strong non-private benchmarks. They introduce a

technique known as *Ghost Clipping*. Ghost Clipping is an important step in the landmark of private tuning because it greatly reduces memory usage, aligning the efficiency of private learning with that of non-private learning. Before that, DP-tuning was much slower and heavier than normal tuning to the point where it was almost not feasible (especially for large batch sizes, which are necessary for models like BERT) and even if it was, it would reach much lower performances. Ghost Clipping is an expansion of the concept described in (Lee & Kifer, 2021), it avoids the need to instantiate per-example gradients for each linear layer by handling one layer at a time and necessitating an additional backpropagation pass per batch. Extensive testing across various models—RoBERTa for natural language understanding (NLU), GPT2 for natural language generation (NLG), and both GPT-2 and DialoGPT for chit-chat dialogue generation—reveals that private models perform nearly as well as public ones, with only minor differences in perplexity and BLEU scores, showcasing competitive results. When compared to using vanilla Opacus, Ghost Clipping shows a 3x improvement in efficiency. While Ghost Clipping is already a big step in efficiency, Ding et al. (2024) points out that this method cannot be applied to standard Transformers because of non-plain feed-forward (and back-propagation) topology caused by embedding sharing that hinders Differential Privacy guarantees. The authors proposed *Phantom Clipping*, which is able to handle parameter sharing thanks to the manipulation of tensor operations. This ensures that the shared parameters' gradients are computed and clipped without redundancy, hence preserving the Differential Privacy guarantees. They also show and solve another problem typical of DP-tuning with transformers: attention distraction. Attention distraction is a phenomenon that occurs when noise added in DP-SGD to maintain privacy distorts the self-attention mechanism, typical of transformers networks, causing the model to inaccurately weight parts of the input data. This leads to less effective learning as the model might focus on irrelevant details or overlook important information. To address this, the proposed "Re-Attention Mechanism" recalibrates attention scores by tracking and adjusting for the variance introduced by the noise. This adjustment ensures that attention scores more accurately reflect the true underlying data relationships despite the noise, thus maintaining the model's performance and efficiency under differential privacy constraints. It should be noted that these issues and fixes are general to transformers model and not specific to LLMs. However, they can greatly speed up fine-tuning for LLMs, which is quite important given their size. *Ghost Clipping* and *Phantom Clipping* are already a big step towards making private learning as efficient as standard learning, but there is still a gap to close. In order to get even closer, Bu et al. (2023b) advance the methodology proposed by Li et al. (2022) with a novel Book-Keeping (BK) technique. This new method requires only a single back-propagation round and does not necessitate the instantiation of per-sample gradients. As a result, it significantly enhances throughput compared to Ghost Clipping while maintaining a similar memory footprint. This means that this new technique is a big step forward as it offers a more scalable solution for handling data of higher dimensions.

It is often possible that there is some overlap between pre-training data and fine-tuning data. While this is usually not an issue, in a private setting, where only fine-tuning is done privately and pre-training is standard, this kind of contamination would result in a data leakage that could expose sensitive data. In this sense, the work done by Kurakin et al. (2024) is important as it addresses the potential data contamination between pre-training and fine-tuning datasets. To tackle this issue, they employ an LLM based on LaMDA (Thoppilan et al., 2022), initially applying deduplication techniques (Lee et al., 2022) to ensure fine-tuning data are excluded from the pre-training dataset. Following deduplication, they proceed with standard pre-training and subsequently conduct private fine-tuning using both full and Parameter Efficient Fine-Tuning (PEFT) methods. The model is then utilized to generate a synthetic dataset for a downstream sentiment analysis task, employing BERT and WordCNN for classification. Although the results are good, showing only minor performance drops for DP-tuning with respect to standard tuning, it is important to note the use of relatively high privacy budgets (e.g., $\epsilon$=10). Moreover, this approach is tested only on binary classification tasks, as opposed to the more complex multi-class classification challenges addressed by Li et al. (2022).

Mattern et al. (2022a) introduce a variant of DP fine-tuning by applying it to GPT2 for generating private versions of IMDb and Amazon reviews datasets, which are then used to fine-tune models for classification. Following the methods of Schick & Schütze (2020), during the fine-tuning of GPT-2, they enhance sample generation fidelity by incorporating instructions based on the original dataset's characteristics, such as sentiment or category, into templates. Since the datasets are made for classification, the data is labelled. The idea is to use these labels to steer the generation, also during training, by asking specifically for a sample corresponding to a label, like, for example, asking for a "positive" review. This approach integrates

classification labels directly into the LLM's fine-tuning prompts to prevent generating mismatched samples (like a positive review that reads like a negative one) and introduces a novel loss function that penalizes such mismatches. Since this method takes advantage of prompts to steer the generation, it is specific to LLMs. After training, the model samples a subset of the original training set, which is then utilized for further fine-tuning on binary classification tasks. For comparative analysis, the authors also fine-tune models using DP on the actual data. The main result is the demonstration that models trained on synthetic data generated via DP and then fine-tuned in a non-private manner outperform those trained directly on real data with DP. Additionally, the researchers assess data privacy by examining duplicates between original and synthetic datasets, noting a significant reduction, with duplicates completely eliminated in some instances. In this study, similarly to what was shown by Kurakin et al. (2024), the downstream task is binary classification, which is not particularly hard.

Differential Privacy (DP) introduces various challenges, notably due to the addition of noise. One such issue, the increased memory consumption from noise addition, is addressed by Ghost clipping (Li et al., 2022). To further mitigate these challenges, Behnia et al. (2022) introduce a new framework designed to reduce the overall amount of noise required to maintain privacy guarantees. This framework utilizes the Edgeworth accountant (Wang et al., 2023), which provides a "finite sample guarantee" unlike the typically asymptotic nature of DP guarantees. This is particularly advantageous for fine-tuning an LLM where the number of tuning steps is relatively limited. By calculating the necessary noise levels based on privacy parameters and the number of compositions for a specific dataset, this new framework significantly reduces noise impact—by up to 5.6%. Additionally, it achieves a performance improvement of up to 1.1% across various natural language understanding (NLU) tasks compared to previous benchmarks. This makes the approach relevant as it not only enhances privacy but also improves efficiency in DP applications.

Although pre-training on public data followed by fine-tuning on private data has long been considered the standard and safest method for private model training, recent evidence suggests that even this approach carries certain risks. Feng & Tramèr (2025) recently showed how privacy backdoors can be injected in pre-trained models, making them susceptible to different kinds of attacks. This research explores how neurons can be subtly manipulated to retain their primary function while gaining malicious capabilities. By enabling neurons to hold a gradient for a single input and "deactivate" to prevent overwriting during subsequent training, the model becomes vulnerable to gradient inversion attacks, similar to those used in federated learning models (see 3.3 for details). While such attacks typically require white-box access, the researchers also demonstrate attacks on black-box models. Specifically, they enable membership inference attacks (MIAs, see 3.2) by modifying units in the model's second-to-last layer, which are uniquely activated by specific training data points. By adjusting the model's activation, they can determine whether a sample was part of the training set by comparing logits before and after fine-tuning. Although the study only tests on MLPs, ViT, and BERT, and it remains unclear if these vulnerabilities apply to LLMs, it highlights a significant risk in the public pre-training and private fine-tuning paradigm: if the pre-trained model or its provider is untrusted, loose privacy budgets commonly assumed in the literature are unsafe. Any efforts to establish tighter privacy guarantees must account for the model's integrity.

### 5.1.3 Parameter-Efficient Fine-Tuning with Differential Privacy

Modern large language models (LLMs) often contain billions of parameters, making even full fine-tuning impractical due to high computational demands, particularly when paired with noise injection for Differential Privacy (DP). Instead, Parameter Efficient Fine-Tuning (PEFT) methods like adapter tuning (Houlsby et al., 2019), compacter tuning (Karimi Mahabadi et al., 2021), prompt tuning (Lester et al., 2021), prefix tuning (Li & Liang, 2021), and notably LoRA (Hu et al., 2022), have become more prevalent. Around the same time LoRA came out, also Reparametrized Gradient Perturbation (RGP) was introduced (Yu et al., 2021), although with less success. RGP is a PEFT technique specific to the field of DP. It enhances efficiency by reconfiguring each weight matrix into smaller matrices, optimizing gradient storage and computation. These techniques, which involve adapting a subset of the model's parameters or incorporating external parameters, facilitate faster training and enhanced modularity. They allow the model to manage various tasks without requiring complete re-training. Remarkably, even with fewer parameters being updated, these methods often achieve performance that is comparable to or even surpasses full fine-tuning, depending on the specific task

and dataset (Sun et al., 2023; Suri et al., 2023). Because of the huge number of parameters in LLMs, they seem to suit particularly well these methods, even though there is no theoretical limitation in applying them to other models. In this section, we survey methods that apply Differential Privacy with PEFT techniques in order to reduce the computational burden of fine-tuning while keeping privacy standards. A synthetic overview of these methods is available at B.

Yu et al. (2022) applied all the PEFT techniques just mentioned with DP to RoBERTa and GPT2 without any particular challenge. As a matter of fact, they achieve performances nearly equivalent to their public counterparts, especially with RoBERTa, where many PEFT techniques with DP notably outperformed full fine-tuning. The study also highlights the efficiency of DP when combined with LoRA in terms of speed and memory use, underscoring the higher viability of private learning in a PEFT framework. This is done by using DP-SGD with LoRA.

In a continuation of similar research, Li et al. (2022) conduct tests with both RoBERTa and GPT2, employing DP-Adam, which is similar to DP-SGD but based on Adam, for updates and moment accumulation) and the Ghost Clipping technique. This introduction reduces memory usage, making the difference in efficiency between private and non-private learning negligible (see §B for more info). At the same time, also the performance drop is proven to be negligible, making DP-PEFT a viable option. GPT2 is evaluated in tasks like table-to-text and chit-chat generation, where the performance of full fine-tuning and PEFT proves to be closely matched, with full fine-tuning showing a slight edge both with and without Differential Privacy. This indicates that PEFT can match the efficacy of full fine-tuning even in a private setting. Additionally, the results challenge previous assumptions seen in the work of Yu et al. (2022), where PEFT's superior performance was attributed to lesser noise addition. The findings here suggest that higher noise levels in full fine-tuning might not detrimentally affect the privacy/utility tradeoff as previously thought. This observation prompts further inquiry into the mechanisms of private learning, offering a pathway to deeper insights into how different fine-tuning methods impact model performance under privacy constraints.

While LoRA remains the predominant technique for private tuning of language models, alternative methods also demonstrate effectiveness. Li et al. (2023e) introduce RAPT, a privatization framework designed for prompt-tuning and prefix-tuning on a model that does not necessarily need to be locally run. This is made specifically to make it possible to privately train a remote language model through prompt tuning APIs, like Nvidia NeMo. This involves adding noise to the prompts before they are sent, addressing the issue that direct prompt tuning on privatized data can significantly impair downstream performances. To counteract this, they propose a novel privatized token reconstruction task, which is trained concurrently with the downstream task. The experiments utilize BERT and T5 models across various datasets such as SST (Socher et al., 2013), QQP (Lili Jiang, 2017), and TP-UK (Hovy et al., 2015), with adjustable privacy parameters. Privacy is assessed through the defensive capabilities against both an embedding inversion attack (see §3.3.1) and an attribute inference attack, where both prompt-tuning and prefix-tuning show enhanced robustness. Utility tests reveal that performances suffer without the use of the privatized token reconstruction, which significantly boosts performance and confirms its efficacy in helping LLMs develop better representations.

## 5.2 Inference with Differential Privacy

> **Inference with Differential Privacy**
>
> **Definition:** Inference with Differential Privacy consists of all the approaches that apply Differential Privacy at inference time without changing internal parameters. DP is used in two classes of DP-inference approaches: differentially private in-context learning (ICL) or noising-denoising external modules.

The methods presented here, be them in the first or second class, are all specifically devised for LLMs as they take advantage of prompting.

The first class of DP-inference models is based on In-context learning (ICL) (Brown et al., 2020) (see §2.1). Indeed, ICL may be used to increase privacy in LLMs since private data can be given to LLMs only at

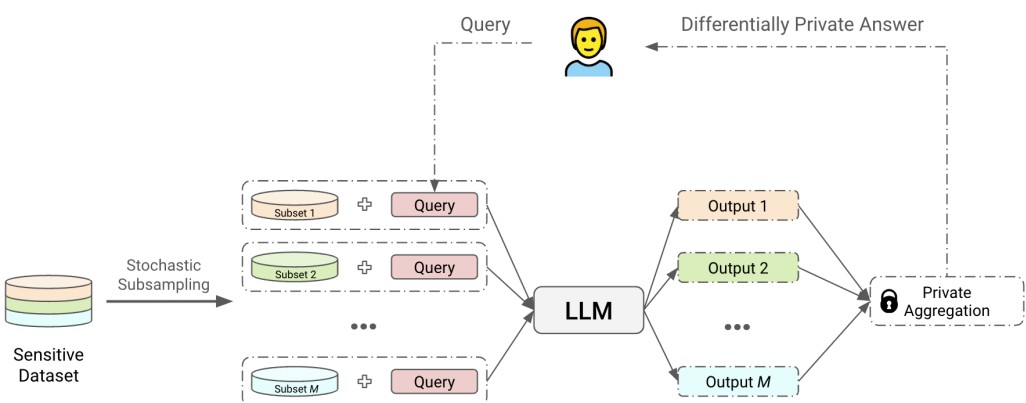

Figure 7: **Differentially Private In-Context Learning.** High-level description of applying Differential Privacy during in-context learning: the original dataset is used to create several subsamples via a stochastic sampling procedure; these subsamples are then combined with the user query (prompt) and fed to an LLM; all the generated outputs are aggregated in a private fashion, producing a single differentially private prediction.

inference time. Thus, this private data should not be included in generalistic LLMs and can stay under the control of the owners holding the data rights. For example, if a medical decision on a specific patient should be taken by considering all patients in a specific hospital, these patients can be given as part of the prompt in in-context learning (ICL) approach. However, nothing guarantees that a malicious user can try to extract this personal data by using specific prompts following the general ICL prompt (see §3.1.2). This is an active area of research that aims to embed Differential Privacy mechanisms in inference, performing what is called *differentially private in-context learning*. The work presented here all follow a similar high-level structure represented in Figure 5.2: the original dataset is used subsampled or partitioned in several subsets; the subsamples are then combined with the user prompt and fed to an LLM; all the generated outputs are aggregated in a private fashion, producing a single differentially private prediction.

To address the issue rising with ICL, Wu et al. (2024b) introduce a novel approach, called Private in-context learning, to ensure privacy in ICL operations by partitioning the private dataset into M subsets, each containing N samples. These subsets are used as N-shot examples for generating M different ICL prompts, which in turn produce responses for the next token prediction or output classification. Results from each prompt are then privately aggregated using techniques tailored to the specific task: (1) Report-Noisy-Max with Gaussian noise for text classification that ensures differential privacy; (2) aggregation methods for text generation. Two aggregation methods are explored: Embedding Space Aggregation, where outputs are mapped to an embedding space to compute and noise the mean output, followed by querying the LLM multiple times to find the closest match with cosine similarity, and Keyword Space Aggregation, which involves noisy top-k counting of words across outputs using mechanisms like the Joint Exponential Mechanism (Gillenwater et al., 2022) or Propose-Test-Release (Zhu & Wang, 2022). Despite operating within a low privacy budget (e.g., $\epsilon = 1$), Private in-context learning (ICL) achieves performance comparable to non-private methods across various tasks such as text classification, document question answering, and text summarization. This method not only demonstrates minimal performance drop with added privacy but also offers a less computationally intense alternative to fine-tuning with DP, providing a practical, lightweight solution for enforcing privacy in LLMs.

Tang et al. (2024) added another security level on top of the method of Wu et al. (2024b): their method generates samples with DP before giving the input as in-context learning to LLMs. Then, the rest is the same, that is, possibly dividing the private dataset into M subsets containing the N samples. Clearly, Tang et al. (2024) show that the distortion of the input private data samples has little influence with respect to target applications. The effectiveness of this method is examined over benchmarks on standard Text

Classification and Information Extraction datasets. Performance results are presented as the mean and standard deviation across five runs. The private models demonstrate performance on par with non-private models, indicating that it is feasible to achieve private learning without sacrificing accuracy. The study also evaluates the performance of Gaussian and Exponential mechanisms in the aggregation phase, with the Gaussian mechanism consistently showing slightly better results. This reinforces the viability of using synthetic data for privacy-preserving ICL applications.

To improve the level of security in these ICL-based approaches, Duan et al. (2024a) propose to use a teacher-student approach where real data are given to the teacher model in a way similar to Wu et al. (2024b). The teacher model is then used to give synthetic examples for the ICL of the student model. In this way, the student model, which is exposed to final users, does not have access to real data. The method is called PromptPATE, and follows the same general workflow of the standard Private Aggregation of Teacher Ensembles (PATE), presented by Papernot et al. (2018): first, an ensemble of teacher models is trained, then private knowledge transfer is performed, and lastly, a student model is trained. All this training has to be intended as ICL. In this case, instead of training the teacher models, the private dataset is used to generate a set of disjoint prompts for the LLM, so teacher models (or the "flock of stochastic parrots") are actually models performing ICL with disjoint parts of the original dataset. The private knowledge transfer is basically a majority voting from the teacher models made in order to choose the next token prediction. The noisy majority voting is done with the Confident GNMAX algorithm (Papernot et al., 2018) just like in the standard PATE. The chosen token is added to the public input to create a single, differentially private example that can replace private data for ICL on a student model so that the student model does not actually need to undergo real training but can take advantage of the elasticity of prompting. Similarly to Tang et al. (2024), this approach also involves generating synthetic samples for later use in ICL. This method, too, achieves performance comparable to non-private methods, even under stringent privacy conditions.

Hong et al. (2024) present a similar work, with a different aim, which is prompt engineering, and some differences in the aggregation mechanism, called DP-OPT (Differentially Private Offsite Prompt Tuning). In this approach, an ensemble of local models receives disjoint subsets of data to generate the next token. The Exponential mechanism is then employed to privately select the token that appears most frequently. Following this, a Deep Language Network, as described by Sordoni et al. (2023), is used to identify the optimal candidate prompt, applying the Exponential Mechanism again to ensure confidentiality. A significant advantage of this method is that the LLM-engineered prompts are transferable, often yielding results on par with non-private alternatives, depending on the task and the underlying model. The primary distinction here is that DP-ICL is utilized to develop prompts based on training data without any data leakage. This technique demonstrates competitive results on downstream tasks and effectively transfers across several models, including GPT3.5, Llama2, and Vicuna.

The second class of DP-inference models is introduced by Mai et al. (2024). Their method involves partitioning the network into three components: a local encoder responsible for adding Laplacian noise to the data, a remote (often black-box) model such as GPT2, T5, BERT, or Llama2 that processes the privatized embeddings and returns the output, and a local denoising module that refines the output. This configuration enables the use of black-box models while maintaining privacy by adding local preprocessing and postprocessing steps. Since the text is perturbed locally, Differential Privacy is applied on the input, which corresponds to 1) in Figure 5.1 (DP on input). Tested on various downstream tasks, this method shows promising performances, closely trailing behind its non-private counterparts. Additionally, the study conducted inversion attacks to assess robustness, revealing that BERT is particularly resilient at lower noise levels, GPT2-XLarge performs better under higher noise conditions, and T5 appears to be the most susceptible, requiring stringent privacy settings to ensure safety.

Similarly, Tong et al. (2024) propose InferDPT, which includes a local perturbation algorithm called RANTEXT (discussed in §4.1.2). The perturbed text is sent to a remote black-box LLM, and a local extraction module — a smaller model — processes the perturbed output, reconstructing it to ensure coherence, consistency, and alignment with the original prompt. Users then combine the original prompt and the perturbed generation results to produce the final output. InferDPT, using the RANTEXT mechanism, achieves text generation quality comparable to non-private GPT-4, maintaining high utility despite the added privacy-preserving perturbations.

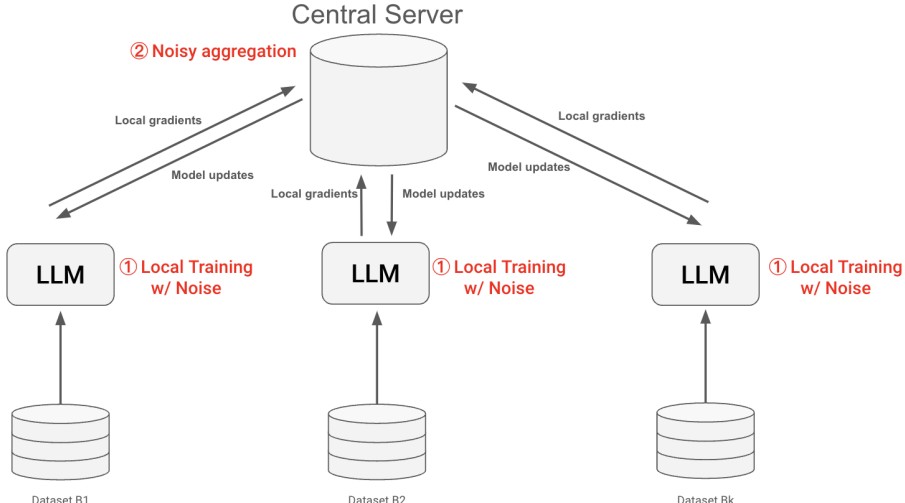

Figure 8: **Federated learning with Differential Privacy.**Differential Privacy in a federated setting can be applied according to 2 different models: 1) in the Local model, local gradients are clipped and noised before being sent to the central server; **or** 2) in the Central model plain gradients are sent to the central server which aggregates them in a noisy manner.

## 5.3   Federated Learning with Differential Privacy

Differential Privacy and federated learning are two complementary techniques in privacy-preserving machine learning that tackle different aspects of data security. Federated learning (§2.3) is a distributed computation method that enables learning on private data from various sites without transferring the data offsite. However, this alone is not sufficient for data security, as gradient inversion attacks can potentially recover the original training data by intercepting gradients sent from clients to the server (check discussion in §3.3.2). While federated learning helps mitigate memorization issues (Thakkar et al., 2021), real privacy guarantees require Differential Privacy. In federated learning, Differential Privacy can be implemented in different ways: Figure 5.3 illustrates its application either locally (local Differential Privacy) or globally (central Differential Privacy).

A clarification is needed: central and local models of Differential Privacy (DP) are not exclusively related to federated learning (FL). In central Differential Privacy, perturbation is applied at a global level, typically during aggregation. In contrast, local Differential Privacy applies perturbation locally before the data leaves the site. While DP learning often aligns with FL and involves perturbing gradients, DP was initially developed for the statistical analysis of databases, and both central and local models exist independently of learning processes. §C.1 provides several examples of techniques that fall under the local model of DP, where text perturbation is applied to create a noisy private dataset that can be shared later, while §C contains examples of techniques of central Differential Privacy, where the noise is applied at aggregation time.

Unfortunately, at the moment of writing, the intersection of differential privacy and federated learning did not produce much that is very relevant for LLMs, mainly due to the size of the models, which makes difficult parameter sharing and updating, and due to data dimensionality, which is an important constraint for text. There is still some interesting research we came across, even though not very relevant for LLMs, hence we leave the discussion for that in the appendix at §B.

The most notable example of central differential privacy is Google's implementation in Gboard. Xu et al. (2023b) discuss and analyze twenty Gboard language models trained for Next Word Prediction using DP-FTRL (refer to §B for a thorough explanation), indicating that all future Gboard models will incorporate differential privacy guarantees. In their DP-FTRL setup, model updates undergo clipping, and noise is applied during global model updates, ensuring Central Differential Privacy. Adaptive clipping, which adjusts

the clip norm each round by privately estimating the model's norm at a targeted quantile as described by Andrew et al. (2021), was also utilized. However, adaptive clipping did not show a significant improvement and was less robust during experiments with large report goals, sometimes causing catastrophic failure or a lack of progress in the first 1000 rounds. Despite these issues, adaptive clipping can help reduce hyper-parameter tuning when the privacy budget allows. Public pre-training on C4, as explained by Wang et al. (2024b), reduced the rounds needed to achieve target utility by about 1000 under the same noise multiplier, thereby enhancing privacy guarantees. Additionally, DP-FTRL was combined with Secure Aggregation, a cryptographic protocol ensuring the central server only accesses aggregated updates and not individual client updates, as described by Bonawitz et al. (2017). While Secure Aggregation supports DP, it introduces considerable computational slowdown and requires different system configurations for large report goals. The techniques for training language models with differential privacy in Gboard have resulted in models with stronger privacy guarantees than some established standards, such as those used by the US Census Bureau. In Europe, these differentially private neural network models outperform older N-gram models in terms of utility. In the US for English and in Brazil for Portuguese, the new models are comparable to or slightly better than their non-private versions.

For the local model of differential privacy, most works (as discussed in §B) use it for telemetry data collection, which means that they use very low dimensionality data. Data and model dimensionality are very problematic in this setting. Luckily, some recent developments can mitigate these problems. There is a very recent work (Liu et al., 2024b) that shows how this model can be applied to an LLM thanks to LoRA. In this work, the authors developed a federated version of LoRA with Differential Privacy, which allows the sharing of a much smaller number of parameters while keeping the privacy guarantee. The method makes use of LoRA to locally train the model, performs clipping and noising at a local level and then uploads the weight updates to the central server. They use this method to fine-tune Llama-7B and ChatGLM-6B for general and medical question answering, respectively, using SlimPajama and a medical dataset crawled from two online healthcare services platforms. The results for several $\varepsilon$ and $\delta$ show that, once again, there is a degradation in performance when using DP, but it is mostly negligible, especially considering the privacy gain. Unfortunately, in this paper, there is no concrete experiment to test to which extent this method prevents training data leakage.

### 5.4 Machine Unlearning

> **Machine Unlearning**
>
> Machine unlearning refers to the process by which a machine learning model is able to forget or remove knowledge about certain data points from its training (Cao & Yang, 2015).

Recent regulations like GDPR guarantees the "right to be forgotten", which allows individuals to withdraw their consent and request the deletion of their personal data (Zhang et al., 2024b). Unfortunately, differential privacy (DP) cannot guarantee this because, for a data sample to be truly "forgotten", it must have zero influence on the training process. This requirement translates to a privacy budget of $\epsilon = 0$, which would prevent the model from learning anything.

Machine Unlearning (Cao & Yang, 2015) is an orthogonal field to Differential Privacy, focusing on how to eliminate the influence of specific data samples. The key works related to this topic are summarized in Table 7 in the Appendix. The aim is to make the model equivalent to one trained on a dataset that excluded unwanted entries, as illustrated in Figure 5.4. While test-time prediction to measure the influence of training data have been proposed defining influence functions (Koh & Liang, 2017), this technique does not reveal the impact of individual data point model parameters. Although training strategies designed to ensure unlearning have been proposed and tested for different architectures (Bourtoule et al., 2019), most of the unlearning solutions for LLMs adopt the idea of modifying model parameters after the training phase.

Jang et al. (2022) propose a method focused on unlearning specific information, particularly for language models. This method negates the training loss function – performing gradient ascent – to maximize loss on target sequences, enabling unlearning with minimal performance loss on unrelated sequences. Sequential unlearning proves more effective than batch unlearning. Wang et al. (2024c) propose to perform the un-

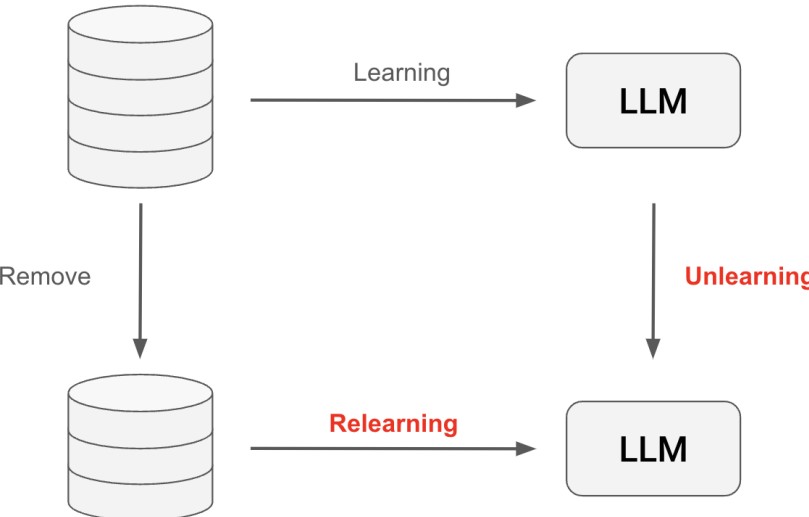

Figure 9: **Machine Unlearning.** Machine Unlearning refers to the process of removing the influence of specific training samples from a machine learning model. This process aims to make the model as if it was never trained on the removed data. Essentially, it ensures that any impact the specified data had on the model's parameters is reversed, effectively erasing its contribution. The image shows that the only real alternative to Unlearning is Relearning, which theoretically works but is extremely impractical for big models like LLMs.

learning only of problematic spans in the sentence, rather than on entire instances to minimize the impact on performance. Despite its effectiveness, Eldan & Russinovich (2024) show limitations of gradient ascent in certain scenarios, such as making Llama2-7B forget about Harry Potter. The challenge lies in the high likelihood of several candidate tokens related to Harry Potter, necessitating many gradient descent steps. To address this, they map Harry Potter terms to generic equivalents and then compare predictions from a baseline model with mapped input and a reinforced model trained on Harry Potter with the original input. By reducing the likelihood of tokens predicted by the reinforced model, they fine-tune the baseline model using generic labels. This results in the model losing familiarity with the target knowledge, with minimal leakages resembling secondary knowledge. Performance drops vary by task but remain generally tolerable. Recent findings suggest that gradient ascent should be combined with other techniques to be effective while maintaining model utility: Yao et al. (2024a) describe a unified unlearning framework for LLMs and find that gradient ascent is more robust when combined with gradient descent on a retain dataset.

Chen & Yang (2023) propose creating unlearning layers and training them with a selective student-teacher objective based on KL-divergence. The student model (the additional layers) maximizes divergence from the teacher model (the LLM) on the data to be forgotten while minimizing it on the rest. Different unlearning layers handle different data, but they also provide a method to fuse these layers into one. The model is evaluated on three sets: the Retained Set (where high performance is desired), the Forgot Set (where low performance is desired), and a Test Set. This method yields excellent results, outperforming other unlearning techniques on the Test Set, maintaining raw performance while effectively unlearning the target data.

Unlearning can also be achieved with Reinforcement Learning through an approach called DeMemorization (Kassem et al., 2023). This method takes samples from the pre-training dataset, divides them into prefix and suffix pairs, inputs the prefix, and uses a negative similarity metric to evaluate how different the generated suffix is from the real one. This metric serves as a reward signal, encouraging the language model to develop a paraphrasing policy that generates different tokens to minimize memorization. DeMemorization achieves better performance on downstream tasks because it doesn't remove samples from the dataset. However, for the same reason, it shows less robust privacy performance than actual unlearning methods, though the privacy performance remains high.

In Ishibashi & Shimodaira (2023), it is argued that unlearning approaches can lead to hallucination and may not make the generation harmless. To address this, they propose a different approach called knowledge sanitization. This approach uses LoRA to instruct the model to respond with a *sanitization phrase* like "I don't know" when encountering potentially harmful prompts. To achieve this, they build a training set where answers to harmful questions are removed. This method, similar to Chen & Yang (2023), is evaluated on both data to be retained and data to be forgotten. The performance is high for both retaining and forgetting targets.

For large language models, In-Context Learning can be used for unlearning, as demonstrated by ICUL (In-Context UnLearning) (Pawelczyk et al., 2024). This technique builds a context where the data point to be unlearned is included but with a flipped label (e.g., changing a positive label to a negative) alongside correctly labelled examples. Compared to Jang et al. (2022), this approach achieves similar performance for both unlearning and the primary task without requiring any parameter updates. While this method was tested on a binary sentiment classification task, theoretically, it could be applied to other tasks by incorrectly labelling categorical samples, though no experiments have been conducted in such contexts to our knowledge.

Model Editing can also be used, in principle, to perform unlearning of private information. Building on the idea that the linear layers in the Transformer architecture can be interpreted as key-value memories that store information (Geva et al., 2021), different model editing techniques have been developed to modify – without retraining – these specific parameters. Meng et al. (2022) introduce ROME (Rank-One Model Editing), allowing factual predictions to be traced to specific layers for individual manipulation. Furthermore, a similar approach called MEMIT (Meng et al., 2023) proves to effectively modify multiple examples without affecting the utility of the model. While grey-box attacks that examine intermediate hidden states can still recover deleted data 38% of the time (Patil et al., 2024), and being mostly focused on factual associations, ROME and MEMIT show potential for tracing other data types, substituting the private information with an anonymous one.

With a more specific focus on privacy, Wu et al. (2023) introduce DEPN (Detecting and Editing Privacy Neurons) inspired by ROME. This method assume that private information, like factual information, might be encoded in specific neurons. They propose detecting these neurons and setting their weights to zero to block information flow. Results show good validation perplexity, sometimes surpassing models trained with Differential Privacy but with higher privacy risks. However, DEPN significantly reduces training time, offering a better balance between computational efficiency and performance. Venditti et al. (2024) focus on the idea of breaking the association between an individual and their private information inadvertently included in the training data. Their method is called PAE and is based on MEMIT: PAE aims to make multiple edits feasible, both in a sequential and batch fashion, by taking into account the norm of the update matrix with respect to the original weights without sacrificing the model's utility.

## 5.5 Homomorphic Encryption

> **Homomorphic Encryption**
>
> **Definition:** Homomorphic Encryption (HE) is a cryptographic technique that allows computations to be performed directly on encrypted data without the need to decrypt it first. The result of these computations, when decrypted, matches the result of operations performed on the plaintext data.

Homomorphic Encryption (HE) offers a theoretical framework for preserving privacy in machine learning by enabling computations on encrypted data. In the context of Large Language Models (LLMs), HE can be considered for performing inference on sensitive inputs without exposing the raw data to the model provider. This could be particularly useful when the LLM is hosted on external servers or cloud services, and users wish to keep their input data confidential. Regarding training, HE is likely to not be able to prevent memorization as the patterns in the plaintext that are usually memorized by LLMs should be preserved by the encryption step.

The application of Homomorphic Encryption to LLMs aims to address privacy concerns during the inference phase by ensuring that sensitive user inputs remain encrypted throughout the computational process. Users

encrypt their queries using an HE scheme and send the encrypted data to the server hosting the LLM. The server performs computations directly on the encrypted data and returns an encrypted result, which the user can decrypt to obtain the plaintext output. While this approach theoretically provides strong privacy guarantees, several significant challenges hinder its practical implementation with LLMs.

First, **computational overhead and latency** are substantial. HE schemes, especially those capable of supporting the complex operations required by deep neural networks, introduce significant computational overhead. Operations on encrypted data are significantly slower than their plaintext counterparts, often by several orders of magnitude. This performance penalty makes real-time inference with LLMs impractical using current HE technologies. Second, **model complexity and compatibility** pose challenges. LLMs typically consist of billions of parameters and involve complex architectures with non-linear activation functions, such as GELU or Softmax, and operations like layer normalization and attention mechanisms. Many of these operations are not directly compatible with standard HE schemes, which primarily support arithmetic operations (addition and multiplication). Approximating these non-linear functions with polynomial functions compatible with HE can lead to a loss in model accuracy and additional computational complexity. Additionally, **memory and storage requirements** increase significantly. Encrypted data under HE schemes can be much larger than plaintext data due to encryption overhead. This ciphertext expansion results in higher memory and storage requirements on both client and server sides, making it challenging to handle large encrypted inputs and model parameters. Finally, there is **limited support for training**. While HE can, in theory, be applied to both training and inference, the vast computational demands make training LLMs under HE impractical with current technology. Consequently, most considerations of HE in the context of LLMs focus solely on the inference phase.

**Current Research and Limitations**   Research into applying Homomorphic Encryption to neural networks has shown some success with smaller models and simpler tasks. For instance, Gilad-Bachrach et al. (2016) introduced CryptoNets, demonstrating the feasibility of performing inference on encrypted data with neural networks for basic classification tasks like recognizing handwritten digits (MNIST dataset). Similarly, Al Badawi et al. (2021) proposed optimizations for running convolutional neural networks under HE but were limited to relatively small models and datasets. Extending these approaches to LLMs presents additional challenges.

Recent efforts have attempted to apply HE to Transformer models and LLMs. Hao et al. (2022) proposed *Iron*, a method that modifies Transformers to be more compatible with HE by approximating non-linear operations with polynomial functions, aiming to reduce computational complexity while preserving accuracy. Pang et al. (2024) introduced *BOLT*, which provides privacy-preserving and efficient inference for Transformers by reformulating attention computation to be more HE-friendly. Similarly, Chen et al. (2022a) presented *THE-X*, a framework that redesigns critical components of the Transformer architecture to suit HE constraints, optimizing encoding schemes to minimize performance degradation. Zheng et al. (2023) developed *Primer*, focusing on fast private Transformer inference on encrypted data by introducing optimizations like efficient polynomial approximations and streamlined encrypted computations.

Recently, Hou et al. (2023) introduced *CipherGPT*, a framework designed for secure two-party GPT inference that addresses privacy concerns in LLMs. CipherGPT enables a client (holding the input data) and a server (hosting the model) to collaboratively perform GPT inference without revealing the client's input or the server's model parameters to each other. This is achieved through the use of cryptographic protocols such as secure multiparty computation and homomorphic encryption, optimized specifically for GPT models.

CipherGPT introduces several technical innovations to improve the efficiency and practicality of secure inference with large models like GPT. These include a customized secure matrix multiplication protocol using subfield vector oblivious linear evaluation (sVOLE), an efficient method for computing the GELU activation function using spline-based approximations, and protocols for secure top-$K$ selection and sampling required for the text generation process. By addressing the computational and communication overheads typically associated with homomorphic encryption, CipherGPT demonstrates the feasibility of performing privacy-preserving inference with large-scale language models. However, despite these advancements, CipherGPT still faces challenges related to computational overhead and scalability. The secure protocols, although optimized, introduce latency and resource consumption that may hinder real-time applications, especially

with larger models. Additionally, the approximations used for non-linear functions may affect the model's performance and the quality of the generated text. Implementing CipherGPT also requires specialized expertise in cryptography and deep learning, which can be a barrier to widespread adoption.

While these works represent significant progress, they are often constrained by computational overhead and are applied to smaller models or simplified versions of Transformers. One major challenge is **approximating non-linear functions**. LLMs rely heavily on non-linear operations not directly supported by HE schemes. Researchers have explored using polynomial approximations of these functions, but such approximations can degrade model performance. For example, approximating the Softmax function used in attention mechanisms with a polynomial can lead to less accurate attention scores, affecting the overall quality of the generated text.

Another issue is the pronounced **performance trade-off** between privacy and utility. With LLMs, due to their size and complexity, this trade-off is more significant. While HE provides strong privacy by keeping data encrypted, the resulting computational costs and potential loss in accuracy make it less feasible for practical applications involving LLMs.

Furthermore, there is a **lack of practical implementations** for full-scale LLMs. Although recent works like Rho et al. (2024) have provided implementations for encoder-only models, they still face scalability issues when applied to large models. Li et al. (2023a) introduced *MPCFormer*, which enables private Transformer inference using Secure Multiparty Computation (MPC) instead of HE, optimizing the Transformer architecture to be compatible with MPC protocols. While MPCFormer shows improved performance over traditional HE methods, it still encounters challenges with communication overhead and scalability when applied to large-scale LLMs.

While Homomorphic Encryption presents an attractive theoretical solution for preserving privacy during inference with Large Language Models, current technological limitations render it impractical for real-world applications involving LLMs. The substantial computational overhead, memory requirements, and challenges in handling complex non-linear operations make HE unsuitable for deployment with current models. Alternative privacy-preserving techniques, such as Secure Multiparty Computation (SMC) and Trusted Execution Environments (TEEs), face similar challenges when applied to LLMs. Differential Privacy (as discussed in §5.2) and Federated Learning (§5.1) currently offer more practical approaches for preserving privacy in the context of LLMs, balancing efficiency with privacy guarantees. Ongoing research in cryptographic techniques and hardware acceleration may, in the future, alleviate some of the computational burdens associated with Homomorphic Encryption. Until such advancements are realized, HE remains a topic of interest in theoretical discussions but is not a practical solution for privacy preservation in Large Language Models.

# 6   Tools & Frameworks

After discussing the approaches one may take to tackle the problem of privacy-preserving LLMs, we briefly see what frameworks and libraries are already in place for researchers and practitioners interested in the topic.

**Data Anonymization.**   Data anonymization has been extensively used to tackle privacy problems in ML for years, which means several frameworks and libraries can apply anonymization to datasets. The most prominent examples are:

- **ARX Data Anonymization Tool**: ARX is a comprehensive open-source tool that provides scalability and usability for data anonymization. It supports a variety of anonymization techniques, data quality analysis methods, re-identification risk assessments, and privacy models like k-anonymity, l-diversity, t-closeness, and differential privacy. It's updated regularly and is available in Java.

- **Open Data Anonymizer**: Aimed at Python users, this library focuses on data anonymization and masking for data science tasks. It integrates well with Python's data-handling libraries, such as pandas, making it a suitable choice for projects involving data frames and machine learning.

- **Microsoft Presidio**: Presidio offers detailed anonymization capabilities for text, including the identification of personally identifiable information (PII) and its subsequent anonymization or de-identification. It features two main modules: the Presidio Analyzer for detecting PII in text and the Presidio Anonymizer for removing detected PII using various operators like redaction, replacement, hashing, or encryption.

- **Spark NLP for Healthcare** by John Snow Labs: Specifically designed for the healthcare sector, Spark NLP for Healthcare provides state-of-the-art accuracy for extracting, classifying, and structuring clinical and biomedical text data. It's optimized for Databricks, offering solutions like OCR for document processing and NLP models for detecting and obfuscating protected health information (PHI) in unstructured text

**Differential Privacy.**   We have seen that Differential Privacy has been considered a gold standard for privacy protection in machine learning for years, with several techniques to enforce it and big players using it in very used applications. Naturally, this means that there are many frameworks to implement, some of which are open-source. Most of these frameworks, like most of the research in the field, focus on differentially private training. Here are presented the most prominent ones, for the tabular version of this list, refer to 6:

- **TensorFlow Privacy**: Similar to Opacus but for TensorFlow, it is a Python library that includes implementations of TensorFlow optimizers for training machine learning models with DP. The library has tutorials and analysis tools for computing the privacy guarantees provided.

- **PyVacy**: PyTorch translation of TensorFlow Privacy.

- **OpenDP project**: incubated by Harvard University's Privacy Tools and Privacy Insights projects, the OpenDP project (OpenDP, 2020) aims to make DP tools available to everyone freely. It includes several components: OpenDP Library, a core component consisting of a collection of algorithms for generating differentially private statistical releases, developed in Rust with Python bindings; SmartNoise, designed for creating DP reports, dashboards, synopses, and synthetic data releases. It supports SQL queries over Spark and popular database engines and includes a variety of synthesizers for generating privacy-preserving synthetic data; DP Creator, a web-based application, is focused on enabling the calculation of DP statistics for statistical queries intended for public release.

- **Diffprivlib**: IBM's general-purpose library (Holohan et al., 2019), stands out for its focus on providing a wide range of DP tools for ML and data analysis tasks, supporting experimentation and development.

- **Google DP**: Provides a broad set of DP tools in Java, Go and C++. (1) Privacy on Beam: an end-to-end DP framework built on top of Apache Beam. (2) Three "DP building block" libraries in C++, Go, and Java implement basic noise addition primitives and DP aggregations. Privacy on Beam is implemented using these libraries. (3) A stochastic tester is used to help catch regressions that could make the DP no longer hold. (4) A DP accounting library is used to track the privacy budget. (5) A command line interface for running differentially private SQL queries with ZetaSQL. DP Auditorium is a library for auditing differential privacy guarantees.

- **dp-few-shot-generation**: Microsoft's contribution to the DP landscape (Tang et al., 2024) is the only inference-DP framework you will find in this list. As explained in §5.2, DP in-context-learning is a relatively new idea, and this is the only public library to do it, to the best of our knowledge.

- **EKTELO**: made by Zhang et al. (2018), it stands out for its focus on a flexible, extensible framework for privacy-preserving data analysis, enabling complex queries and analytics while ensuring differential privacy. As described on the GitHub page, they have 2 main objectives: Isolate private interactions with data in a compact and secure kernel and modularize privacy-related algorithms into operators, which promote code reuse and assist in keeping the kernel compact and secure.

- **PyTorch Opacus**: made by Yousefpour et al. (2021), it enables training PyTorch models with differential privacy. It supports training with minimal code changes required on the client, has little impact on training performance, and allows the client to track online the privacy budget expended at any given moment. This is all taken care of by an abstraction called PrivacyEngine.

- **private-transformers**: Provides a privacy engine built off Opacus but rewritten specifically to facilitate integration with HuggingFace transformers.

- **dp-transformers**: It is a toolkit that provides a simplified integration of transformers training with DP. It also provides some examples and tips for DP training.

- **Chorus**: Chorus (Johnson et al., 2018) is unique for its DBMS-independent architecture, focusing on scalable, DP statistical queries through a cooperative query processing system. This library was actually deployed at Uber.

- **autodp**: Automates the process of calculating the privacy guarantees for complex algorithms and supports several standard mechanisms. Users can define their own privacy mechanisms or modify existing ones.

**Transformers.** The main focus of this work is LLMs, which are based on the transformers architecture (Vaswani et al., 2017). Most transformers nowadays are implemented with the Transformers Library by HuggingFace. This library quickly became a cornerstone in AI because it provides most state-of-the-art models already pre-trained and ready to be used in a plug-and-play fashion, allowing researchers to fast and efficient prototyping. It provides tools for pre-training and fine-tuning like SFT for supervised learning and TRL for reinforcement learning. The main reason for the success of this library is its versatility: there are also integrations for distributed training with Accelerate and DeepSpeed and it even supports Parameter-Efficient Fine-Tuning with an ad-hoc library called PEFT, which are all fundamental elements when you use big models like modern LLMs. There has been some work to try and integrate DP libraries and Transformers, and in particular there are two public implementations of Opacus with Transformers: private-transformers and dp-transformers which both provide implementations of some recent LLMs and other models with the integration of Opacus for Differential Privacy.

**Federated Learning.** Federated learning is standard when the task requires handling sensitive data, as it makes it possible to keep the data on-site without sending them to a central server and, therefore, gives a (false §3.3) sense of security. There are several frameworks for federated learning. Let's quickly see the most famous ones:

Table 1: Comparison of Differential Privacy Frameworks. Categorized here based on the components they offer: Optimizer means that the framework offers some optimizer that can be used in combination with any model to train it with DP; Mechanism means that the framework provides noise injection mechanisms like the Exponential Mechanism; Models means that the framework provides some DP models ready to be used; In-Context-Learning means that the framework offers the possibility to perform ICL with your model.

| Framework | Optimizer | Mechanisms | Models | ICL | Language |
|---|---|---|---|---|---|
| Tensorflow Privacy | ✓ | ✗ | ✗ | ✗ | Python |
| PyVacy | ✓ | ✗ | ✗ | ✗ | Python |
| OpenDP | ✗ | ✓ | ✗ | ✗ | Rust [3] |
| Diffprivlib (IBM) | ✗ | ✓ | ✓ | ✗ | Python |
| Google DP | ✗ | ✓ | ✗ | ✗ | C++,Java,Go [4] |
| dp-few-shot-generation | ✗ | ✗ | ✗ | ✓ | Python |
| EKTELO | ✗ | ✓ | ✗ | ✗ | Python |
| Opacus | ✓[5] | ✗ | ✗ | ✗ | Python |
| private-transformers | ✓[6] | ✗ | ✗ | ✗ | Python |
| dp-transformers | ✓[7] | ✗ | ✗ | ✗ | Python |
| Chorus | ✗ | ✓ | ✗ | ✗ | Scala |
| autodp | ✗ | ✓ | ✗ | ✗ | Python |

- **Flower**: Flower (Beutel et al., 2022) is an open-source,framework-agnostic FL library designed to enable the building of distributed systems with any ML framework. It focuses on usability, scalability, and compatibility, aiming to make FL accessible to researchers and practitioners working with different tools. Flower provides a flexible and easy-to-use solution since it allows for experimentation across different frameworks, making it ideal for researchers and developers exploring federated learning in diverse environments.

- **pfl**: Apple's pfl library is a federated learning library specifically devised for quick research simulations. It also provides tight integration with local and central DP mechanisms, allowing quick prototyping of differentially private federated networks.

- **TensorFlow Federated (TFF)**: TensorFlow Federated is an open-source framework for machine learning and other computations on decentralized data. TFF enables developers to simulate FL in their machine learning models using TensorFlow. It's designed for flexibility and ease of use when experimenting with FL algorithms. TFF is a natural choice for those who use TensorFlow since it provides seamless integration with existing TensorFlow models and extensive documentation and tutorials from Google.

- **PySyft**: PySyft is a Python library for secure and private deep learning that extends PyTorch. It supports FL, DP, and SMPC, aiming to make privacy-preserving machine learning accessible. PySyft is part of the OpenMined initiative, which fosters an open-source community focusing on privacy-preserving AI.

- **FATE (Federated AI Technology Enabler)**: Webank's FATE is an open-source project intended to provide a secure computing framework to support the federated AI ecosystem. It's designed for financial institutions, enabling them to build machine learning models without sharing their data, thus complying with regulations. FATE is particularly suited for organizations in the financial sector looking to engage in federated learning. It provides a robust framework that emphasizes security and privacy, making it ideal for use cases that require strict data protection measures.

---

[3]Bindings for Python
[4]Bindings for Python
[5]Opacus provides a privacy engine that encapsulates the model and the optimizer
[6]Built off Opacus, which provides a privacy engine that encapsulates the model and the optimizer
[7]Built off Opacus, which provides a privacy engine that encapsulates the model and the optimizer

- **PaddleFL**: Baidu's PaddleFL is an open-source federated learning framework based on PaddlePaddle, Baidu's ML platform. It supports various FL strategies and provides a flexible definition of federated learning tasks. It's designed with industrial applications in mind, offering tools for model training and inference in a privacy-preserving manner. This is the go-to for those who are already using PaddlePaddle.

## 7 Limitations and Future Directions

In this section, we provide a discussion on current limitations and future directions for preserving privacy in large language models, analyzing all the different approaches to preserving privacy proposed in this survey.

**Preserving Privacy in Data.** Incorporating privacy directly within data before it is input into a large language model offers a practical alternative to more complex, model-based privacy-preserving techniques. For example, the large-scale pretraining of prominent models like LLaMA3 and OLMo leverages data anonymization techniques, applying privacy measures at the data level before it enters the LLM. However, given the vast scale of pretraining data, these projects have largely relied on basic regular expressions to filter out PII and streamline the preprocessing (Dubey et al., 2024; Soldaini et al., 2024). While efficient, this approach is quite limited as it lacks comprehensive privacy protection and does not offer formal privacy guarantees such as those provided by DP. As Subramani et al. (2023) notes, "Running an out-of-the-box tool such as Presidio is the bare minimum that should be applied to all new and existing corpora". To support progress in this area, sections 4.1 and 6 of this survey present practical insights for researchers and practitioners aiming to build the next generation of privacy-preserving LLMs. While current models primarily rely on regex-based filtering for pretraining data, several approaches discussed in section 4.1, including metric Differential Privacy methods, can be scaled and applied to enhance privacy preservation in LLMs. We believe these methods hold significant untapped potential for advancing privacy-preserving techniques in data handling for LLMs and that we are just beginning to explore their full possibilities.

**Preserving Privacy in Model.** Model-based approaches based on DP offer mathematically-proven privacy guarantees within defined privacy parameters ($\epsilon, \delta$), but they come with significant computational costs and require substantial, often complex, modifications to models and training procedures. Of all the approaches discussed in section 5, DP is considered the gold standard for privacy-preserving machine learning, as it is the only method that provides a formal mathematical privacy guarantee. However, this assurance typically comes at the expense of some utility loss, especially when DP is applied throughout the entire learning pipeline (Tramèr et al., 2024b; Feng & Tramèr, 2025). DP's applicability spans data preparation, training, fine-tuning, and inference. In contrast, the privacy guarantee provided by other approaches like Machine Unlearning (§5.4) is only empirical but necessary to implement the "right to be forgotten" of privacy laws like GDPR (Zhang et al., 2024b). Future research could focus on building a rigorous mathematical basis for Machine Unlearning, developing techniques to seamlessly combine it with DP, and implementing accessible interfaces that allow users to specify data they wish to be erased from the model.

Despite the extensive application scope of Differential Privacy, it assumes a structure of private data where every bit of language can potentially contain secrets that are not meant to be shared indiscriminately. Differential Privacy cannot directly preserve secrets – information that can be shared only to the correct group of people (Brown et al., 2022b). The capability of LLMs to keep secrets is still little explored, with the development of the first benchmarks to assess the ability of such models to disclose private information only to users who should be allowed to access them (Mireshghallah et al., 2024; Priyanshu et al., 2023; Wang et al., 2024a). Future research should focus on advancing DP to selectively protect specific language elements, enhancing utility by applying privacy constraints only where necessary. As LLMs are increasingly integrated as assistants in applications, functioning as agents among groups of users who wish to share certain information while keeping other details private, addressing this issue will become more critical.

Looking ahead, applying DP to the training, tuning, or conditioning of larger, more capable models presents both a challenge and an opportunity. Although most current studies focus on smaller models, the advancement of quantization and Parameter Efficient Fine-Tuning (PEFT) techniques promises to make DP viable in larger models as well. These advancements reduce computational overhead and have facilitated DP-tuning in smaller models, such as RoBERTa and GPT-2 via LoRA, which could soon extend to modern models like Llama-3 and Mistral with negligible impact on computational resources (§5.1.3). We believe that the next step in this field will be the development of privacy-preserving LLMs that leverage these techniques, paired with state-of-the-art models. With the existing building blocks (techniques and models) in place, this development hinges primarily on computational resources for private fine-tuning. Our ideal vision includes models pre-trained entirely with DP, which would significantly mitigate privacy risks from the outset. However, we

recognize that fully DP-compliant pre-training remains challenging, as it demands significant computational resources and can reduce model effectiveness. A practical compromise could involve pre-training the model using data anonymization techniques, which are less computationally intensive (as discussed in Section 4.1) while reserving DP methods for the subsequent stages of instruction tuning and alignment. In the short term, we believe this approach offers a reasonable balance.

Federated Learning (§5.3) offers another promising avenue for enhancing privacy in LLMs, albeit with limitations. Although FL with Central DP is feasible, constraints on device capacity mean that edge devices, such as mobile phones, often lack the processing power to maintain a locally stored LLM. This limitation is slowly diminishing, as illustrated by devices capable of running small-scale models like Gemini-Nano locally (Team et al., 2024). With the ongoing miniaturization and optimization of LLMs, we anticipate that FL-driven, DP-compliant training on local devices will become increasingly feasible. Nevertheless, implementing a local DP model remains challenging due to the high dimensionality of the data and model size; PEFT techniques provide some relief by minimizing parameter sharing, but more research is needed to refine utility and privacy outcomes within this framework.

Inference under DP constraints has shown initial success, suggesting that LLMs may offer privacy-preserving inference without requiring task-specific fine-tuning (§5.2). While current research in DP-constrained inference is limited, early results are promising, especially as models become more compact without sacrificing capability. In practical terms, this progress allows for privacy-preserving inference in cases where offsite models are preferable or where specific fine-tuning is unnecessary. Future work could explore larger models, potentially introducing DP to inference on locally stored models, where the option of performing inference without any data sharing would be highly advantageous.

Despite the appeal of Homomorphic Encryption (HE) for privacy-preserving inference with LLMs, significant challenges limit its practicality. As outlined in §5.5, HE introduces considerable computational overhead and latency, making real-time inference with large models unfeasible. Issues with approximating non-linear functions can degrade performance, while ciphertext expansion adds memory demands that complicate deployment on limited-resource systems. Recent advancements like CipherGPT (Hou et al., 2023) have optimized HE for LLMs, yet scalability, performance trade-offs, and complexity remain concerns. Future research should aim to reduce overhead through more efficient protocols, improve non-linear approximations, and explore hybrid methods that combine HE with other privacy techniques.

In conclusion, our findings reveal that while privacy-preserving techniques for ML are advancing, efficiency remains a central focus, as existing methods often involve trade-offs in training speed and model performance. Recent innovations have brought privacy-preserving approaches closer to practical deployment in public-facing models, reducing these trade-offs and making DP increasingly accessible. We are confident that this trend will persist, paving the way for the widespread use of privacy-preserving ML as a practical alternative in the near future.

## 8 Conclusion

In this work, we conducted an extensive survey on preserving privacy in Large Language Models (LLMs), meticulously analyzing the various dimensions of privacy concerns and the corresponding solutions.

Firstly, our investigation began by exploring the different types of privacy attacks (§3) that these models are susceptible to, including Training Data Extraction (§3.1), Membership Inference (§3.2), and Model Inversion (§3.3). Each type exposes significant vulnerabilities. Training Data Extraction attacks compromise privacy by enabling adversaries to reconstruct sensitive training data such as personal identifiers or confidential communications, which erodes trust in LLM applications. Membership Inference attacks reveal whether specific data was used in training, potentially disclosing sensitive personal details like medical conditions, which could lead to discrimination. Model Inversion further extends these risks by allowing attackers to deduce personal characteristics from model outputs, promoting unauthorized profiling and significant privacy breaches.

Secondly, we presented an extensive view of the state-of-the-art solutions for preserving privacy in large language models (§4-5). These solutions include approaches that inject privacy directly into the training

data, like anonymization techniques (§4) or approaches that act on the model (§5). While there are many possible approaches to the problem, no single solution fits all scenarios. We presented solutions for applying privacy mechanisms at several points in the learning pipeline: from data preparation (before training - §4), to private training (during training - §5.1), to private inference (§5.2-5.5), and finally after training (§5.4). We refer the reader to section 7 for a thorough discussion on current limitations and future directions for preserving privacy in large language models.

This work highlights the critical need to address privacy vulnerabilities in Large Language Models (LLMs) by systematically analyzing various privacy attacks and their profound risks to data confidentiality and user trust. We presented current state-of-the-art solutions such as data anonymization, Differential Privacy, and Machine Unlearning, emphasizing the importance of continuously advancing and integrating these techniques. As LLMs become integral to sensitive domains like healthcare and finance, developing robust privacy safeguards is essential. Future research must enhance these methods, improve the privacy-utility trade-off, and ensure that larger models adhere to stringent privacy standards, ultimately fostering the creation of secure and reliable LLM technologies that uphold user privacy and trust.

## Acknowledgements

The authors gratefully acknowledge the support of the EU-funded project DataTools4Heart (grant No. 101057849), which provided crucial funding and resources for this research. This work has also been carried out in the scope of Project ECS 0000024 Rome Technopole – CUP B83C22002820006, NRP Mission 4 Component 2 Investment 1.5, funded by the European Union – NextGenerationEU.

We also extend our sincere thanks to the reviewers and the action editor of TMLR for their invaluable feedback and constructive comments. Their insights have significantly enhanced the quality and clarity of this paper.

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

# A Going Further

As shown throughout this work, the topics discussed here have been subject to extensive research in recent years. In our work, we have shown how privacy-preserving techniques like anonymization or Differential Privacy have been used to build private Language Models. For those readers interested in deepening their knowledge on a single subject, we present here a list of surveys specific to each single topic:

- **Data Leakage**: Jegorova et al. (2021) presents a comprehensive survey on privacy leakage at inference time. They cover involuntary leakage and adversarial leakage and also touch on the available defence mechanisms. The main difference with our work is that we are focused on LLMs, which is also reflected in the techniques shown as solutions, which are much more recent.

- **Attacks**: There are several surveys investigating attacks. Rigaki & García (2020) investigate the privacy vulnerabilities of machine learning in general, while some other works are more focused on a specific topic. For example, Enthoven & Al-Ars (2021); Zhang et al. (2022b); Liu et al. (2022) focus on attacks against federated learning models or Derner et al. (2023); Li et al. (2023b) focus on security risks of LLMs, including possible attacks and mitigations. While some of these works have partial overlap with our section §3, we mostly focus our investigation on *privacy attacks* against LLMs and possible solutions.

- **Privacy Preserving Deep Learning**: Literature yields many works on the topic of privacy-preserving deep learning. Some of them are generic, without focusing on any specific model. This is the case for Boulemtafes et al. (2020); Tanuwidjaja et al. (2020); Tariq et al. (2020); Liu et al. (2021); Tanuwidjaja et al. (2019); Mirshghallah et al. (2020) which all touch on privacy issues in deep learning and possible solutions. Although these works are relevant, they primarily focus on deep learning in general and do not address the recent advancements brought by LLMs. Recently, Yao et al. (2024b) provided a systematic review on security and privacy in language models divided into three sections: good applications, bad applications, and security risks and solutions. This work is complementary to our study since it covers a broader range of risks (not just privacy risks) while allocating less space to solutions. Ultimately, works focused on privacy in NLP, such as Sousa & Kern (2023), concentrate on NLP and anonymization, while Guerra-Manzanares et al. (2023) targets medical applications without necessarily focusing on textual data.

- **Machine Unlearning**: There are several surveys on the topic of Machine Unlearning: Xu et al. (2023a); Nguyen et al. (2022); Shaik et al. (2024) are general, while Si et al. (2023); Liu et al. (2024a) only focus on LLMs and the differences in application.

- **Differential Privacy**: With the growing interest in Differential Privacy in recent years, numerous literature reviews have emerged. They all focus on different aspects of Differential Privacy: Prokhorenkov & Cao (2023) is focused on the privacy risks of Differential Privacy, how to define it and measure it. Several works investigated the intersection between Differential Privacy and deep learning like Baraheem & Yao (2022); Zhao et al. (2019b); Ponomareva et al. (2023); Demelius et al. (2023); Chen et al. (2024a) focus on generative models in particular; Klymenko et al. (2022) focus on DP in the context of NLP; Garrido et al. (2023) present a particularly interesting work as it focuses on the industry, the difficulties in the adoption and gives some suggestions to close the gap between industry and academia; Ouadrhiri & Abdelhadi (2022); Zhang et al. (2023a); Odera (2023) focus on the applications in federated learning.

- **Federated Learning**: Regarding federated learning, several surveys are available with different perspectives. Kairouz et al. (2021b) propose a general survey on federated learning, examining the state of the art and open problems. Hasan (2023); Ma et al. (2022); Truong et al. (2021) give an overview of the privacy and security issues, with Truong et al. (2021) also putting it in relation to GDPR. Lastly, Brauneck et al. (2023) gives an overview of FL in medical research.

- **Large Language Models**: Chang & Bergen (2024); Zhao et al. (2023); Minaee et al. (2024) present comprehensive surveys on large language models (LLMs), covering various aspects such as

their architecture, training techniques, applications, and the challenges they face. These surveys provide valuable insights into the current state of LLM research, highlight recent advancements, and discuss future directions for the development and deployment of these models.

- **Anonymization** For text data anonymization, the most important works are related to the healthcare domain, which is the case for Larbi et al. (2022); Zuo et al. (2021); Meystre et al. (2010).

# B   Summary Tables

In this section, we present summary tables for all the approaches discussed in the text. Table 2 and Table 3 summarize the anonymization techniques respectively presented in §4.1 and §4.1.2. Table 4 presents some information about the works that perform a fully private training, discussed in §5.1.1, with information about the model, the datasets used both for pre-training and fine-tuning and the optimization method. Similarly, in Table B, there is an overview of the works that perform private fine-tuning, discussed in §5.1.2. Table 6 summarizes the works found in the literature for private inference, which are discussed in §5.2. Lastly, Table 7 presents a brief summary of the Unlearning methods presented in §5.4.

**Data Anonymization**

Table 2: Classical methods for text anonymization

| Method | Summary | Reference |
|---|---|---|
| Modular | Modular system that has pre-processing, NER, CRR and Anonymization. Performances dependent on NER module. | Mamede et al. (2016) |
| Perturbation | Altering data while preserving statistical significance. Effectively anonymizes data but may compromise the accuracy. | Zuo et al. (2021) |
| DataSifterText | Sensitive information is masked and then imputed with BERT. Whitelist to mask sensitive terms, blacklist for meaningful terms to retain. Retains high data utility. | Zhou et al. (2022) |
| Transformer | Text analyzed to extract identifying features, which are then smoothed. Universal Transformer is used to generate text incorporating the features. Integrates modern model with classical approaches. Specific to Russian. | Romanov et al. (2019) |
| RLTA | Attention-based text representation learner extract latent embeddings and a deep RL-based privacy preserver manipulates them. Effectively balances privacy and utility, but is dependent on precise parameter tuning and has data-specific limitations. | Mosallanezhad et al. (2019) |
| LLMs | Use an LLM to impute previously masked tokens. They find NEs with NER and then mask them. Performances are significantly better when compared to a model trained on the masked data. | Vats et al. (2024) |
| HaS | Two local LLMs, the first one anonymizes the prompt before sending it to a black-box LLM, the second one de-anonymizes the output from the black-box LLM. They show a good privacy-utility tradeoff. | Chen et al. (2023b) |

**Data Anonymization with DP**

Table 3: Methods for text anonymization with DP

| Method | Summary | Reference |
|---|---|---|
| ADePT | Clipping and noising the embeddings before decoding to build a DP latent representation. DP proof is problematic. | Krishna et al. (2021) |
| DP-VAE | Apply isotropic Gaussian noise in the latent space. Good results for authorship anonymization. | Weggenmann et al. (2022) |
| DP-BART | Clipping and noising the embeddings, new clipping and iterative pruning allows for a smaller encoder. Performances over the baseline. | Igamberdiev & Habernal (2023) |
| BERT | Purkayastha Mechanism used to take advantage of the angular distance as a metric. Performances close to non-private and increased MIAs robustness. | Du et al. (2023b) |
| SANTEXT | Words mapped into embeddings, then compute distances and use them to sample replacement inversely proportional to the distance | Yue et al. (2021) |
| CUSTEXT | Replace every token: mapping function determines output space and sampling function chooses replacement tokens. Good privacy/utilty tradeoff. | Chen et al. (2023a) |
| RANTEXT | Random adjacency lists to increase embedding inversion attacks robustness. Used with black-box LLMs and a de-anonymization module for open-ended text generation, shows very strong performances | Tong et al. (2024) |
| biLSTM | New noise distribution specifically designed for DP. Epsilon calibration based on geometric properties of the word embedding space. Shows a favorable privacy/utility tradeoff. | Feyisetan et al. (2019b) |
| TEM | Truncated Exponential Mechanism to replace standard Exponential Mechanism, which is not suitable for metric DP as it does not consider the specific structure of the embedding space. Substantial improvements over baseline. | Carvalho et al. (2021) |
| RMM | Regularized Mhalanobis Metric used for text perturbation to add elliptical noise to ensure that words in sparse regions have a sufficient likelihood of replacement without sacrificing overall utility. | Xu et al. (2020) |
| Syntactic Approach | Include grammatical categories into privatization to preserve grammatical properties. Improves performances but slightly decreases privacy. | Arnold et al. (2023) |
| Paraphrase | Fine-tuned GPT2, to introduce noise, temperature is sampled based on the exponential mechanism. Superior performance with respect to word-level DP. | Mattern et al. (2022b) |

**Models pre-trained with DP**

Table 4: Models pre-trained with DP

| Model | Method | Datasets | Reference |
|---|---|---|---|
| BERT | DP-SGD | BERT dataset+MIMIC-III pre-training
i2b2-2010 fine-tuning | Hoory et al. (2021) |
| BERT | DP-SGD
SGD | BookCorpus+Wikipedia+Legal pre-training
Overruling dataset/CaseHOLD fine-tuning | Yin & Habernal (2022) |
| Feedforward NN | DP-SGD | Brown corpus pre-training
Reddit comments dataset fine-tuning | Kerrigan et al. (2020) |
| GPT2 | DP-SGD | Selected 15% of OpenWebText pre-trainig
Enron email dataset fine-tuning | Tramer & Boneh (2021) |

**Models fine-tuned with DP**

Table 5: Models fine-tuned with DP

| Model | Method | Dataset | PEFT | Reference |
|---|---|---|---|---|
| GPT2 | DP-SGD | Open Yelp Dataset | ✗ | Yue et al. (2023) |
| RoBERTa | DP-ADAM | MNLI,QQP,QNLI,SST-2 | ✗ | Li et al. (2022) |
| GPT2 | DP-ADAM | E2E dataset | ✓ | Li et al. (2022) |
| DialoGPT | DP-ADAM | Persona-Chat dataset | ✗ | Li et al. (2022) |
| GPT2 | DP-SGD | IMDb, Amazon reviews dataset | ✗ | Mattern et al. (2022a) |
| RoBERTA | DP-SGD | MNLI, SST-2, QQP, QNLI | ✓ | Yu et al. (2022) |
| GPT2 | DP-SGD | E2E, DART | ✓ | Yu et al. (2022) |
| BERT | ADAM | SST, QQP, TP-UK | ✓[a] | Li et al. (2023e) |
| T5 | ADAM | SST, QQP, TP-UK | ✓[b] | Li et al. (2023b) |
| GPT2 | DP-SGD, DP-PPO | Reddit TL;DR, IMDb | ✗ | Wu et al. (2024a) |

[a]The authors used prompt-tuning and prefix-tuning after obfuscating the embeddings
[b]The authors used prompt-tuning and prefix-tuning after obfuscating the embeddings

**DP-Inference methods**

Table 6: DP-Inference methods

| Model | Method | Dataset | Reference |
|---|---|---|---|
| GPT-3 | Report-Noisy-Max+GM
ESA
KSA | SST-2, Amazon,
AGNews, TREC,
PFL-DocVQA, SAMSum | Wu et al. (2024b) |
| GPT-3 | GM,
EM | AGNews, DBPedia, TREC
MIT-G, MIT-D | Tang et al. (2024) |
| GPT-3 | PromptPATE | SST-2, AGNews, TREC,
DBPedia | Duan et al. (2024a) |
| Llama-2,
GPT-3,
Vicuna | DP-OPT | SST-2, TREC, Mpqa,
Disaster | Hong et al. (2024) |
| T5, BERT,
GPT-2 | Split-and-Denoise | TweetEval Offensive, Hate Speech 18,
Healt Fact, Daily Dialogue and others | Mai et al. (2024) |
| GPT-4,
Vicuna | InferDPT | CNN/Daily Mail, Wikitext-103-v1,
ArXiv Dataset | Tong et al. (2024) |

**Machine Unlearning Methods**

Table 7: Machine Unlearning Methods

| Method | Summary | Reference |
|---|---|---|
| ROME | Allows to track factual predictions back to single neurons and edit them individually. Proven to be ineffective in some cases. | Meng et al. (2022) |
| MEMIT | With a similar approach with respect to ROME, scale the editing on multiple examples. The discussion is focused on factuality. | Meng et al. (2022) |
| DEPN | This method assumes that private information is encoded in specific neurons: their weights are set to zero to block information flow. | Wu et al. (2023) |
| PAE | Model editing to break the association between a user identity and its PII. The edit method is effective both in batch and in sequential editing. | Venditti et al. (2024) |
| SISA | Divide training data in shards, further divided in slices and then train instances of the model on each shard by incrementally using slices and saving parameters at every step. Requires limited re-training. | Bourtoule et al. (2019) |
| KU for privacy | Negate the loss function used in training to maximize the loss on the target sequence (to unlearn) | Jang et al. (2022) |
| Approximate UL in LLMs | Fine-tune a model on target sequences, then use it to lower the likelihood sequences predicted by this model. | Eldan & Russinovich (2024) |
| UL layers | Build Unlearning layers and train them with a selective student-teacher objective based on KL-divergence to maximize divergence from teacher model | Chen & Yang (2023) |
| DeMemorization | Take samples from pre-training dataset, divide in prefix and suffix, use a negative similarity metric to evaluate how different is the generated suffix from the real one. This is used as a reward signal to encourage the LLM to develop a paraphrasing policy | Kassem et al. (2023) |
| Knowledge Sanitization | Use LoRA to instruct the model with a sanitization phrase when a potentially harmful prompt is given | Ishibashi & Shimodaira (2023) |
| ICUL | Build a context where the datapoint you want to unlearn is included but with a flipped label | Pawelczyk et al. (2024) |

## C  Discussion on Federated Learning

The central model of Differential Privacy (CDP) enforces privacy at the participant level rather than the record level. Record level Differential Privacy corresponds to the classical definition (see §2.2 for more details) and states that the outcome of a differentially private computation should be the same (with probability $\epsilon$) regardless of whether a particular record is or is not in the dataset. As Naseri et al. (2020) explain, participant-level DP ensures (with probability $\epsilon$) that the aggregation function produces the same result regardless of whether a particular client participates in the training process.

Although this method is more secure than sending plain data, it requires a high level of trust in the central server and in the communication channel. Clients send unperturbed gradients, which can lead to training data reconstruction via gradient inversion (Zhang et al., 2022a), and even averaged gradients before perturbation can be exploited for reconstruction, as shown by Geiping et al. (2020), see §3.3 for more information. This means that an honest-but-curious (or semi-honest (Paverd & Martin, 2014)) server could potentially access the training data. Given these risks, Central Differential Privacy is not widely used. However, Naseri et al. (2020) demonstrates that CDP can reduce federated learning's vulnerability to backdoor and white-box Membership Inference Attacks (§3.2). For instance, on the CIFAR-10 dataset, they achieved a 23% reduction in the accuracy of MIAs without a significant loss in utility.

McMahan et al. (2018) introduce DP-FedAvg, a variation of the FedAVG algorithm by McMahan et al. (2016), incorporating the moments accountant (Abadi et al., 2016) to calculate the privacy budget used during training. Typically, FedAVG performs part of the training locally on a subset of clients selected based on specific criteria. The clients then send their updated gradients to the server, which averages them, updates the global model, and sends the updated model back to the clients. In DP-FedAvg, gradients are clipped locally but not obfuscated. The central server adds noise after receiving and aggregating the gradients to prevent inversion, aligning the algorithm with central DP since local clipping alone does not meet the Local DP criteria. This approach was tested by training an LSTM for the next token prediction on a large dataset of Reddit posts described by Al-Rfou et al. (2016), using various noise levels and clipping strategies. The model's performance was evaluated using AccuracyTop1, the probability that the predicted word is correct. The study demonstrated that, with sufficient data, it is possible to train a model with privacy guarantees without significantly sacrificing performance, provided the noise added is kept within reasonable limits.

Ramaswamy et al. (2020) apply DP-FedAVG to develop a private Next Word Prediction (NWP) model using an LSTM in a federated setting. Their goal is to train production-quality NWP models with DP-FedAvg in a real-world environment, specifically on a diverse fleet of mobile phones used for Gboard. Unlike DP-FedAVG's typical Poisson sampling, where each user is selected independently with a fixed probability for each round, this work employs fixed-size federated rounds. In this method, a fixed number of users is randomly sampled to participate in each round, ensuring a consistent number of participants and facilitating better management of computational and communication resources in a production environment like Gboard. The model's performance is evaluated on a user-generated dataset from Gboard, using the same private dataset for training, and is measured in terms of Top-1 Recall and Top-3 Recall. The results surpass the non-private baseline N-gram FST (Finite State Transducer). Privacy attacks based on the work of Carlini et al. (2019) are conducted to test the model's resistance to memorizing canaries. The findings show that the model does not memorize canaries unless they are repeated approximately 200 times or shared among roughly 16 users. The authors emphasize that the Differential Privacy approach relies on several assumptions—mostly sound but challenging to verify—highlighting the importance of empirical measurements like canary tests to validate privacy guarantees.

Currently, large language models (LLMs) represent the state of the art due to their superior power, though they are much larger than the small LSTMs seen in previous studies. Wang et al. (2024b) focus on using LLMs to enhance the privacy/utility tradeoff of local LMs in differentially private federated learning. Similar to Yu et al. (2024), which samples pre-training data based on its similarity to fine-tuning data, they propose a distribution matching algorithm to sample public data close to the private data distribution, significantly improving sample efficiency. They utilize DP-FTRL (Kairouz et al., 2021a), an online optimization algorithm combining FTL (follow the leader) and regularization, adding noise during local updates. The local LM is either a single-layer LSTM (Hochreiter & Schmidhuber, 1997) or Transformer (Vaswani et al., 2017) due to computational limits on edge devices, while the LLM used is LaMDA (Thoppilan et al., 2022). Their tests yielded several notable results. First, using a public tokenizer with a public vocabulary instead of a private one helps prevent privacy leakage. Second, pre-training local models on a public dataset (C4 dataset) for over a week of single GPU time improved next-token prediction accuracy (tested on the private StackOverflow dev set). They also built a distillation corpus by saving the top-k logits with k nonzero entries from the teacher LLM's next token predictions as a silver label dataset. Pre-training on this distillation corpus showed accuracy comparable to pre-training with the original dataset, even with just 1% distillation

coverage. Furthermore, their new distribution matching technique allows using only 0.08% of the pre-training dataset to achieve performance equal to 1% distillation coverage, greatly enhancing sample efficiency.

Xu et al. (2023b) present and analyze twenty Gboard LMs trained for Next Word Prediction using DP-FTRL, asserting that all future Gboard models will have DP guarantees. In their DP-FTRL implementation, model updates are clipped, and noise is added during global model updates, ensuring Central Differential Privacy. They also use adaptive clipping (Andrew et al., 2021), which adjusts the clip norm each round by privately estimating the model's norm at a targeted quantile. However, adaptive clipping showed no significant improvement and was less robust in experiments with large report goals, sometimes causing catastrophic failure or no progress in the first 1000 rounds. Despite this, adaptive clipping can reduce hyperparameter tuning when the privacy budget permits. Public pre-training on C4, as explained by Wang et al. (2024b), reduced the rounds needed to reach target utility by 1000 under the same noise multiplier, enhancing privacy guarantees. They also combined DP-FTRL with Secure Aggregation (Bonawitz et al., 2017), a cryptographic protocol ensuring the central server only accesses aggregated updates, not individual client updates. While SecAgg aids DP, it introduces significant computational slowdown and requires different system configurations for large report goals. The techniques for training language models with differential privacy in Gboard have produced models with stronger privacy guarantees than some established standards, such as those used by the US Census Bureau. In Europe, these differentially private neural network models outperform older N-gram models in utility. For English in the US and Portuguese in Brazil, the new models are on par with or slightly better than their non-private counterparts.

Basu et al. (2021) provide another example of differentially private training of LMs within a central model in a federated setting. They trained BERT, RoBERTa, DistillBERT, and ALBERT on tweets labelled to detect depression (Depression dataset) and sexual harassment (Sexual harassment dataset). Although the authors do not detail the training procedure, their results show that all models were trained with various epsilon values. Interestingly, the smaller models performed more favourably under these conditions.

### C.1 Federated Learning with Local DP

Initially formalized by Kasiviswanathan et al. (2008), local Differential Privacy (LDP) involves clipping and adding noise locally at the clients' level before sending gradients (see Figure 5.3). This method removes the need for a trusted central server and ensures safety even if an eavesdropper intercepts the gradients as they are noisy. It makes the process safer, but it comes with its own challenges. Though primarily used in networks handling image data, LDP approaches are worth mentioning for LLMs, as there is no theoretical barrier to their application. The main obstacle is the computational burden coming from the high dimensionality of text data, which may be mitigated in the future.

Shokri & Shmatikov (2015) introduce DSSGD (Distributed Selective Stochastic Gradient Descent) for private distributed training of MLP and CNN models on MNIST and SVHN datasets for digit classification. While not directly related to language modeling, the algorithm is broadly applicable to various network types. DSSGD builds on SSGD, a local Selective SGD that selectively applies gradients based on certain criteria to reduce the amount of training data information used for updates, thereby enhancing privacy, though not guaranteeing it. In DSSGD, each client trains a local model using SSGD's selection mechanism, and then shares selected parameters with a central server. The server aggregates these parameters to update the global model and redistributes it to clients for further local updates. This approach, however, has potential privacy risks from gradient selection and sharing. To address these, the authors use the Sparse Vector Technique (Dwork et al., 2006a; Hardt & Rothblum, 2010) to randomly select gradients exceeding a threshold and add noise using the Laplacian mechanism. Their experiments show that models trained with DSSGD achieve accuracy close to non-private methods. Specifically, on the MNIST dataset, the accuracy is almost indistinguishable from non-private training. Additionally, even with only 1% of parameters shared, the collaborative model significantly outperforms single local models.

Youn et al. (2023) introduce RQM (Randomized Quantization Mechanism), a novel approach to enforce DP in federated settings with improved communication efficiency, used alongside Distributed SGD on a CNN for image classification on EMNIST. RQM enhances DP-SGD by eliminating the need for explicit noise addition; instead, RQM is performed after gradient clipping. In RQM, gradient updates are encoded in quantization

levels, which are then randomly subsampled. A randomized rounding procedure maps continuous gradient values to the nearest subsampled quantization level. RQM falls under Renyi Differential Privacy (Mironov, 2017), a relaxation of the original differential privacy definition, placing the method under Local DP since RQM is performed on the clients. Tests show that RQM outperforms the Poisson Binomial Mechanism (Chen et al., 2022b) (a traditional noise injection method) and noise-free clipped SGD, achieving better accuracy with stronger privacy guarantees.

Bebensee (2019) provide a comprehensive overview of Local Differential Privacy (LDP) and its notable implementations, highlighting the increased variance and noise compared to the central model of DP. In LDP, perturbations are performed locally, leading to higher total variance, which directly depends on the number of participants. The lower error bound in the local setting is dependent on $n$ (the number of participants), necessitating a very high number of participants to inject sufficient noise. Additionally, since each client adds its own noise, the total noise is much higher, requiring significantly more data for efficient learning or a higher $\epsilon$.

LDP is widely used in the industry to collect and aggregate telemetry data. While these approaches are not directly relevant to the research presented here, they demonstrate how high-profile companies use DP in daily operations. Google (Erlingsson et al., 2014), Microsoft (Ding et al., 2017), and Apple (Team, 2017) have developed their own LDP algorithms to collect and aggregate usage statistics for applications like Chrome and the Apple keyboard. This also goes to show once again how local DP is widely used, but for low dimensionality data (telemetry data are usually counters), while for high dimensionality data like text there is still no sound implementation.

