# OpenReview forum: "Preserving Privacy in Large Language Models: A Survey on Current Threats and Solutions"
_TMLR — Accepted by TMLR_

### Review · Reviewer_Fx2K · 2024-09-19

**Summary Of Contributions:**

This paper is a comprehensive survey on the existing literature of privacy threats and potential solutions that exist in Large Language Models (LLMs). The authors start by introducing the threats to personal identifiable information in LLM training data and how it can be exposed via. data extraction, membership inference, and model inversion attacks. Then, the authors review the existing solutions in the field, and find they fall under two categories: directly applying privacy to the training data itself, or during the training methodology. Finally, the paper lists the many existing tools/frameworks and concludes that differential privacy (DP) is the benchmark/standard for privacy preserving in machine learning. The authors suggest that more work should be done applying DP to large scale start of the art LLMs.

**Audience:**

Yes

**Claims And Evidence:**

Yes

**Requested Changes:**

- **(critical):** based on the large amounts of DP-related works in this space, are there any high-level insights or directions on how future research surrounding DP and LLMs can be practically implemented?
- **(critical):** decrease amount of time/space spent on non-LLM models, in particular sections 3.2.1, 3.3.1, 5.3. Expand upon the works that deal with LLMs (like GPT-2) and use earlier works purely as guiding intuition.
- (strengthen): Including paragraph headers in addition to sub-subsections. Some sub-sub-sections like 4.1.2 are dense collections of paragraphs without any pointers to the topic of each paragraph, which makes identifying specific points and insights harder for readers.
- (strengthen): Since this survey is dedicated to large language models, sections 3.3, 5.3 seem a bit out of place in the context of its previous sections. As indicated in 5.3, federated learning is still in its infancy with large language models, which leads the survey to focus largely on transformers and BERT-like models. This detracts from the focus on the scale of LLMs. It might be beneficial to pull out these subsections into it’s own section in the end (or in the appendix), and mention that while distributed learning has not been widely studied with LLMs yet, this space is likely to face similar privacy concerns.

**Strengths And Weaknesses:**

**Strengths**

- Comprehensive detail of existing works in both the privacy-preserving and privacy-threatening field in LLMs.
- Clear taxonomy and useful reference of existing tools and what they are used for (section 6).

**Weaknesses**

- The paper makes a distinction in section 2 that models such as BERT are not large enough to be considered LLMs, which are the focus of the survey paper. Yet the large portion of the mentioned literature are based on experiments around BERT-like masked language models with downstream classification tasks.
- The paper claims that the main takeaway is that DP is the gold-standard for privacy protection due to its mathematical guarantee for privacy. However, the presentation of existing literature does not offer insight into how this gold-standard can be meaningfully met in the context of LLMs outside of a case by case approach given the many but scattered work of DP in language models.

---

> ### Author Response · Authors · 2024-11-14
> **Answer Reviewer Fx2K 1/2**
>
> We thank the reviewer for their time and thoughtful feedback on this survey. We are pleased that our work has been recognized for its comprehensive detailing of existing research in both privacy-preserving and privacy-threatening domains, along with its clear taxonomy and useful reference to existing tools.
>
> Thanks to the reviewers’ insights, we have significantly improved our survey. We invite the reviewer to read our responses to the specific points raised in this section, as well as the general response summarizing all the changes made to enhance the paper.
>
>
> ## Weaknesses and Requested Changes
>
> > The paper makes a distinction in section 2 that models such as BERT are not large enough to be considered LLMs, which are the focus of the survey paper. Yet the large portion of the mentioned literature are based on experiments around BERT-like masked language models with downstream classification tasks.
>
> The paper does not intend to imply that models like BERT are not large enough to be considered LLMs, nor does it explicitly state this in any part. We apologize for any misunderstanding on this point. To clarify, we have added a definition of LLM at the beginning of `Section 2`, specifying that for the purposes of this survey, we consider LLMs primarily as autoregressive generative models rather than defined by their size.
>
> While our survey focuses on generative LLMs, we include certain findings related to models like BERT when these insights on privacy threats and solutions are still relevant and applicable to larger generative models. This approach allows us to highlight research that, while originating with models like BERT, may offer valuable, generalizable contributions to the broader field of privacy-preserving LLMs. We added to these approaches an explaination on why they are relevant for generative LLMs.
>
> > - The presentation of existing literature does not offer insight into how this gold-standard can be meaningfully met in the context of LLMs outside of a case by case approach given the many but scattered work of DP in language models.
> > - Based on the large amounts of DP-related works in this space, are there any high-level insights or directions on how future research surrounding DP and LLMs can be practically implemented?
>
> We thank the reviewer for raising this. To adress this point we added a new section before the conclusion (`Section 7 - Limitations and Future Directions`) that that thoroughly discusses current limitations and future directions for preserving privacy in large language models. This new section provides high-level insights into various privacy-preserving techniques, including DP, and explores practical steps that future research could take to implement these methods effectively within LLMs. We hope this addition offers valuable guidance for advancing privacy protections in LLM development and deployment.
>
> > -  Decrease the amount of time/space spent on non-LLM models, in particular sections 3.2.1, 3.3.1, 5.3. Expand upon the works that deal with LLMs (like GPT-2) and use earlier works purely as guiding intuition.
>
> We have revised the specified sections by condensing discussions of non-LLM methods into a more general overview and expanding our focus on methods specifically involving LLMs.
> Specifically:
> - We removed the `Section 3.2.1 MIA with Shadow Models` since most of the literature agrees on the impossibility of training shadow models for LLMs. There are positive examples of that with smaller models (Hisamoto et al., 2020) and on GPT-2 models (Carlini et al., 2022): however, since they are isolated cases we removed the subsection, shortening its content and merging it with the general information at the beginning of `Section 3.2`.
> - To expand on the LLMs literature regarding MIAs, we also add some discussion on threshold-base MIAs at the end of the corresponding subsection (previously `Section 3.2.2`, now `3.2.1 - MIA with Thresholds`).
> - Similarly, we removed from `Section 3.3.1` the citations to non Transformer-based LMs and we have includeed them in the general discussion at the beginning of `Section 3.3`, while expanding the discussion on Inversion of LLMs in `Section 3.3.1`.
> - In `Section 5.3` we refactored the discussion on Federated Learning by keeping works most relevant to autoregressive models in the main paper and moving to the appendix works less focused on LLMs. We still believe that Federated Learning remains pertinent for future developments and thus should be included in the discussion, even though, at the moment of writing, most of the models proposed are not particularly large due to the computational complexity intrinsic to this approach.

---

> ### Author Response · Authors · 2024-11-14
> **Answer Reviewer Fx2K 2/2**
>
> > - Including paragraph headers in addition to sub-subsections. Some sub-sub-sections like 4.1.2 are dense collections of paragraphs without any pointers to the topic of each paragraph, which makes identifying specific points and insights harder for readers.
>
> We thank the reviwer for the suggestion. We have added topic paragraphs within all the sections in `4.1` to improve readability and make it easier for readers to locate specific points and insights.
>
> > -  Since this survey is dedicated to large language models, sections 3.3, 5.3 seem a bit out of place in the context of its previous sections. As indicated in 5.3, federated learning is still in its infancy with large language models, which leads the survey to focus largely on transformers and BERT-like models. This detracts from the focus on the scale of LLMs. It might be beneficial to pull out these subsections into it’s own section in the end (or in the appendix), and mention that while distributed learning has not been widely studied with LLMs yet, this space is likely to face similar privacy concerns.
>
>
> As the reviewer suggested, `Section 3.3` has been revisited to make it more centered around LLMs rather than the on the general approach. We summarized the contribution of relevant pre-LLM methods and analysis, while expanding on LLMs-based methods. Furthermore, we refactored the discussion on Federated Learning in `Section 5.3` by keeping works most relevant to autoregressive models in the main paper and moving to the appendix works less focused on LLMs. We still believe that Federated Learning remains pertinent for future developments and thus should be included in the discussion, even though at the moment of writing most of the models proposed are not particularly large due to the computational complexity intrinsic of this approach.

---

> > ### Comment · Reviewer_Fx2K · 2024-11-27
> > **Thank you for rebuttal**
> >
> > I would like to thank the authors for the time they took to make very thorough responses. In general, I think most of my comments have been addressed.

---

> > > ### Author Response · Authors · 2024-11-29
> > > **Thank you for your time and valuable feedback**
> > >
> > > We sincerely thank once again the reviewer for their time and valuable feedback

---

### Review · Reviewer_iggB · 2024-09-25

**Summary Of Contributions:**

This survey considers the problem of preserving privacy in large language models (LLMs) and presents existing privacy threats associated with LLMs, countermeasures, and tools for implementing them. The presentation of existing privacy attacks covers training data extraction, membership inference, and model inversion attacks. The solutions span privacy-preserving data publishing (k-anonymity and differential privacy-based), differential privacy at model level (pre-training with DP, public pretraining + private fine tuning with DP...), federated learning and machine unlearning. Finally, the presentation on existing tools covers open-source libraries for privacy-preserving data publication, differential privacy, and federated learning.

**Audience:**

Yes

**Claims And Evidence:**

Yes

**Requested Changes:**

- Provide some references for the definition of PII. I would suggest the NIST and GDPR definitions
- The preliminaries section can be shortened by removing subsections on machine learning and deep learning (just provide references to existing resources like books)
- Consider clarifying why input privacy at inference is not discussed
- Section 2 introduces black-box and white-box access settings. However, grey-box setting, which is later used in the paper, was never properly defined
- Consider defining sensitivity, Laplace/Gaussian/Exponential mechanisms in the background section, right after DP definition
- The definition Rényi differential privacy introduces a mechanism \mathcal{M} but another mechanism f is used in the inequality.
- Since the survey focuses on LLMs, I think the discussion on the solution should focus on the methods tested for such models. For example, the discussion on traditional anonymization techniques mentions techniques that leverage LLM to sanitize datasets. However, the datasets mentioned are not used to train LLM. So, these techniques do not qualify as solutions to enhance privacy in the context of using LLMs. I have a similar remark regarding anonymization with DP. The discussion should focus on large language models. Some discussion is needed on how these methods can be applied to the training data of large language models.
- For example, SANTEXT focuses on differentially private text analytics. Is it relevant to LLM's context?


- The section on Federated learning is excessively long, with focus on models/tasks that are not relevant to LLM. Consider focusing directly on LLM. Same for Unlearning


- Consider presenting tools that specifically focus on privacy-enhancing methods for LLM. For example this could be a table summarizing repositories to reproduce results, type of LLM supported, data modalities ...




## Minor
- I would suggest (Gentry 2009) as a reference for homomorphic encryption
- For oblivious transfer, I would suggest (Michael O. Rabin, 1981), (Even, Goldreich, Lempel, 1985), or (Brassard, Crépeau, Robert 1986)
- Page 19: "Furthermore, referring to Figure 5" => "Furthermore, referring to Section 5"

**Strengths And Weaknesses:**

## Strengths
- The survey focuses on privacy in the context of LLMs, which is an important research direction
- The organization is simple and easy to follow

## Weaknesses
- The paper sometimes provides excessive detail on aspects not central to understanding it (e.g., machine learning, deep learning) while offering too little detail on important aspects (differential privacy mechanisms such as Laplace/Gaussian/Exponential mechanisms)
- The informations presented in tabular format are redundant
- The current presentation of threats focuses only on the LLM, ignoring privacy threats from user perspectives (i.e., input privacy at inference time)
- LLMs can also be vulnerable to model stealing, which can also have privacy implications. I would like to see some discussion on that aspect
Some solutions/tools do not seem to apply directly to LLMs. Given the survey's focus, it would be great to focus on what can apply directly to LLMs.

---

> ### Author Response · Authors · 2024-11-14
> **Answer Reviewer iggB 1/2**
>
> We thank the reviewer for their feedback. We are pleased that our work has been recognized as an important research direction and that its clear, straightforward organization has been appreciated. In the remainder of this response, we address each specific point raised, along with the actions we have already implemented in the survey. We also invite the reviewer to refer to our general response for an overview of all the changes made.
>
>
> ## Weaknesses
>
> > - The paper sometimes provides excessive detail on aspects not central to understanding it (e.g., machine learning, deep learning) while offering too little detail on important aspects (differential privacy mechanisms such as Laplace/Gaussian/Exponential mechanisms)
>
>
> We thank the reviewer for this feedback. We fixed this by restructuring `Section 2 - Preliminaries` according to the suggestions provided.
>
>
> > - The informations presented in tabular format are redundant
>
> We thank the reviewer for pointing out this. We recognize that the tabular format may appear redundant, but it was a deliberate choice intended to provide a clear, high-level overview of the papers and topics covered in each section. This layout serves as a complementary guide, offering readers a concise and accessible summary of the content, which we believe adds value by making the information easier to navigate and synthesize. Since it is presented in the appendix, we believe it can assist interested readers without disrupting the main flow.
>
> > - The current presentation of threats focuses only on the LLM, ignoring privacy threats from user perspectives (i.e., input privacy at inference time)
>
> We have added clarification at the beginning of `Section 3`, please continue reading below for a comprehensive answer.
>
> > - LLMs can also be vulnerable to model stealing, which can also have privacy implications. I would like to see some discussion on that aspect Some solutions/tools do not seem to apply directly to LLMs. Given the survey's focus, it would be great to focus on what can apply directly to LLMs.
>
> We thank the reviewer for this valuable feedback. In response, we have added a dedicated discussion on model stealing, now located in `Section 3.3.3`, where we describe research specifically focused on LLMs and explain the foundations of model-stealing techniques. Additionally, we updated the first paragraphs of `Section 3.3` to reflect this new content.
>
> We also broadened our discussion to address how previously established techniques apply within the context of LLMs. For instance, we expanded the subsection on Membership Inference Attacks (MIA) with Thresholds to cover recent LLM-specific attack methodologies. Similarly, `Section 3.3.1` has been updated to include more recent work relevant to LLMs, ensuring that our survey remains directly applicable to these models.
>
> ## Requested Changes
>
> > - Provide some references for the definition of PII. I would suggest the NIST and GDPR definitions
>
> We thank the reviewer for their feedback. We have added references to the NIST and GDPR definitions of PII in the corresponding informative card in `Section 1 - Introduction`
>
> > - The preliminaries section can be shortened by removing subsections on machine learning and deep learning (just provide references to existing resources like books)
>
> We thank the reviewer for this suggestion. We have streamlined `Section 2 - Preliminaries` by removing the detailed subsections on Machine Learning and Deep Learning. Instead, we now provide references to well-established resources for these topics.
>
> > - Consider clarifying why input privacy at inference is not discussed.
>
> We thank the reviewer for this suggestion. We have added a new subsection `3.4 - Privacy Threats at Inference Time` regarding the privacy implications of including private information as input to an LLM. Specifically, we now discuss that inserting private information into prompts can lead to privacy threats not only because such prompts may be used as training data in the future, but also because LLM-powered systems -- prompted with private, in-context examples -- can be attacked by a malicious user of the system.
>
> > - Section 2 introduces black-box and white-box access settings. However, grey-box setting, which is later used in the paper, was never properly defined
>
> We have added a definition of grey-box access settings in `Section 2`.
>
> > - Consider defining sensitivity, Laplace/Gaussian/Exponential mechanisms in the background section, right after DP definition
> > - The definition Rényi differential privacy introduces a mechanism $\mathcal{M}$ but another mechanism $f$ is used in the inequality.
>
> We added the mentioned definitions in `Section 2` and fixed the definition of Rényi differential privacy.

---

> ### Author Response · Authors · 2024-11-14
> **Answer Reviewer iggB 2/2**
>
> > - Since the survey focuses on LLMs, I think the discussion on the solution should focus on the methods tested for such models. For example, the discussion on traditional anonymization techniques mentions techniques that leverage LLM to sanitize datasets.
>
> We thank the reviewer for this important point. We would like to clarify that the focus of our survey is not on using LLMs to preserve privacy (such as leveraging LLMs to sanitize datasets), but rather on privacy preservation within LLMs themselves. While it is true that the anonymization techniques discussed in Section 4 are not exclusive to LLMs, they can still be applied to anonymize textual data before it is used to train LLMs. Thus, these methods represent viable solutions for ensuring privacy in both the data and the LLMs, even though the anonymization process itself may not involve LLMs. To avoid confusion, we have added a clarification at the beginning of `Section 4 - Data` to better explain this distinction.
>
> > However, the datasets mentioned are not used to train LLM. So, these techniques do not qualify as solutions to enhance privacy in the context of using LLMs. I have a similar remark regarding anonymization with DP. The discussion should focus on large language models. Some discussion is needed on how these methods can be applied to the training data of large language models.
>
> Thank you for this valuable suggestion. To address this point, we have added a discussion at the beginning of `Section 4 - Data` to clarify how current data anonymization techniques are applied during the pretraining phase of large language models. We highlighted several LLMs works that used data anonymization during pretraining, connetting also these approaches with tools mentioned in Section 6. This addition helps to connect the anonymization methods directly to the context of LLMs, ensuring the survey focuses on privacy-enhancing solutions within the specific scope of LLMs.
>
>
> > - For example, SANTEXT focuses on differentially private text analytics. Is it relevant to LLM's context?
>
> While some of these methods, such as SANTEXT, do not directly demonstrate fine-tuning of LLMs with anonymized data, they can still be used to anonymize data to be used for training a LLM. Additionally, we note that some sanitization methods are also used to protect privacy by sanitizing prompts before they are fed into LLMs, making these approaches relevant for multiple aspects of LLM privacy. To clarify, we have also expanded our explanations of individual methods in `Section 4 - Data` to better highlight their relevance to LLM privacy.
>
> > - The section on Federated learning is excessively long, with focus on models/tasks that are not relevant to LLM. Consider focusing directly on LLM. Same for Unlearning
>
> We thank the reviewer for this feedback. We refactored the discussion on Federated Learning in `Section 5.3` by keeping works most relevant to autoregressive models in the main paper and moving to the appendix works less focused on LLMs. We still believe that Federated Learning remains pertinent for future developments and thus should be included in the discussion, even though at the moment of writing most of the models proposed are not particularly large due to the computational complexity intrinsic of this approach. Regarding the section on Machine Unlearning `Section 5.4`, we have streamlined it by summarizing key contributions from pre-LLM research at the start of the section, and then expanded the discussion to focus more directly on recent advancements in LLM-specific unlearning techniques for the remainder of the section.
>
> > - Consider presenting tools that specifically focus on privacy-enhancing methods for LLM. For example this could be a table summarizing repositories to reproduce results, type of LLM supported, data modalities ...
>
> We thank the reviewer for this suggestion. Given the rapid pace of new developments in this field, we intend to keep tables and references updated in a dedicated GitHub repository. Due to anonymity policies, we are currently unable to share this repository, but we plan to include it, along with the suggested tables, in the camera-ready version to provide a more comprehensive and up-to-date resource for readers.
>
> > - I would suggest (Gentry 2009) as a reference for homomorphic encryption
> > - For oblivious transfer, I would suggest (Michael O. Rabin, 1981), (Even, Goldreich, Lempel, 1985), or (Brassard, Crépeau, Robert 1986)
> > - Page 19: "Furthermore, referring to Figure 5" => "Furthermore, referring to Section 5"
>
> We thank the reviewer for all these suggestions, they have been updated in the paper.

---

> > ### Comment · Reviewer_iggB · 2024-11-29
> > **Thank you for your responses**
> >
> > I appreciate the authors' thorough responses. My comments have been adequately addressed, and I do not have any other questions.

---

> > > ### Author Response · Authors · 2024-11-29
> > > **Thank you for your time and valuable feedback**
> > >
> > > We sincerely thank once again the reviewer for their time and valuable feedback.

---

### Review · Reviewer_CECd · 2024-11-01

**Summary Of Contributions:**

The paper is a survey article on a broad range of privacy preserving techniques in AI, with a particular focus on Large Language models. The range of privacy techniques/ML topics covered include: classic anonymization, differential privacy, multi-party secure computations, federated learning, machine unlearning etc.

The survey starts with a detailed discussion on a wide range of privacy attack models: training data extraction (extract real examples that were used in the training), membership inference attacks (identify the training examples from a pool of examples) and model inversion (extracting the blackbox model using outputs and gradients.)

Subsequently, they present the broad range of solutions available in literature, discuss their limitations and applicability in various scenarios. These techniques include: classical anonymization, anonymization using differential privacy, fine tuning using differential privacy approaches, federated learning approaches within the context of differential privacy.

Finally, the survey discusses the topic of machine unlearning, where the goal is to "delete" some of the training examples from a trained model (especially, a large language model such as GPT2).

**Audience:**

Yes

**Claims And Evidence:**

Yes

**Requested Changes:**

I would like to see a few paragraphs included for Fully Homomorphic encryption based machine learning. This will make the survey article really and truly complete.

**Strengths And Weaknesses:**

Strengths:
1. Extremely comprehensive survey that deals with an extremely important topic of privacy preserving ML. I think this is a timely survey too, as the issues raised in this work are some of the foundational issues that we are already facing with LLMs.
2. The survey is painstaking well written and it makes reading it quite enjoyable.

Weaknesses:

1. Well, maybe it is too long -- but it is expected to be for such a broad area.
2. For such a long article, missing out homomorphic encryption based ML is unacceptable. It may be well advised to cover at least the latest and greatest works in this area.

---

> ### Author Response · Authors · 2024-11-14
> **Answer Reviewer CECd**
>
> We thank the reviewer for their review, acknowleding that our survey is comprehensive, timely and well-written.
>
> > - [W1] Too long
>
> We thank the reviewer for this feedback. We have streamlined `Section 4 - Privacy Attacks` and `Section 5.3 - Federated Learning` moving some content to the appendix. This adjustment helps address the concerns of other reviewers reducing the length on the main paper while maintaining comprehensive coverage of this broad area. It's worth noting that the length is comparable to other surveys in TMLR.
>
> > - [W2] Missing out homomorphic encryption
>
> We thank the reviewer for this feedback. We had previously included a brief mention of homomorphic encryption within the Secure Multi-Party Computation paragraph in `Section 2 - Preliminaries`. However, when this survey was initially conceived, there were limited applications of homomorphic encryption in the context of *generative* large language models, with most approaches focused on smaller text models. To better address this feedback, we have now expanded our discussion and dedicated a new section, `Section 5.5 - Homomorphic Encryption`, to cover recent approaches on homomorphic encryption in greater detail, highlighting their relevance and potential for LLM privacy.
>
> We thank the reviewer once again for their time and feedback. We kindly invite them to review the newly added section and let us know if there are any additional relevant works we may have missed in our discussion.

---

> > ### Author Response · Authors · 2024-11-29
> >
> > As the discussion period is coming to an end and we will no longer be able to provide responses, we kindly ask the reviewer to let us know if they have any additional comments or feedback they may have on the paper.

---

> > > ### Comment · Reviewer_CECd · 2024-11-29
> > > **Thank you authors**
> > >
> > > The authors have addressed all my comments thoroughly.. I have no further suggestions to add

---

> > > > ### Author Response · Authors · 2024-11-29
> > > > **Thank you for your time and valuable feedback**
> > > >
> > > > We sincerely thank the reviewer for their time and valuable feedback.

---

### Author Response · Authors · 2024-11-14
**General Answer**

We would like to thank all the reviewers for the time and attention devoted to this survey. We recognize that given its length this was no easy task and we truly appreciate the thorough and insightful feedback provided. We are happy that our work has been recognized as a timely and important research (`Reviewer CECd` and `Reviewer iggB`), valued for its depth and thoroughness (`Reviewer CECd` and `Reviewer Fx2K`), with a clear structure, taxonomy and engaging readability (`Reviewer CECd`, `Reviewer iggB` `Reviewer Fx2K`).

In this general response, we offer an overview of all the revisions made to address your concerns. We invite each reviewer to refer to their individual responses for a detailed account of how we approached specific observations.

An updated draft incorporating these changes has been uploaded to the platform. We kindly ask reviewers to take a look at the current changes and provide feedback on whether these revisions sufficiently address all of their concerns. A PDF diff tool should be available within OpenReview to facilitate this process; upon request, we can also upload a version of the PDF with the differences directly highlighted.

We thank you once again for your time and thoughtful input.

## Changelog
- `Section 1 - Introduction`: Updated informative card of PII (Suggestions by `Reviewer iggB`) and updated taxonomy.
- `Section 2 - Preliminaries`: Added definitions of Laplace/Gaussian/Exponential Mechanisms and fixed typo in Rényi Differential Privacy (Suggestions by `Reviewer iggB`)
- `Section 2 - Preliminaries`: Removed subsections on machine learning and deep learning (Suggestions by `Reviewer iggB`)
- `Section 2 - Preliminaries`: Clarified the definition of LLMs as autoregressive generative models and explained why insights from models like BERT are still important to LLM privacy (Suggestions by `Reviewer Fx2K`)
- `Section 3.2 - Membership Inference Attacks (MIA)`: We revisited this section to limit the space dedicated to non-LLM attacks (as suggested by `Reviewer Fx2K`). In particular:
    - We removed `Section MIA with Shadow Models` and summarized the more relevant findings at the beginning of `Section 3.2`
    - We expanded the discussion around threshold-base MIAs at the end of the corresponding subsection (previously `Section 3.2.2`, now `3.2.1 - MIA with Thresholds`)
- `Section 3.3 - Model Inversion and Stealing`: We revisited this Section by summarizing relevant pre-LLMs contributions (as suggested by `Reviewer Fx2K`) and added some relevant works to the discussion (`Reviewer Fx2K` and `Reviewer iggB`):
    - `Section 3.3.1 - Model Output Inversion`: We streamlined this Section removing pre-LLM contributions and adding them to the general discussion at the beginning of `Section 3.3`, while adding some newer and relevant contributions to the discussion (as suggested by `Reviewer Fx2K`)
    - `Section 3.3.3 - Model Stealing (New!)`: We discuss the possibility of model stealing in the LLM context (as suggested by `Reviewer iggB`)
- `Section 3.4 - Privacy Threats at Inference Time (New!)`: We added a new section discussing the risks related to prompting LLM with private data at inference time (as suggested by `Reviewer iggB`)
- `Section 4 - Data`: Added clarifications for why data anonymization is important in the context of LLMs (`Reviewer iggB`); Added LLMs works that use data anonymization techniques in the pretraining phase (`Reviewer iggB`).
- `Section 4.1 - Anonymization`: Added subparagraphs to increase readability (Suggestions by `Reviewer Fx2K`)
- `Section 5.3 - Federated Learning`:  Streamlined this section by keeping works relevant to LLMs in the main paper and moving other studies to the appendix. (Suggestions by `Reviewer iggB` and `Reviewer Fx2K`)
- `Section 5.4 - Machine Unlearning`: Streamlined this section focusing more directly on recent advancements in LLM-specific unlearning techniques and expanded the discussion with some relevant contributions (Suggestion by `Reviewer iggB`)
- `Section 5.5 (New!) - Homomorphic Encryption`: We added a new section with works using cryptography-based approaches (Suggestion by `Reviewer CECd`)
- `Section 7 (New!) - Limitations and Future Directions`: We added a new section discussing current limitations and future directions for privacy preservation in LLMs (Suggestion by `Reviewer Fx2K`)

---

### Decision · Action_Editor_9EoM · 2024-12-10

**Recommendation:** Accept with minor revision

**Comment:**

This is a survey article on broad areas of privacy preserving in LLMs. All reviewers lean towards acceptance, and think most of their concerns have been addressed. Some still have concerns around inadequate discussions of open questions and lack of scalability considerations. I agree with the reviewers that this submission can be published in TMLR with minor revision as follows.

* Add more discussions throughout the paper to highlight whether the mentioned tools/techniques/challenges/methods are unique with LLMs, or if they can be applicable to broad ML problems. Since this survey targets at LLMs, it’d be helpful to more clearly inform readers what is new with LLMs.

* Comment more on the scalability/efficiency properties of various measurements/approaches applied to large models.

* Figure 1: It is not clear why those papers are selectively highlighted in Figure 1 given that those areas have been studied extensively. The specific choices of papers seem random. I suggest changing paper citations to more fine-grained subareas or directions under those subtopics.

* It would be better to add a table of content to help with navigation.

**Audience:**

Yes.

**Claims And Evidence:**

This submission is a survey article, and it covers a broad range of research areas in privacy preserving LLMs including both attacks, definition, and defenses. Many papers are cited and well-discussed.

---

> ### Author Response · Authors · 2025-01-17
>
> We sincerely thank the action editor for their time and thoughtful feedback on our paper. We are delighted to know that our work has been recognized as a comprehensive discussion of the relevant literature.
>
>
>
> > - Add more discussions throughout the paper to highlight whether the mentioned tools/techniques/challenges/methods are unique with LLMs, or if they can be applicable to broad ML problems. Since this survey targets at LLMs, it’d be helpful to more clearly inform readers what is new with LLMs.
>
>
> We thank the action editor for this feedback. We decided to add some pieces of discussion to address the specificity of proposed methods whenever it was not clear enough, nor specified. This involves both introductory sections (`Section 3 - Attacks`,`Section 4 - Data`; `Section 5 - Model`) and specific methods discussed in subsections. We added some discussion around methods initially formulated to attack to other ML models. Additionally, we added comments on methods applied to relatively smaller Transformer-based models to explain how they could be generalized to larger models, with some comments on the possible difficulties of such generalization.
>
>
> > - Comment more on the scalability/efficiency properties of various measurements/approaches applied to large models.
>
> We thank the action editor for this suggestion. As per our previous answer, we expanded the discussions in many methods to make considerations and educated guesses on how the application to LLMs or LLM-sized datasets might turn out.
>
> > - Figure 1: It is not clear why those papers are selectively highlighted in Figure 1 given that those areas have been studied extensively. The specific choices of papers seem random. I suggest changing paper citations to more fine-grained subareas or directions under those subtopics.
>
> We thank the action editor for this feedback which helped us recognize that our selection of papers in Figure 1 was not as clear as intended. Our goal was to highlight key works in the referenced sections, but we agree that this approach could appear not really clear. To address this, we have decided to remove the highlighted papers and instead focus on providing factual information. This adjustment not only ensures a more balanced presentation but also makes the taxonomy more adaptable for future developments, as the structure will be less dependent on specific choices of references.
>
> > - It would be better to add a table of content to help with navigation.
>
> We thank the action editor for this feedback. We added a table of content to help users navigate the paper.
>
> ----
>
> ## Changelog
> - `Taxonomy`: We decided to remove the cited papers to avoid appearing prioritizing specific works and to ensure the taxonomy remains more adaptable to future developments, as the conceptual divisions are less likely to require updates.
> - `Section 3 - Attacks` :    We have added an additional discussion of the methods initially formulated to attack other ML models and indicated more clearly how they were generalised for the LLM scenario (`Section 3.2` and `3.3`).
> - `Section 4 - Data`: Added a general discussion on incorporating data methods into the pre-processing pipeline of an LLM; Included considerations on how these methods might scale for datasets of the size used in LLMs.
> - `Section 4.1 - Anonymization`: Expanded the discussion on several methods, focusing on their general applicability, relevance to LLM training data, and scalability to large datasets.
> - `Section 4.2 - Anonymization with Differential Privacy`: Added details to better explain the usage of these methods in the context of LLMs; Included considerations on how the proposed methods would scale for the substantial datasets required for LLM pre-training.
> - `Section 5 - Model`: Clarified the generalizability of DP-SGD.
> - `Section 5.1.1 - Training Large Language Models with Differential Privacy`: Added discussion to emphasize potential implications when scaling the proposed methods—particularly those applied to smaller language models—to larger models.
> - `Section 5.1.2 - Fine-Tuning with Differential Privacy`: Expanded on methods developed for other networks to clarify their suitability for LLMs; Highlighted methods specifically designed for LLMs; Added discussions to underline the scalability potential of applicable methods.
> - `Section 5.1.3 - Parameter Efficient Fine-Tuning with Differential Privacy`: Included additional discussion to highlight how these methods are particularly suitable for LLMs.
> - `Section 5.2 - Inference with Differential Privacy`: Added discussion to underscore the specificity of these methods to LLMs.
> - `Acknowledgements`: Added Acknowledgements.